# Pairwise Elimination with Instance-Dependent Guarantees for Bandits with Cost Subsidy

**Ishank Juneja, Carlee Joe-Wong & Osman Yağan**
Department of Electrical and Computer Engineering
Carnegie Mellon University
Pittsburgh, PA 15213, USA
`{ijuneja,cjoewong,oyagan}@andrew.cmu.edu`

## Abstract

Multi-armed bandits (MAB) are commonly used in sequential online decision-making when the reward of each decision is an unknown random variable. In practice however, the typical goal of maximizing total reward may be less important than minimizing the total cost of the decisions taken, subject to a reward constraint. For example, we may seek to make decisions that have at least the reward of a reference "default" decision, with as low a cost as possible. This problem was recently introduced in the Multi-Armed Bandits with Cost Subsidy (MAB-CS) framework. MAB-CS is broadly applicable to problem domains where a primary metric (cost) is constrained by a secondary metric (reward), and the rewards are unknown. In our work, we address variants of MAB-CS including ones with reward constrained by the reward of a known reference arm or by the subsidized best reward. We introduce the Pairwise-Elimination (PE) algorithm for the known reference arm variant and generalize PE to PE-CS for the subsidized best reward variant. Our instance-dependent analysis of PE and PE-CS reveals that both algorithms have an order-wise logarithmic upper bound on Cost and Quality Regret, making our policies the first with such a guarantee. Moreover, by comparing our upper and lower bound results we establish that PE is order-optimal for all known reference arm problem instances. Finally, experiments are conducted using the MovieLens 25M and Goodreads datasets for both PE and PE-CS revealing the effectiveness of PE and the superior balance between performance and reliability offered by PE-CS compared to baselines from the literature.

## 1 Introduction

Online sequential decision-making problems capture many applications where decisions must be made without knowing their outcomes in advance. After each decision, the resulting outcome or reward is observed, and an internal model is updated to improve future decisions. In clinical trials for example, the goal is to compare the therapeutic value of various drugs against an ailment. The decisions in this case represent administering a certain drug and the rewards are the apriori unknown efficacies of the candidate drugs. Communication networks are another example. Here decisions must be made about the communication channel to be employed. In this scenario the reward represents the success or lack thereof of communicating over a chosen channel. Multi-Armed Bandits (MABs) (Lattimore & Szepesvári, 2020) are a framework for *stateless* sequential decision making where the available decisions are abstracted as arms of a MAB problem instance. The stateless assumption implies that the distribution of rewards associated with an arm is not affected by the choices of past arms. The setting within MABs we work with is that of *stationary stochastic bandits* where the distribution of arm rewards does not evolve with time.

The generality of the assumptions imposed by the stationary stochastic bandits setting provides a wide net to capture a range of problem domains. However, in real-world applications there are often several competing objectives that go beyond the limited goal of maximizing reward. For instance consider the problem faced by a marketing agency where there are several communication modalities available to communicate the agency's advertising message. Blindly maximizing the overall success rate (reward) in this case would be naive. Since such an approach would ignore the

drastically different costs of using these modalities. Our example reveals that costs being associated with the sampling of any particular arm is a structure that appears quite naturally in applications.

In the marketing agency problem we know that the various communication modalities shall have unknown success rates for any brand new ad campaign. However, the cost of employing any modality will typically be known. These known arm sampling costs might manifest themselves in the form of a prescribed cost budget (Badanidiyuru et al., 2018), or as a metric whose cumulative value is to be minimized (Sinha et al., 2021), to specify two among many possible cost structures. We work with the latter among these settings. In particular our paper works with variants of the MAB with Cost-Subsidy (MAB-CS) framework introduced recently in Sinha et al. (2021).

## 1.1 Multi-Armed Bandits with Cost Subsidy (MAB-CS) Setting

What makes the MAB-CS setting so interesting is that it requires the bandit policy minimize cumulative costs while obtaining cumulative reward that is *feasible* and not necessarily maximal. The problem therefore involves dual objectives: minimizing costs while ensuring that the reward meets or exceeds the feasibility threshold, denoted by $\mu_{CS}$. For an MAB-CS problem instance with $K$ arms we use $c_i$ to denote the known cost associated with sampling any arm $i \in [K]$, and $\mu_i$ to denote the expectation of the reward received from sampling arm $i$. The *optimal arm* in our setting is then the least cost feasible arm. Mathematically if $S = \{i \in [K] : \mu_i \geq \mu_{CS}\}$ denotes the set of feasible arms, then the optimal arm $a^* = \arg\min_{i \in S} c_i$.

To honor the dual objectives of minimizing cost while maintaining feasible reward, cumulative cost and quality regret over a time horizon $T$, for instance $\nu$, with policy $\pi$ are defined as follows,

$$\text{Cost\_Reg}(T, \nu, \pi) = \sum_{t=1}^{T} \mathbb{E}_{\pi}\left[ (c_{k_t} - c_{a^*})^+ \right] = \sum_{i=1}^{K} \Delta_{C,i}^+ \mathbb{E}\left[ n_i(T) \right]$$

$$\text{Quality\_Reg}(T, \nu, \pi) = \sum_{t=1}^{T} \mathbb{E}_{\pi}\left[ (\mu_{CS} - \mu_{k_t})^+ \right] = \sum_{i=1}^{K} \Delta_{Q,i}^+ \mathbb{E}\left[ n_i(T) \right].$$

Where $k_t$ is the arm sampled by bandit policy $\pi$ in time slot $t$, and $n_i(T)$ is a random variable denoting the number times arm $i$ was sampled over $T$ time steps. The initial definitions of regret accumulated over horizon $T$ in the center have been re-written in terms of the expected number of samples of each of the $K$ arms at the right. Moreover we have used the definitions of cost gap $\Delta_{C,i} = (c_i - c_{a^*})$ and quality gap $\Delta_{Q,i} = (\mu_{CS} - \mu_i)$ in the decomposition. We highlight that as is standard for stationary stochastic bandits, the operator $\mathbb{E}_{\pi}$ represents the expectation over the choices of arm $k_t$ made by policy $\pi$ during time slot $t$. In the remainder of this section, we discuss three variants of MAB-CS, each of which have a different specification for $\mu_{CS}$. We end by motivating the importance of $\Delta_{C,i}^+, \Delta_{Q,i}^+$, where $x^+$ denotes $\max\{0, x\}$, in the regret definitions.

Take as an example the problem faced by a marketing agency. Consider that there are three modalities available for the agency to deliver their message, which are: (1) very expensive personalized door-step solicitation, (2) moderately expensive automated phone call, and (3) inexpensive email. Given these modalities, the agency's goal may be to achieve a prescribed sales rate with the minimum possible cost. Or, the goal may be that sales be at least a prescribed fraction of the sales of a certain communication modality. Finally, we may not have a reference mode in mind, and we may just desire a conversion rate that is (say) 80% as much as the highest unknown sales rate.

The first setting is captured by our novel contribution of the *fixed threshold* setting. In the fixed threshold setting we specify $\mu_{CS} = \mu_0$ for a known $\mu_0 \in \mathbb{R}$. We call the second setting the *known reference arm* setting which specifies $\mu_{CS} = (1 - \alpha)\mu_\ell$. Here $\ell$ is the index of the reference arm, and $\alpha \in [0, 1]$ is a known subsidy factor. The third setting was introduced in prior work and specifies $\mu_{CS} = (1 - \alpha)\mu^*$, where $\mu^*$ is the largest among expected rewards. In particular, $\mu^*$ is the expected reward from sampling arm $i^* = \arg\max_{i \in [K]} \mu_i$. Similar to the *known reference arm* setting $\alpha$ is the subsidy factor. We refer to this third setting as the *subsidized best reward* setting.

With the MAB-CS framework, we target applications that are agnostic to the level of quality as long as the quality exceeds threshold $\mu_{CS}$. This structure necessitates the zero-clipped operation inside the cost and quality regret definitions. Consider for example a problem where it is known that customers need to be provided a certain (possibly unknown) service quality level for them to

continue their subscription. Any quality on top of the feasibility level $\mu_{CS}$ would not improve the performance of our solution. For the marketing communication example from earlier, the quality threshold represents a sales conversation rate beyond which the profitability of the campaign is ensured. In this case too we would like decisions that are agnostic to sales success over $\mu_{CS}$.

In the absence of the zero-clipped structure, cost and quality regret that are sub-linear in horizon $T$ could be achieved for our examples in an unintended fashion. An algorithm that samples sub-optimal arms $i \neq a^*$ a linear fraction of times but balances positive regret from infeasible decisions, by negative regret from stellar decisions would satisfy the un-clipped quality constraint but would be unsuitable for our example applications.

## 1.2 Key Contributions

Our first contribution is to extend the MAB-CS framework to include two new settings. (1) The *fixed threshold* setting, and (2) The *known reference arm* setting. Second, we present instance dependent lower bounds on the expected number of pulls of any sub-optimal arm $i \neq a^*$ for our two novel settings as well as the third *subsidized best reward* setting from the literature.

Next, we present an original order optimal algorithm Pairwise Elimination (PE) for regret minimization in the known reference arm setting. Under PE non-reference arms are pit against the reference arm in the ascending order of their costs. PE uses a principled elimination based regret minimization algorithm called Improved-UCB (Auer & Ortner, 2010) to determine whether an arm provides feasible rewards. We show that our PE has an instance dependent upper bound on both expected cost and quality regret that is $\mathcal{O}(\log T)$, and that PE only samples arms more expensive than the optimal action $a^*$ at most a constant number of times under expectation.

Next, we develop a generalization of PE for the subsidized best reward setting called PE-CS. We show that PE-CS too admits an $\mathcal{O}(\log T)$ instance dependent upper bound on both cost and quality regret that involves both notions of conventional sub-optimality gaps and quality gaps. Not only is PE-CS the first algorithm for the subsidized best reward setting with instance dependent upper bounds on regret, but also PE-CS offers a substantial improvement in performance over the only other algorithm from the literature for the subsidized best reward setting which admits a guarantee, namely ETC-CS. Although we find that PE-CS is not order optimal for all instances, our contribution includes characterizing the class of instances for which PE-CS is order optimal.

## 2 Related Work

**Structured Bandits:** There have been numerous works that impose additional structure onto the stationary stochastic bandits problem with the goal of better addressing specific application domains. This structure can sometimes come in the form of relationships imposed on the rewards of arms. These reward-relationships may be known (Kleinberg et al., 2008) or unknown (Gupta et al., 2021). Adding constraints that depend on the risks associated with sampling the rewards of an arm as in Wu et al. (2016) or Chen et al. (2022) is another form of the structured problem.

**Bandits with Costs:** We contextualize the core contributions of this paper by comparing and contrasting our setting and methods with related ones from the literature. We build on the MAB-CS setting introduced in Sinha et al. (2021). A core component of the MAB-CS setting is that there is a known cost associated with sampling any arm that is specified as part of the problem instance. There have been numerous works within the MAB literature that include the notion that a price has to be paid for sampling an arm. Notably the Bandits with Knapsacks (Badanidiyuru et al., 2018) line of work also considers a setting with known costs. However, in Badanidiyuru et al. (2018) there is a limited cost-budget and reward must be optimized while satisfying strict budget constraints. In MAB-CS and its variants on the other hand, the goal is to minimize cumulative costs without there being any explicit constraints on cost. In MAB-CS, the constraints are in fact on reward, and are referred to as quality constraints.

**Bandit with Constraints:** The quality constraints in our work closely resemble the constraints on expected rewards that are imposed in the work on Conservative Bandits (Wu et al., 2016). In both our work and in Wu et al. (2016), there is a constraint that requires the accumulated reward to exceed a $(1 - \alpha)$ discounted version of the reward of a reference arm. The conservative bandits

setting only considers the cases where either the return of the reference arm is a known constant $\mu_0$, or the case where the reference arm is known, but its return is unknown. The primary difference in our work is that in addition to satisfying a quality constraint, in the MAB-CS setting, we must work to minimize the cumulative cost. The notion of costs are completely absent in Wu et al. (2016), moreover, in addition to the cases with a known reward threshold $\mu_0$, and a known reference arm with an unknown return, which we consider as novel extensions to the MAB-CS framework, we also address the problem of the original MAB-CS framework where the reference arm is the unknown best reward arm. Another key difference is that Wu et al. (2016) imposes the reward constraints in a cumulative anytime manner, whereas we impose it at every time-step.

**BAI and Improved UCB:** In our paper, we work with the notions of cost regret and quality regret which are identical to the ones introduced by Sinha et al. (2021), however unlike the setting in Sinha et al. (2021) which only considers the case where the reference reward comes from the so-far unidentified best reward arm, we consider the additional cases (1) Where there is a fixed known threshold to be exceeded, and (2) When there is a known reference arm $\ell$ whose reward $\mu_\ell$ has to be exceeded however $\mu_\ell$ itself is unknown. In addition to our novel PE-CS algorithm for the setting from Sinha et al. (2021) we present novel algorithms for our new settings (1) and (2) as well. In Sinha et al. (2021) the authors present three novel algorithms for the MAB-CS setting, the former two among which construct a set of empirically feasible arms by interleaving exploration and exploitation. We build up our approach to optimizing for the regret objectives by first solving the known reference arm $\ell$ with unknown reward $\mu_\ell$ setting using a successive elimination style algorithm that compares candidate arms one at a time against the reference arm to see if they are feasible. We call this approach Pairwise Elimination (PE), and we adapt the elimination based regret minimization algorithm Improved-UCB (Auer & Ortner, 2010) to develop it. Then we generalize PE to the case where the reference arm is the unknown best reward arm by prepending PE with a Best-Arm-Identification (BAI) stage and call this latter algorithm PE-CS.

## 3 Algorithms and Analysis

As discussed in Section 1, we introduce novel settings called: (1) Fixed threshold setting with $\mu_{CS} = \mu_0$, and (2) known reference arm setting with $\mu_{CS} = (1 - \alpha)\mu_\ell$. In interest of building up to our presentation on PE-CS we start with the known reference arm setting and relegate the discussion and analysis of the known threshold setting to Appendix H. For the known reference arm setting, we present the lower bound and our novel Pairwise-Elimination algorithm in Section 3.1.

Our PE algorithm builds upon Improved-UCB (Figure 1 in Auer & Ortner (2010)), a regret minimization algorithm for the stationary stochastic bandits setting. We choose to build on the method since its successive elimination approach to regret minimization readily incorporates our insight that arms be evaluated in the order of their costs. One of the core features of Improved-UCB is that the cadence of sampling and elimination is governed by *rounds*. Each round is a period where every active arm is sampled to the extent prescribed by a function of the *round number*. A round concludes when all active arms have been sampled sufficiently, and at the end of a round, unsatisfactory arms are eliminated. We inherit the use of these rounds and associated formulas from Improved-UCB.

Finally, our third setting has the reward threshold $\mu_{CS} = (1 - \alpha)\mu^*$. This subsidized best reward setting is strictly more challenging than the known reference arm $\ell$ setting since the best reward arm $i^*$ itself is unknown. To solve it, we extend the PE algorithm by pre-pending it with a Best-Arm-Identification (BAI) stage to develop the PE-CS algorithm. The lower bound for the setting and the PE-CS algorithm are presented in Section 3.2.

### 3.1 Known Reference Arm Setting

The known reference arm setting specifies $\mu_{CS} = (1 - \alpha)\mu_\ell$ as the subsidized (unknown) return of a known reference arm. Without loss of generality, we assume that all MAB-CS setting bandit instances have arms indexed in the ascending order of their costs. In the known reference arm setting any arm with cost higher than $c_\ell$ is necessarily sub-optimal and can be pruned out. Moreover, in any instance with $K$ arms, we can think of arm $\ell$ as the arm with index $K + 1$. First we provide an instance dependent lower bound on the expected number of samples of sub-optimal arms for a class of consistent policies. The definition of consistent policies is available in Appendix E.

**Theorem 3.1** (Lower bound for known reference arm setting)**.** *Under any consistent policy $\pi$ the expected number of samples of a low cost arm and of the reference arm $\ell$ are lower bounded as,*

$$\liminf_{T \to \infty} \frac{\mathbb{E}\left[n_i\left(T\right)\right]}{\log T} \geq \frac{2}{\Delta_{Q,i}^2}, \quad \text{for arms } i < a^*, \quad \liminf_{T \to \infty} \frac{\mathbb{E}\left[n_\ell\left(T\right)\right]}{\log T} \geq \max_{i \leq a^*} \frac{2(1-\alpha)^2}{\Delta_{Q,i}^2}.$$

*Where $T$ denotes the problem horizon and the rewards of all $K$ arms are Gaussian distributed with variance $\sigma^2 = 1$. Low cost is a term used relative to the cost of optimal arm $a^*$.*

The proof of Theorem 3.1 (available in Appendix E) uses analytic techniques and arguments similar to those used to establish lower bounds for the classical multi-armed bandit setting (Garivier et al., 2019). We now discuss our PE (Algorithm 1) for the known reference arm setting. Under this setting, quality regret is calibrated against the expected reward of the reference arm $\ell$ which we denote $\mu_\ell$. For jointly optimizing cost and quality regret, we take an approach where the feasibility of arms is evaluated in the order of the costs of the arms: cheapest first. This insight motivates a pairwise comparison between the reference arm $\ell$ and the non-reference candidate arms.

---

**Function 1:** Pairwise Elimination Function PE()

**Function** *PE( $n$: Sample Vector, $\hat{\mu}$: Empirical Means, $\omega$: Round Numbers, $T$: Horizon, $i$: Episode, $\ell$: Reference Arm, $\alpha$: Subsidy Factor )***:**

1    $\tilde{\Delta} \leftarrow 2^{-\omega_i}$

2    $\tau \leftarrow \left\lceil \frac{2\log(T\tilde{\Delta}^2)}{\tilde{\Delta}^2} \right\rceil$

3    **for** $k \in \{i, \ell\}$ **do**

4      **if** $n_k < \tau$ **then**

5        **return** $k, \omega, i$    `// k_t = k, round numbers ω and ep. i unchanged`

6    $\beta \leftarrow \sqrt{\frac{\log(T\tilde{\Delta}^2)}{2\tau}}$

7    **if** $(1 - \alpha)\left(\hat{\mu}_\ell + \beta\right) < \hat{\mu}_i - \beta$ **then**

8      **return** $i, \omega, None$      `// Declare i as winner, set episode to None`

9    **else if** $\hat{\mu}_i + \beta < (1 - \alpha)\left(\hat{\mu}_\ell - \beta\right)$ **then**

10      **return** $i + 1, \omega, i + 1$      `// Sample next candidate arm, Rounds ω`    `unchanged, Update episode to that of next candidate arm`

11    **else**

12      $\omega_i \leftarrow \omega_i + 1$   `// Increment round only for arm i being evaluated`

13      **return** $i, \omega, i$        `// Move to next round within same episode`

---

**Algorithm 1:** Pairwise Elimination (PE) for a known reference arm $\ell$

**Inputs:** Bandit Instance $\nu$, Horizon $T$, Reference Arm $\ell$, Subsidy Factor $\alpha$ .

**Initialize:** Samples $n_k = 0$, Empirical Means $\hat{\mu}_k = 0$, Current Rounds $\omega_k = 0$,        $\forall\, k \in [K] \cup \{\ell\}$, PE Episode $i = 1$, Time $t = 1$ .

1 **while** $t \leq T$ **do**

2    **if** $i \notin \{None, \ell\}$ **then**

3      $k_t, \omega, i \leftarrow \text{PE}(n, \hat{\mu}, \omega, T, i, \ell, \alpha)$      `// receive arm to be sampled,`    `updated round numbers, and updated episode number`

4    **else**

5      $k_t \leftarrow k_{t-1}$        `// sample winning arm for remaining budget`

6    $\hat{\mu}(t+1), n(t+1), t \leftarrow \text{sample\_and\_update}(k_t, \hat{\mu}(t), n(t), t)$      `// in Appendix B`

---

PE tracks and updates three bookkeeping variables. The number of times each arm has been sampled $n$, the empirical mean reward sampled from each arm $\hat{\mu}$, and the round ongoing[1] by each arm $\omega$.

---

[1]Although the round number $\omega_k$ achieved by an arm $k$ can be uniquely determined from its number of samples $n_k$, we track them separately for a more lucid presentation.

For PE we asses candidate arms in the ascending order of their cost by assigning an *episode* to each candidate arm. This is represented by the main loop in Algorithm 1 in which $i$ is used to refer to both the index of the arm being evaluated for feasibility and to the episode number associated with that arm. PE( ) keeps getting invoked until all $K$ candidate arms are evaluated and we reach episode $\ell$, or until when a lower cost feasible arm has been identified.

Inside the PE( ) subroutine the sample prescription $\tau$ is computed using the current round number $\omega_i$ associated with candidate arm $i$ (line 2). Once the sample prescription $\tau$ is met for both arm $i$ and arm $\ell$, we check for elimination. Since no samples are ever discarded, episodes that are further downstream will re-use samples of arm $\ell$ from prior episodes when $n_\ell > \tau$. If candidate arm $i$ is able to eliminate arm $\ell$ then the optimal arm has been identified. Else if arm $\ell$ is able to eliminate arm $i$, then the episode is incremented and we proceed to evaluating the next cheapest arm. If no elimination occurs, then we simply increment the round number $\omega_i$. For the PE algorithm, we show the upper bound on cumulative cost and quality regret stated in Theorem 3.2.

**Theorem 3.2** (Instance dependent upper bound on cumulative cost and quality regret for PE). *For bandit instance $\nu$, over horizon $T$, the expected cumulative cost regret $\mathbb{E}\left[Cost\_Reg\left(T,\nu\right)\right]$ and quality regret $\mathbb{E}\left[Quality\_Reg\left(T,\nu\right)\right]$ of the PE algorithm are upper bounded respectively as,*

$$
\underbrace{\left(1 + \max_{i \le a^*} \frac{32 \log\left(T\Delta_{Q,i}^2\right)}{\Delta_{Q,i}^2}\right)\Delta_{C,\ell}^+}_{\substack{\text{Contribution from arm } \ell \text{ under} \\ \text{nominal termination in PE episode } a^*}} + \underbrace{\left(\sum_{i=1}^{a^*}\frac{43}{\Delta_{Q,i}^2}\right)\Delta_{C,\ell}^+}_{\substack{\text{Contribution from arm } \ell \text{ under} \\ \text{mis-termination in PE episode } \le a^*}} + \underbrace{\frac{43}{\Delta_{Q,a^*}^2}\max_{a^*<i\le\ell}\Delta_{C,i}^+}_{\substack{\text{Contribution from episodes } >a^* \\ \text{in case of mis-termination during ep } a^*}} \quad,
$$

$$
\underbrace{\sum_{i=1}^{a^*-1}\left(\Delta_{Q,i} + \frac{32\log\left(T\Delta_{Q,i}^2\right)}{\Delta_{Q,i}}\right)}_{\substack{\text{Contribution from arms } i < a^* \text{ under} \\ \text{nominal termination in PE episode } a^*}} + \underbrace{\sum_{i=1}^{a^*-1}\frac{43}{\Delta_{Q,i}}}_{\substack{\text{Contribution from arms } i < a^* \text{ under} \\ \text{mis-termination in PE episode } \le a^*}} + \underbrace{\frac{43}{\Delta_{Q,a^*}^2}\max_{a^*<i<\ell}\Delta_{Q,i}^+}_{\substack{\text{Contribution from episodes } >a^* \\ \text{in case of mis-termination during ep } a^*}} \quad.
$$

There are two dimensions to the execution of PE being nominal. First, within an episode the worse quality arm should be eliminated after a reasonable amount of sampling. Second is that execution across episodes should terminate in episode $a^*$ (with arm $\ell$ eliminated). To prove Theorem 3.2 we separately bound the expected number of samples of arms with cost lower and higher than the optimal action $a^*$, and the reference arm $\ell$. To bound these samples we condition on the desirable and anticipated events at both the intra and inter episode levels and show that the probability of these desirable events not occurring is small. The details are available in Appendix F.

Overall, we find that not only does PE achieve cost and quality regret that are $\mathcal{O}\left(\log T\right)$, but also PE matches up to constant factors the lower bound on cost and quality regret implied by Theorem 3.1. To see this clearly, we breakdown Theorem 3.2 by contribution. For cost regret, the $\mathcal{O}\left(\log T\right)$ dependence on arm $\ell$ is contained in the first term, and the dependence on higher cost arms is a constant captured by the third. For quality regret the first term captures the $\mathcal{O}\left(\log T\right)$ dependence on low cost arms, and the entire contribution of higher cost arms is captured by a constant.

**Practical Extensions of PE.** We highlight that although PE makes comparisons between the candidate arms and the reference arm in a pairwise manner, samples of the reference arm are re-used across episodes. Since during most episodes the samples accrued shall be limited to ones of the candidate arm undergoing evaluation, this sample reuse is a key feature of the PE algorithm and endows it with good sample efficiency. In PE as presented in Algorithm 1, during an arbitrary episode evaluating the candidacy of arm $i$, the reference arm $\ell$ shall only ever have to be sampled if the samples of arm $i$ start to exceed the samples of reference arm $\ell$ that were already available. In practice, we can implement another version of PE called *asymmetric-PE*. Asymmetric-PE allows for a mismatch between the number of samples of arms $i$ and $\ell$ that go into computing the exploration bonus terms $\beta$. The details of the variant and an example comparing it to vanilla PE are available in Appendix C.

## 3.2 SUBSIDIZED BEST REWARD SETTING

**Theorem 3.3** (Lower bound for subsidized best reward setting). *Under any consistent policy $\pi$, the expected number of samples of low cost arms, high cost arms, and the best reward arm $i^*$ are lower*

*bounded respectively as,*

$$\liminf_{T\to\infty} \frac{\mathbb{E}\left[n_i\left(T\right)\right]}{\log T} \geq \frac{2}{\Delta_{Q,i}^2}, \ i < a^*. \quad \liminf_{T\to\infty} \frac{\mathbb{E}\left[n_i\left(T\right)\right]}{\log T} \geq \frac{2}{(\frac{\mu_{a^*}}{1-\alpha} - \mu_i)^2}, i > a^*, i \neq i^*.$$

$$\liminf_{T\to\infty} \frac{\mathbb{E}\left[n_{i^*}\left(T\right)\right]}{\log T} \geq 2(1-\alpha)^2 \max\left\{\frac{1}{\Delta_{Q,a^*}^2}, \max_{i<a^*} \frac{1}{\Delta_{Q,i}^2}\mathbf{1}[\min_{i<a^*}\Delta_{Q,i} \leq (1-\alpha)\Delta_{min}]\right\}.$$

*Where $\Delta_{min} = \mu^* - \max_{i\neq i^*} \mu_i$ is the smallest conventional gap, $T$ denotes the horizon and the rewards of all $K$ arms are Gaussian distributed with $\sigma^2 = 1$.*

The proof of Theorem 3.3 closely parallels the proof of Theorem 3.1 barring two salient differences. First, in Theorem 3.3 there is a non-trivial lower bound on the number of samples of high-cost arms. This is because unlike the known reference arm setting the arm $i^*$ and its reward $\mu^*$ are unidentified. Second, the lower bound on samples of $i^*$ is not only a function of quality gaps $\Delta_{Q,i}, i \leq a^*$, but also depends on the relationship between $\min_{i\leq a^*} \Delta_{Q,i}$ and the smallest conventional gap $\Delta_{min}$. Next we discuss the extension of PE to the subsidized best reward setting to develop PE-CS.

---

**Algorithm 2:** Pairwise Elimination for Cost Subsidy Problem (PE-CS).

---
**Inputs:** Bandit Instance $\nu$, Horizon $T$, Subsidy Factor $\alpha$.
**Initialize:** Samples $n_k = 0$, Empirical Means $\hat{\mu}_k = 0$, Current Rounds $\omega_k = 0 \ \forall k \in [K]$,
$\quad\quad\quad$ BAI Active Arms $\mathcal{A} = [K]$, Arm $\ell$ = None, PE Episode $i = 1$, Time $t = 1$ .

1 **while** $t \leq T$ **do**
2 $\quad$ **if** $len(\mathcal{A}) > 1$ **then**
3 $\quad\quad$ $k_t, \boldsymbol{\omega}, \mathcal{A} \leftarrow \text{BAI}(\boldsymbol{n}, \hat{\boldsymbol{\mu}}, \boldsymbol{\omega}, T, \mathcal{A})$ $\quad\quad$ // receive arm to be sampled,
$\quad\quad\quad$ updated round numbers, and updated active arms
4 $\quad\quad$ **if** $len(\mathcal{A}) = 1$ **then**
5 $\quad\quad\quad$ $\ell \leftarrow \mathcal{A}[0]$ $\quad\quad\quad\quad\quad\quad\quad$ // set $\ell$ to be identified best arm
6 $\quad\quad\quad$ **continue** $\quad\quad\quad\quad\quad\quad\quad\quad$ // ignore sample recommendation $k_t$
7 $\quad$ **else if** $i \notin \{None, \ell\}$ **then**
8 $\quad\quad$ $k_t, \boldsymbol{\omega}, i \leftarrow \text{PE}(\boldsymbol{n}, \hat{\boldsymbol{\mu}}, \boldsymbol{\omega}, T, i, \ell, \alpha)$ $\quad\quad$ // receive arm to be sampled,
$\quad\quad\quad$ updated round numbers, and updated episode number
9 $\quad$ **else**
10 $\quad\quad$ $k_t \leftarrow k_{t-1}$ $\quad\quad\quad\quad$ // sample winning arm for remaining budget
11 $\quad$ $\hat{\boldsymbol{\mu}}(t+1), \boldsymbol{n}(t+1), t \leftarrow \text{sample\_and\_update}(k_t, \hat{\boldsymbol{\mu}}(t), \boldsymbol{n}(t), t)$ $\quad\quad$ // in Appendix B

---

When solving the subsidized best reward setting we face the additional challenge that the arm $i^*$ is unidentified. To solve this problem, PE-CS (Algorithm 2) comprises two stages, the *BAI stage* and the *PE stage*. The BAI( ) subroutine uses the same *round number* based cadence of sampling and elimination leveraged in PE( ) and is specified in Appendix B, Function 3. Under BAI( ) once the set of active arms collapses to a single arm, sampling decisions are passed over to the PE stage.

The PE stage in the PE-CS algorithm works identically to the PE algorithm described in Algorithm 1. In fact, due to the modular and phased nature of PE-CS we use precisely the same subroutine Function 1 for PE( ). For the PE-CS algorithm, there is all the more reason to make best use of the accumulated samples of the reference arm since not only can $n_\ell$ accumulate during episodes of the PE stage, but also can accrue in the BAI stage where the empirical best reward arm $\ell$ is the last surviving and therefore most sampled arm. In passing control from the BAI stage to the PE stage, a key role is played by the ongoing rounds $\boldsymbol{\omega}$. Used in both BAI( ) and PE( ), $\boldsymbol{\omega}$ is responsible for tracking the round up to which any arm has been sampled at any point in time. Once the time comes for passing over control from BAI( ) to PE( ), the PE stage is able to pick up sampling and elimination checks in any of its pairwise comparison episodes right where the BAI stage left-off. For PE-CS we show upper bounds on expected cumulative cost and quality regret in Theorem 3.4.

**Theorem 3.4** (Instance dependent upper bound on cumulative cost and quality regret for PE-CS)**.** *For bandit instance $\nu$, over horizon $T$, the expected cumulative cost regret $\mathbb{E}\left[Cost\_Reg\left(T, \nu\right)\right]$ and*

*quality regret* $\mathbb{E}\left[Quality\_Reg\left(T,\nu\right)\right]$ *of the PE-CS algorithm are upper bounded respectively as,*

$$\Delta_{C,i^*}^+ \left(1 + \max_{\Delta \in \mathbf{\Delta}} \left\{\frac{32 \log\left(T\Delta^2\right)}{\Delta^2}\right\}\right) + \Delta_{C,i^*}^+ \left(\sum_{i=1}^{a^*-1} \frac{43}{\Delta_{Q,i}^2} + \left(\frac{32}{\Delta_{a^*}^2} + \frac{43}{\Delta_{Q,a^*}^2}\right)\right)$$

$$+ \max_{i > a^*, i \in [K]} \Delta_{C,i}^+ \left(\frac{32}{\Delta_{a^*}^2} + \frac{43}{\Delta_{Q,a^*}^2}\right) + \sum_{i > a^*, i \in [K]\backslash\{i^*\}} \Delta_{C,i}^+ \left(1 + \frac{32 \log\left(T\Delta_i^2\right)}{\Delta_i^2}\right)$$

$$+ \max_{i \in [K]} \Delta_{C,i}^+ \left(\frac{11}{\Delta_{min}^2} + \sum_{j \neq i^*} \frac{32}{\Delta_j^2}\right).$$

---

$$\sum_{i=1}^{a^*-1} \left(\Delta_{Q,i} + \frac{32 \log\left(T\Delta_{Q,i}^2\right)}{\Delta_{Q,i}}\right) + \sum_{i=1}^{a^*-1} \frac{43}{\Delta_{Q,i}} + \max_{i > a^*, i \in [K]} \Delta_{Q,i}^+ \left(\frac{32}{\Delta_{a^*}^2} + \frac{43}{\Delta_{Q,a^*}^2}\right)$$

$$+ \sum_{i > a^*, i \in [K]\backslash\{i^*\}} \Delta_{Q,i}^+ \left(1 + \frac{32 \log\left(T\Delta_i^2\right)}{\Delta_i^2}\right) + \max_{i \in [K]} \Delta_{Q,i}^+ \left(\frac{11}{\Delta_{min}^2} + \sum_{j \neq i^*} \frac{32}{\Delta_j^2}\right).$$

*Where* $\Delta_i = \mu^* - \mu_i$ *are conventional gaps,* $\Delta_{min} = \mu^* - \max_{i \neq i^*} \mu_i$ *is the smallest conventional gap, and* $\mathbf{\Delta} = \{\Delta_{min}\} \cup \{\Delta_{Q,j}\}_{j \leq a^*}$ *is defined to bound samples of arm* $i^*$ *under expectation.*

We bound the expected number of samples of any arm under PE-CS by first conditioning on the outcome of the BAI-stage being either proper ($\ell = i^*$) or improper ($\ell \neq i^*$). The phased nature of PE-CS admits a modular analysis where conditioned on the outcome of BAI being proper, the bounds on samples shown in Theorem 3.2 for PE hold with slight modifications. In particular the three leading terms in the PE-CS bounds correspond directly (in order) to the three terms in the PE bounds for cost and quality regret. The fourth term in both the PE-CS cost and quality regret bounds corresponds to samples of high cost arms accrued during the BAI stage and the fifth term is the constant contribution to regret from an improper BAI outcome. Proof is available in Appendix G.

Finally, as with Theorem 3.2, we see that both the expected cost and quality regret are bounded by a quantity that is $\mathcal{O}\left(\log T\right)$. On comparing the upper bounds in Theorem 3.4 to the regret lower bounds implied by Theorem 3.3 we find a gap between the two. This gap arises from the class of bandit instances where precisely identifying arm $i^*$ using BAI detracts from the actual goal of sampling cheap feasible arms. A more detailed discussion is available in Appendix E.

## 4 EXPERIMENTS

In Section 3 we presented the merits of the PE and PE-CS algorithms in the *known reference arm* and *subsidized best reward* MAB-CS settings. While the presentation so far has been based exclusively on theoretical bounds on cost and quality regret here we complement our theoretical analysis with a study of the empirical performance of our methods[2]. In particular, we compare PE and PE-CS with baselines from Sinha et al. (2021) on problem instances derived from real-world datasets. The real-world datasets we use are MovieLens 25M (Harper & Konstan, 2015) and Goodreads (Wan & McAuley, 2018). Next we describe how we make use of these datasets for our experiments.

The MovieLens 25M dataset consists of 25 million ratings for 62,000 movies rated by 162,000 users. It is a popular dataset for studying the performance of recommendation systems. The movie ratings in the dataset are from numerous users and on a 5 point scale. Moreover, every movie for which ratings are available in the dataset is tagged with one or more genres. The Goodreads dataset already comes organized into overlapping genres with a single genre containing between 36,514 and 335,449 books. Each book is associated with at least one numeric review, and the number of reviews associated with the books tagged with a certain genre range between 150 thousand and 3.5 million.

Our Goodreads and MovieLens experiments simulate a scenario where a book subscription service or movie streaming website attempt to recommend books and movies to their users. The available

---

[2]Code available at `https://github.com/ishank-juneja/bandits-with-costs`

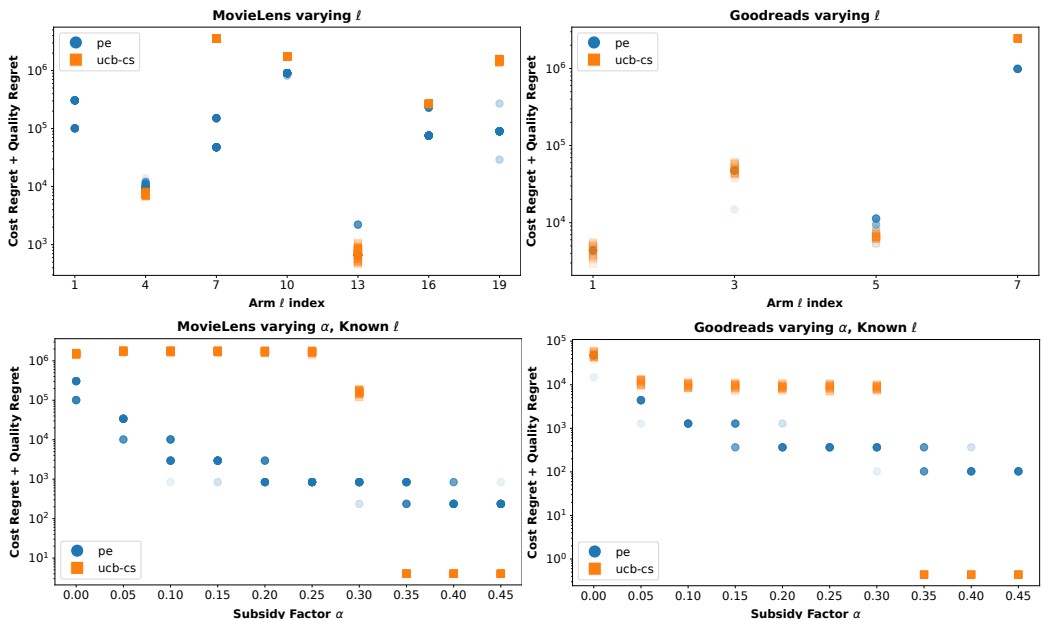

Figure 1: Fig. 1(a) varies the index $\ell$ for MovieLens. Fig. 1(b) does the same for Goodreads ($\alpha = 0$ in both). Fig. 1(c) fixes $\ell = 11$ and varies $\alpha$ for MovieLens while Fig. 1(d) fixes $\ell = 4$ for Goodreads and varies $\alpha$. Data points represent terminal regret at $T = 5M$ and each data point represents the outcome from an experiment. There are 25 such independent runs for each algorithm. There is no inherent notion of a reference arm in either dataset, so $\ell$ is picked arbitrarily.

selection is organized into genres and the service deploys a recommendation system to decide which genre of content be served to its user. Whenever a genre is chosen by the recommendation system, a random book or movie tagged with that genre is drawn with replacement. Under these conditions, bandit arms become a natural abstraction for genres. The cost associated with pulling the arm corresponding to a certain genre is simply the average of the royalties that must be paid to the authors or producers for every new reader of a book or streamer of a movie. Since royalty data is unavailable as part of either two datasets, we draw the costs associated with sampling any bandit arm (genre) to lie uniformly at random between 0 and 1. For every genre we first obtain the mean 5-point scale rating of all books or movies tagged with that genre and then divide this rating by 5 so that it lies between 0 and 1. We then treat this fractional rating as the expected reward return from that genre. Through this process we end up with bandit instances consisting of 20 arms for MovieLens and 8 arms for Goodreads the details of which are available in Appendix A.

In experiments discussed in Section 4 we plot the summed together values of the cost and quality regret. The goal in all MAB-CS settings is to converge onto sampling optimal arm $a^*$. To do so reliably we must explore sufficiently but eventually wane exploration. Looking at trends in summed regret allows us to compare the merit of various methods in achieving this goal. A tangential yet interesting goal is to understand the *trade offs* between cost and quality regret. To better showcase the performance of our methods we relegate trade offs to supplemental experiments in Appendix A.

### 4.1 EVALUATING OUR PAIRWISE ELIMINATION (PE) ALGORITHM

To understand the effectiveness of PE empirically, we compare PE to a natural variant of the UCB-CS algorithm from Sinha et al. (2021). In the specification of UCB-CS (Algorithm 5, Appendix B), the target reference arm (whose reward determines $\mu_{CS}$) is the best reward arm $i^*$. UCB-CS estimates the index of arm $i^*$ as the arm with the largest UCB-index in any time-slot. To develop a comparison with PE for the known reference arm setting, we simply replace this estimate with the known index of the reference arm while retaining the rest of the Algorithm. We call this variant as UCB-CS Known $\ell$ and compare it to PE on MovieLens and Goodreads in Figure 1. From the summed regret results in Figure 1 we find that our approach outperforms UCB-CS invariably for

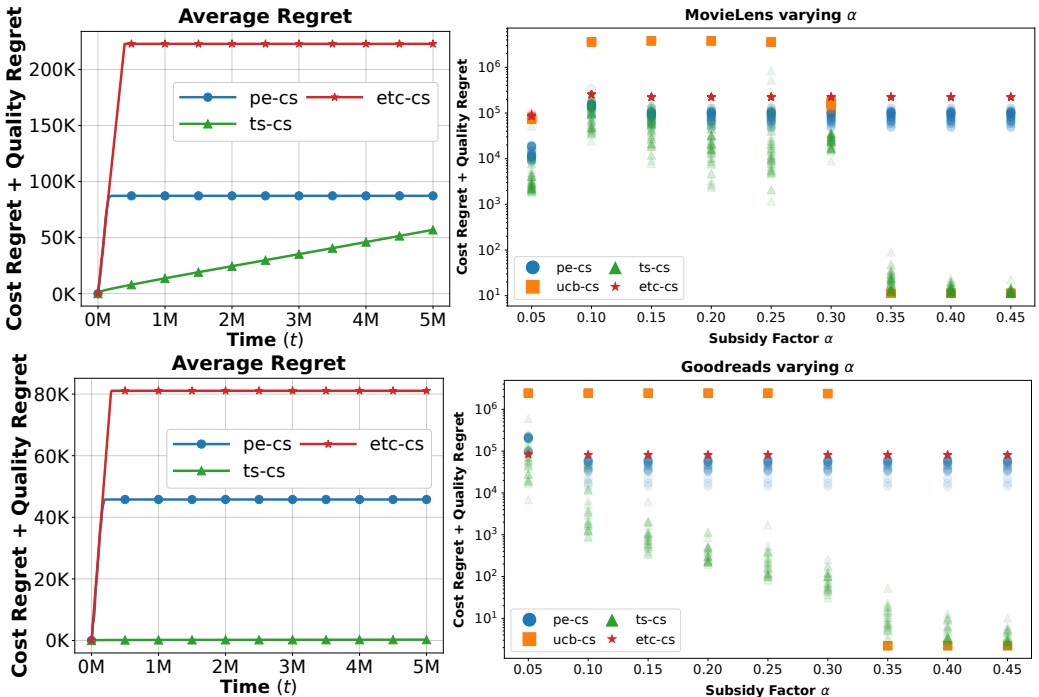

Figure 2: Fig. 2(a) shows the regret trend for MovieLens and Fig. 2(c) does the same for Goodreads. Both are for $\alpha = 0.25$. Similar to Fig. 1(c) and 1(d), Fig. 2(b) and 2(d) show the terminal regret trend ($T$ =5M, 50 runs of each algorithm). UCB-CS is omitted from (a) and (c) as its regret was orders of magnitude worse as can be seen from Fig. 2(b), 2(d).

smaller $\alpha$ and when the reference arm lies in the cheaper half of all available arms. We highlight that UCB-CS lacks any performance guarantees and is susceptible to linear regret, as demonstrated in Appendix A, Figure 7. Overall, we contend that UCB-CS is unreliable and typically performs worse than PE.

### 4.2 EVALUATING OUR PE-CS (PAIRWISE ELIMINATION COST SUBSIDY) ALGORITHM

In Section 3, we saw that PE-CS admitted logarithmic instance dependent guarantees on expected cumulative cost and quality regret. The only algorithm for the full cost-subsidy problem from the literature that has an upper bound guarantee on expected cumulative regret is the ETC-CS algorithm from Sinha et al. (2021). Moreover their work also prescribes the UCB-CS and TS-CS algorithms which are approaches to solving the subsidized best reward problem that interleave exploration and exploitation but lack any performance guarantees. The three algorithms ETC-CS, UCB-CS, and TS-CS comprise all the algorithms from the literature and are specified in Appendix B. We compare PE-CS against all three of these approaches in Figure 2.

ETC-CS consists of a pure exploration phase where each arm is sampled $\mathcal{O}\left(T^{2/3}\right)$ times followed by an exploitation phase. As the only other algorithm with a regret guarantee ETC-CS is our primary competitor. Moreover PE-CS also outperforms the UCB-CS baseline and the latter algorithm has a linear regret trend arising from a persistent mis-identification of the best action. Although we find that the average of the regret for TS-CS is lower than PE-CS, a closer examination of the regret trend (Fig. 2(a)) reveals the problem with the performance of TS-CS. While initially it takes PE-CS more exploration to lock onto the best action, it does so in a consistent and reliable way and once it does, there is no further incremental regret. Whereas for TS-CS, while interleaving exploration and exploitation leads to lower regret at the outset, there is a distinct slow-but steady upward trend in regret observed for the method.

## 5 ACKNOWLEDGMENTS

This research was generously supported by National Science Foundation (NSF) grant CNS-2103024, NSF grant CCF-2007834, Office of Naval Research (ONR) Grant N00014-23-1-2275, Cargenie Institute of Technology Dean's Fellowship and the CyLab Security and Privacy Lab at Carnegie Mellon University.

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

# A    SUPPLEMENT TO EXPERIMENTS

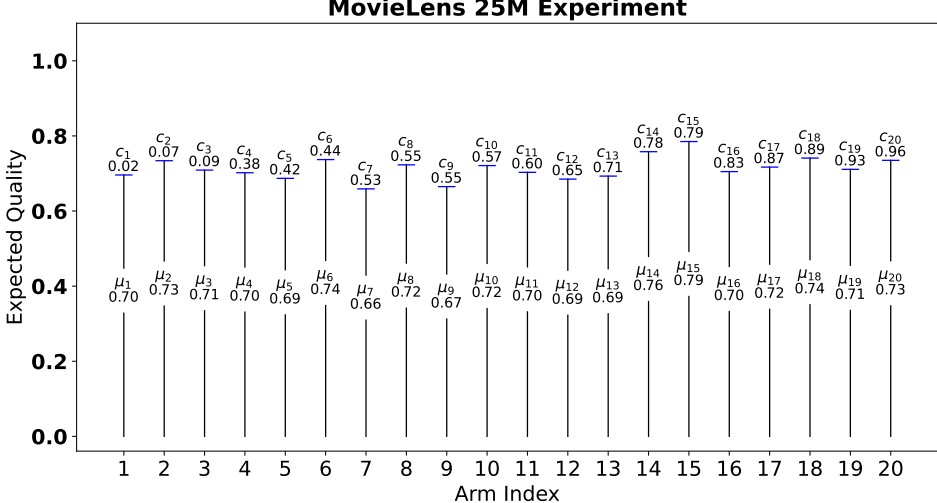

Figure 3: Problem instance for MovieLens 25M experiments

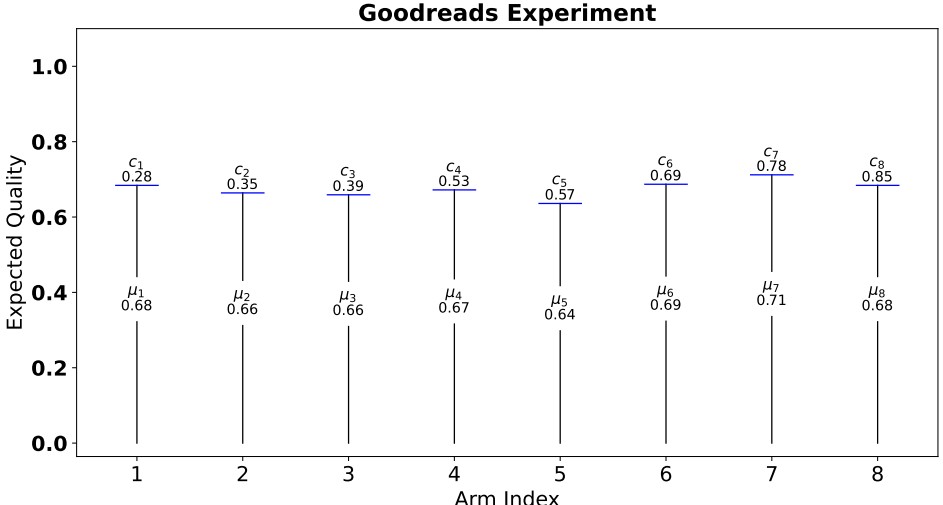

Figure 4: Problem instance for Goodreads experiment

EXTENDED COMMENTARY TO EXPERIMENTS SECTION

A close examination of the Algorithm descriptions for UCB-CS and TS-CS reveals the source of their unreliability. Both these algorithms work by constructing a set of empirically feasible arms and choose to sample the cheapest arm in this empirical set. This approach is vulnerable to a feasible arm being consistently excluded as a result of an initially poor performance. In our PE-CS on the other hand there is a systematic comparison between the arms where they are evaluated in the order of their costs and eliminated only when they have been sampled to be excluded with sufficient confidence.

To take a closer look at the sensitivity of PE-CS and all three baselines, we perform an additional synthetic experiment in the known reference arm setting which we call the toy experiment. For the toy experiment, we create a family of 12 subsidized best reward problem instances each with four arms. Then we run all four algorithms over 50 independent runs of each instance. The expected reward of the first arm in the instance varies uniformly in the range 0.6 through 0.93 over the 12

instances in the family. Since the optimal return in all instances is $\mu^* = 0.95$ and the subsidy factor is $\alpha = 0.2$, the reward threshold $\mu_{CS}$ for all the instances is $0.8 \times 0.95 = 0.76$.

We plot the results from the toy experiment on a scatter plot. On the y-axis is the summed terminal cost and quality regret (in $\log$ scale) and on the x-axis is the value of the varying expected return of the first arm of the instance family. Firstly, we find that on almost all instances, PE-CS performs either similar to or better than our primary comparator ETC-CS. Among the 12 instances tested here, the ones where the return of an arm is close to $\mu_{CS} = 0.76$ are the most challenging. We find that most runs of UCB-CS and several runs of TS-CS are unsuccessful at satisfactorily solving the $\mu_1 = 0.78$ case. Moreover, from Figure 8 we conclude that among the four algorithms tested here, PE-CS arguably offers the best balance between performance and reliability.

TRADE OFF BETWEEN COST AND QUALITY REGRET

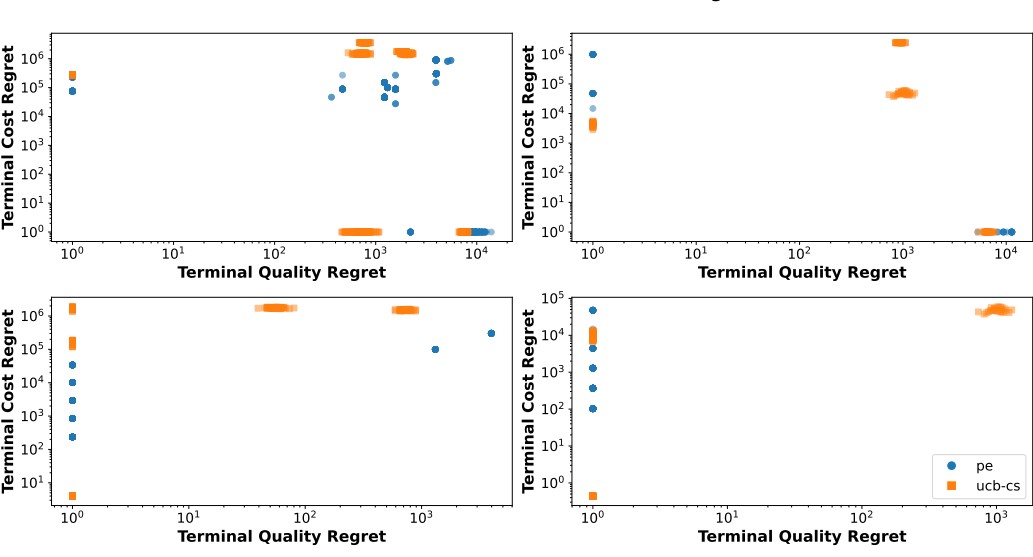

Figure 5: Trade off between cost and quality regret for subsidized best reward setting with Movie-Lens (left) and Goodreads (right). The data used for visualization remain the same as the experiment in Figure 2, Section 4.

Figure 6: Trade off between cost and quality regret for known reference arm setting with MovieLens (left) and Goodreads. The data used for visualization remain the same as the experiment in Figure 1, Section 4. Top two plots we use the data from the varying $\ell$ in known reference arm setting, and bottom two we use the data from the varying $\alpha$ with arbitrary fixed $\ell$ in known reference arm setting.

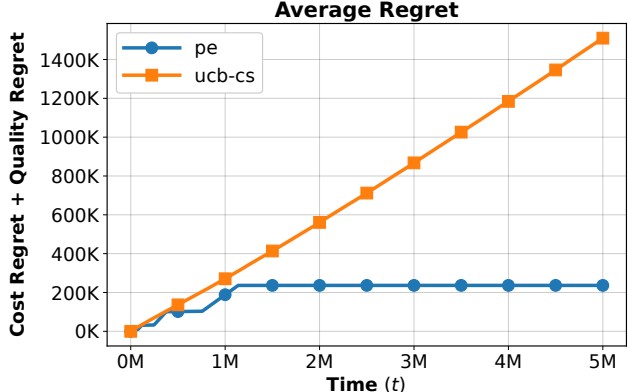

Figure 7: Fixed $\ell$ MovieLens Experiment. In the experiment: $\ell = 11$, $\mu_\ell = 0.703$, $\mu_{\text{CS}} = 0.668$.

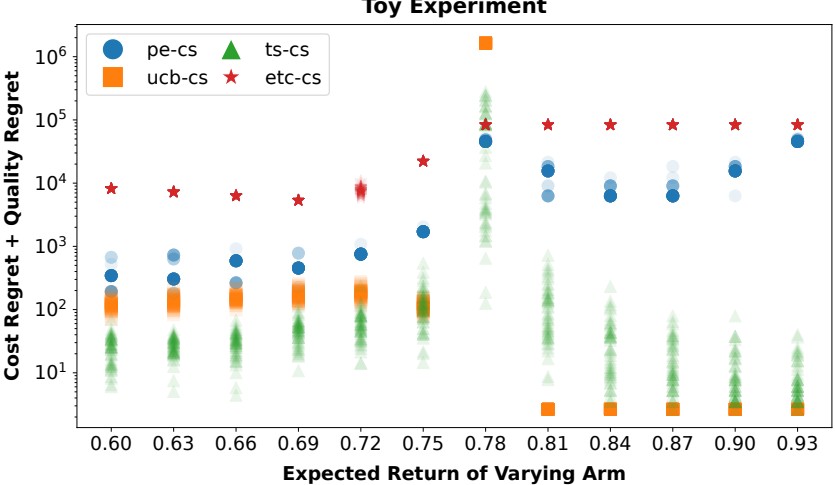

Figure 8: Toy Experiment performed using a family of four armed bandit instances. Expected rewards: $\mu_1 = 0.6, \ldots, 0.93, \mu_2 = 0.81, \mu_3 = 0.95, \mu_4 = 0.8$, and costs: $c_1 = 0.05, c_2 = 0.9, c_3 = 0.9, c_4 = 1.0$, subsidy factor: $\alpha = 0.2$.

## B  Appendix Algorithms

In this Section we collect all the Algorithm blocks relevant to our work that couldn't find space in the main paper. First we present the sample_and_update( ) subroutine used in Algorithm 1 and Algorithm 2 and also used in the baselines in the current Appendix section.

---

**Function 2:** Sample and Update Function to update variables in-place and return them

**Function** *sample_and_update( $k_t$: Chosen Arm, $\hat{\boldsymbol{\mu}}$: Empirical Means, $\boldsymbol{n}$: Sample Vector, $t$: Time-Step )*:

```
1    r_t ← sample(k_t)                              // sample reward for arm k_t
2    μ̂_{k_t}(t+1) ← (μ̂_{k_t}(t) × n_{k_t}(t) + r_t)/(n_{k_t}(t) + 1)        // update mean
3    n_{k_t}(t+1) ← n_{k_t}(t) + 1                   // update number of samples
4    return μ̂(t+1), n(t+1), t+1   // return updated means, samples, and
       time-step
```

---

Next we have the specification for the sub-routine BAI( ) used in PE-CS Algorithm 2.

---

**Function 3:** Best Arm Identification BAI()

**Function** *BAI( $\boldsymbol{n}$: Sample Vector, $\hat{\boldsymbol{\mu}}$: Empirical Means, $\boldsymbol{\omega}$: Round Numbers, $T$: Horizon, $\mathcal{A}$: Active Arms )*:

```
1    Δ̃ ← 2^{-max_{i∈[K]} ω_i}
2    τ ← ⌈ (2 log(T Δ̃²)) / Δ̃² ⌉
3    for k ∈ A do
4        if n_k < τ then
5            return k, ω, A   // next arm to be sampled, unchanged ω and A
6    β ← √( log(T Δ̃²) / (2τ) )
7    for i ∈ A do
8        μ_i^UCB, μ_i^LCB ← μ̂_i + β, μ̂_i − β
9    A⁺ ← {i ∈ A : μ_i^UCB ≥ max_{j∈A} μ_j^LCB}        // update set of active arms
10   k_t ← Uniform(A⁺)        // tentatively, the next arm to be sampled
11   for i ∈ A⁺ do
12       ω_i ← ω_i + 1   // increment round number for still active arms
13   return k_t, ω, A⁺
```

---

Next, we provide a precise specification of algorithms from prior work that we compare our methods against. In particular these are the ETC-CS, TS-CS, and UCB-CS Algorithms introduced by Sinha et al. (2021) and specified in Algorithms 3, 4, and 5 respectively. For the upper bound on regret provided in Sinha et al. (2021) to hold, we require the exploration budget $\tau$ to satisfy $\tau = c \left(\frac{T}{K}\right)^{\frac{2}{3}}$ where $c$ is some unspecified proportionality constant. Based on a few trial runs and examples we pick $c = 5$ since in the average case, the above seemed to give the best performance overall for the ETC-CS approach.

---

**Algorithm 3:** Cost-Subsidized Explore-Then-Commit (ETC-CS)

---

**Inputs:** Bandit instance $\nu$, Cost Vector **c**, Horizon $T$, Exploration Budget $\tau$.
**Initialize:** Empirical means $\hat{\mu}_k = 0$, Number of Samples $n_k = 0$, $\forall k \in [K]$.

1 **while** $t \leq K\tau$ **do**
2 $\quad$ $k_t \leftarrow t \mod K$
3 $\quad$ $\hat{\boldsymbol{\mu}}(t+1), \boldsymbol{n}(t+1), t \leftarrow \text{sample\_and\_update}(k_t, \hat{\boldsymbol{\mu}}(t), \boldsymbol{n}(t), t)$
4 **while** $K\tau < t \leq T$ **do**
5 $\quad$ **for** $i \in [K]$ **do**
6 $\quad\quad$ $\beta_i(t) \leftarrow \sqrt{\frac{2 \log T}{n_i(t)}}$
7 $\quad\quad$ $\mu_i^{\text{UCB}} \leftarrow \min\{\hat{\mu}_i(t) + \beta_i(t), 1\}$
8 $\quad\quad$ $\mu_i^{\text{LCB}} \leftarrow \max\{\hat{\mu}_i(t) - \beta_i(t), 0\}$
9 $\quad$ $\ell_t \leftarrow \arg\max_{i \in [K]} \mu_i^{\text{LCB}}(t)$
10 $\quad$ $\text{Feas}(t) \leftarrow \{i : \mu_i^{\text{UCB}}(t) \geq (1-\alpha)\mu_{\ell_t}^{\text{LCB}}(t)\}$
11 $\quad$ $k_t \leftarrow \arg\min_{i \in \text{Feas}(t)} c_i$
12 $\quad$ $\hat{\boldsymbol{\mu}}(t+1), \boldsymbol{n}(t+1), t \leftarrow \text{sample\_and\_update}(k_t, \hat{\boldsymbol{\mu}}(t), \boldsymbol{n}(t), t)$

---

**Algorithm 4:** Cost-Subsidized Thompson Sampling with Beta Priors (TS-CS)

---

**Inputs:** Bandit Instance $\nu$, Cost Vector **c**, Subsidy Factor $\alpha$, Beta Priors and Binomial
$\quad\quad\quad$ Likelihood (Bernoulli Rewards).
**Initialize:** Successes $S_k = 0$, Failures $F_k = 0 \; \forall k \in [K]$.

1 **while** $t \leq K$ **do**
2 $\quad$ $k_t \leftarrow t$
3 $\quad$ $r_t \leftarrow \text{sample}(k_t)$
4 $\quad$ $S_{k_t}(t+1) \leftarrow S_{k_t}(t) + r_t$
5 $\quad$ $F_{k_t}(t+1) \leftarrow F_{k_t}(t) + r_t$
6 $\quad$ $t \leftarrow t + 1$
7 **while** $K < t \leq T$ **do**
8 $\quad$ **for** $i \in [K]$ **do**
9 $\quad\quad$ $\theta_i(t) \sim \text{Beta}(S_i(t) + 1, F_i(t) + 1)$
10 $\quad$ $\ell_t \leftarrow \arg\max_{i \in [K]} \theta_i(t)$
11 $\quad$ $\text{Feas}(t) \leftarrow \{i : \theta_i(t) \geq (1-\alpha)\theta_{\ell_t}(t)\}$
12 $\quad$ $k_t \leftarrow \arg\min_{i \in \text{Feas}(t)} c_i$
13 $\quad$ $r_t \leftarrow \text{sample}(k_t)$
14 $\quad$ $S_{k_t}(t+1) \leftarrow S_{k_t}(t) + r_t$
15 $\quad$ $F_{k_t}(t+1) \leftarrow F_{k_t}(t) + (1 - r_t)$

---

## C  ASYMMETRIC PAIRWISE ELIMINATION

Algorithms 1 and 2 as described in Section 3 use Function 1 PE() as a sub-routine. While in PE() the round number $\omega_i$ is used to determine the stipulated number of samples for both the candidate arm $i$ and the reference arm $\ell$, this does not have to be the case. By the time we commence episode $i$ to evaluate the candidacy of arm $i$, we would have already accrued numerous samples of arm $\ell$. In particular, we denote the number of samples of arm $\ell$ as $n_\ell(t) = \tau_{\omega_\ell}$, where the vector $\boldsymbol{\omega}$, first introduced in Section 3, is a vector recording highest round up to which the samples of each arm have evolved. Consequently, $\omega_\ell$ is the highest round number reached for the samples of reference arm $\ell$.

Based on this observation about the greater progressed round number $\omega_\ell$ we create a variant of PE called Asymmetric-PE in Function 4 that replaces Function 1 in Algorithm 1.

Asymmetric-PE described in Function 4 has all the same inputs that conventional PE did. In addition, it has an input $\kappa$ called the Maximum round deviation. While we have no bound on how much

---

**Algorithm 5:** Cost-Subsidized UCB (UCB-CS)

**Inputs:** Bandit Instance $\nu$, Cost Vector $\mathbf{c}$, Horizon $T$, Subsidy factor $\alpha$.
**Initialize:** Empirical means $\hat{\mu}_k = 0$, Number of Samples $n_k = 0 \ \forall \, k \in [K]$.

1 **while** $t \leq K$ **do**
2     $k_t \leftarrow t$
3     $\hat{\boldsymbol{\mu}}(t+1), \boldsymbol{n}(t+1), t \leftarrow \text{sample\_and\_update}(k_t, \hat{\boldsymbol{\mu}}(t), \boldsymbol{n}(t), t)$

4 **while** $K < t \leq T$ **do**
5     **for** $i \in [K]$ **do**
6        $\beta_i(t) \leftarrow \sqrt{\frac{2 \log T}{n_i(t)}}$
7        $\mu_i^{\text{UCB}} \leftarrow \min\left\{\hat{\mu}_i(t) + \beta_i(t), 1\right\}$
8     $\ell_t \leftarrow \arg\max_{i \in [K]} \mu_i^{\text{UCB}}(t)$
9     $\text{Feas}(t) \leftarrow \left\{i \in [K] : \mu_i^{\text{UCB}} \geq (1-\alpha) \times \mu_{\ell_t}^{\text{UCB}}\right\}$
10     $k_t \leftarrow \arg\min_{i \in \text{Feas}(t)} c_i$
11     $\hat{\boldsymbol{\mu}}(t+1), \boldsymbol{n}(t+1), t \leftarrow \text{sample\_and\_update}(k_t, \hat{\boldsymbol{\mu}}(t), \boldsymbol{n}(t), t)$

---

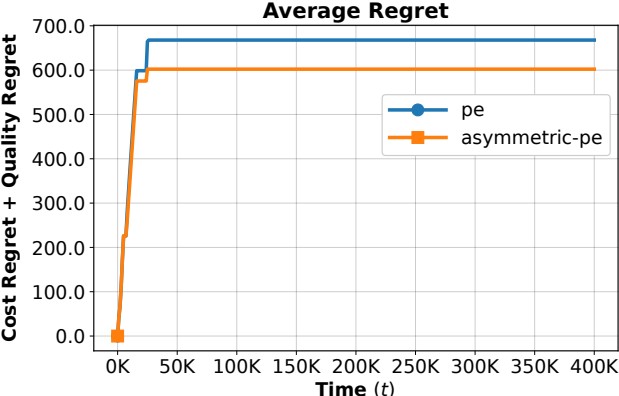

Figure 9: An experiment that illustrates potential savings in regret from the asymmetric-pe optimization. Bandit instance with Expected rewards: $\mu_1 = 0.74, \mu_2 = 0.5, \mu_3 = 0.8, \mu_4 = 0.75$, and costs: $c_1 = 0.15, c_2 = 0.2, c_3 = 0.21, c_4 = 0.25$, subsidy factor: $\alpha = 0.0, \ell = 3$.

larger $\omega_\ell$ is compared to $\omega_i$, when it comes to inferring the gap $\tilde{\Delta}_\ell$ corresponding to arm $\ell$ in Line 2 of Function 4, we restrict the round number we use to be at most $\kappa$ larger than the round $\omega_i$.

Effectively, the use of the further advanced round number $\omega_\ell$ trades performance for tightness of the upper bound (as we shall see in the analysis of Section F). We get an improvement in performance since $\beta_\ell \leq \beta_i$ potentially leading to a resolution in lines 7 or 9 of Function 4 with a smaller value of $\beta_i$ thereby requiring fewer samples of candidate arm $i$.

---

**Function 4:** Asymmetric Pairwise Elimination Function Asymmetric_PE()

---

**Function** *Asymmetric_PE( $\hat{\boldsymbol{\mu}}$: Empirical Means, $\ell$: Reference Arm, $\boldsymbol{n}$: Sample Vector, $T$: Horizon, $\boldsymbol{\omega}$: Round Numbers, $i$: Episode, $\alpha$: Subsidy Factor, $\kappa$: Max Round Deviation )*:

1    **for** $k \in \{i, \ell\}$ **do**

2      $\tilde{\Delta}_k \leftarrow 2^{-\min\{\omega_i + \kappa, \omega_k\}}$      // when $k = i$ we use $\omega_i$, if $k = \ell$, we use the smaller of $\omega_\ell$ or $\omega_i + \kappa$.

3      $\tau_k \leftarrow \left\lceil \frac{2\log(T\tilde{\Delta}_k^2)}{\tilde{\Delta}_k^2} \right\rceil$

4      **if** $n_k < \tau_k$ **then**

5        **return** $k, \boldsymbol{\omega}, i$    // $k_t = k$, round numbers $\omega$ and ep. $i$ unchanged

6      $\beta_k \leftarrow \sqrt{\frac{\log(T\tilde{\Delta}_k^2)}{2\tau_k}}$

7    **if** $(1 - \alpha)(\hat{\mu}_\ell + \beta_\ell) < \hat{\mu}_i - \beta_i$ **then**

8      **return** $i, \boldsymbol{\omega}, None$      // Declare $i$ as winner, set episode to None

9    **else if** $\hat{\mu}_i + \beta_i < (1 - \alpha)(\hat{\mu}_\ell - \beta_\ell)$ **then**

10      **return** $i + 1, \boldsymbol{\omega}, i + 1$      // Sample next candidate arm, Rounds $\omega$ unchanged, Update episode to that of next candidate arm

11    **else**

12      $\omega_i \leftarrow \omega_i + 1$          // Always increment round of arm $i$

13      $\omega_\ell \leftarrow \max\{\omega_i, \omega_\ell\}$    // $\omega_\ell$ unchanged until $\omega_i > \omega_\ell$. After which $\omega_\ell$ tracks $\omega_i$

14      **return** $i, \boldsymbol{\omega}, i$        // Sample arm $i$, Stay in same episode

---

# D PRELIMINARIES

In this section of the appendix, we collate the preliminary results required for the analysis of our cost-subsidy framework algorithms that follow in the forthcoming sections. The results stated without proof are standard results from the MAB literature.

**Definition D.1** (Subgaussian Random Variable). *We say that $X$ is $\sigma$-subgaussian if for any $\epsilon \geq 0$,*

$$\Pr\left(X - \mathbb{E}\left[X\right] \geq \epsilon\right) \leq \exp\left(\frac{-\epsilon^2}{2\sigma^2}\right). \tag{1}$$

**Lemma D.1** (Bounded random variables are Subgaussian, Example 5.6(c) in Lattimore & Szepesvári (2020)). *If Random Variable $X \in [a,b]$ almost surely, then $X$ is $\frac{b-a}{2}$ subgaussian.*

**Lemma D.2** (Hoeffding Bound, Section 5.4 in Lattimore & Szepesvári (2020)). *Let $X_1, X_2, \ldots, X_n$ be $n$ independent random variables, each bounded within the interval $[a,b]$ : $a \leq X_i \leq b$. The empirical mean of these variables is given by,*

$$\bar{X} = \frac{1}{n}\sum_{i=1}^{n} X_i. \tag{2}$$

*Then Hoeffding's inequality states that,*

$$\Pr\left(\bar{X} - \mathbb{E}\left[X\right] \geq t\right) \leq \exp\left(-\frac{2nt^2}{(b-a)^2}\right). \tag{3}$$

**Lemma D.3** (Iterated expectation over mutually exclusive and exhaustive events). *Let $X$ be any integrable random variable over probability space $(\Omega, \mathcal{F}, \Pr)$, and let $\{E_i\}_{i=1}^{n}$ be a collection of mutually exclusive and exhaustive measurable events. That is $\bigcup_{i=1}^{n} E_i = \Omega$ and $E_i \cap E_j = \phi, \forall i, j \in [n], i \neq j$. Then the following identity holds,*

$$\mathbb{E}\left[X\right] = \sum_{i=1}^{n} \mathbb{E}\left[X \mid E_i\right] \Pr\left(E_i\right). \tag{4}$$

*As a special case if the events are just some $E$ and its complement $E^c$, then,*

$$\mathbb{E}\left[X\right] = \mathbb{E}\left[X \mid E\right] \Pr\left(E\right) + \mathbb{E}\left[X \mid E^c\right] \Pr\left(E^c\right). \tag{5}$$

*Proof.* Define a sub $\sigma$-algebra of $\mathcal{F}$, $\mathcal{G} = \{\phi, E_1, E_2, \ldots, E_n, \Omega\}$. Then,

$$\mathbb{E}\left[X\right] = \mathbb{E}\left[\mathbb{E}\left[X \mid \mathcal{G}\right]\right] \text{ (Because } \mathcal{G} \subset \mathcal{F}) \tag{6}$$

$$= \sum_{i=1}^{n} \mathbb{E}\left[X \mid E_i\right] \Pr\left(E_i\right). \tag{7}$$

$\square$

**Lemma D.4** (Expectation is at most equal to larger of the conditioned expectations). *Let $X$ be any integrable random variable over probability space $(\Omega, \mathcal{F}, \Pr)$, and let $\{E_i\}_{i=1}^{n}$ be a collection of mutually exclusive and exhaustive measurable events. That is $\bigcup_{i=1}^{n} E_i = \Omega$ and $E_i \cap E_j = \phi, \forall i, j \in [n], i \neq j$. Then,*

$$\mathbb{E}\left[X\right] \leq \max_{i \in [n]} \left\{\mathbb{E}\left[X \mid E_i\right]\right\}. \tag{8}$$

*Proof.* Lemma D.4 can be considered a Corollary to Lemma D.3 as is illustrated by the following proof,

$$\mathbb{E}\left[X\right] = \sum_{i=1}^{n} \mathbb{E}\left[X \mid E_i\right] \Pr\left(E_i\right) \text{ (From the proof of Lemma D.3).} \tag{9}$$

$$\leq \left( \sum_{i=1}^{n} \Pr\left(E_i\right) \right) \cdot \left( \max_{i\in[n]} \left\{ \mathbb{E}\left[X \mid E_i\right] \right\} \right) \tag{10}$$

$$= \max_{i\in[n]} \left\{ \mathbb{E}\left[X \mid E_i\right] \right\}. \tag{11}$$

$\square$

**Lemma D.5** (Regret Decomposition Lemma, Lemma 4.5 in Lattimore & Szepesvári (2020)). *For any policy $\pi$ and stochastic bandit environment $\nu$ with $K$ arms, for horizon $T$, the Expected Cumulative Regret $Reg_\pi(T, \nu)$ of policy $\pi$ in $\nu$ satisfies,*

$$\mathbb{E}\left[Reg_\pi(T, \nu)\right] = \sum_{i\in[K]} \Delta_i \mathbb{E}\left[n_i(T)\right]. \tag{12}$$

*This result may be trivially generalized to other notions of regret where the gap determining the incremental regret due to arm $i$ is some arbitrary $\Delta_{X,i}$. In this case, the regret decomposition shall be,*

$$\mathbb{E}\left[Reg_\pi^X(T, \nu)\right] = \sum_{i\in[K]} \Delta_{X,i} \mathbb{E}\left[n_i(T)\right]. \tag{13}$$

*In particular in our problem we have Cost and Quality regret which are,*

$$\mathbb{E}\left[Cost\_Reg(T, \nu)\right] = \sum_{i\in[K]} \Delta_{C,i} \mathbb{E}\left[n_i(T)\right] \tag{14}$$

$$\mathbb{E}\left[Quality\_Reg(T, \nu)\right] = \sum_{i\in[K]} \Delta_{Q,i}^+ \mathbb{E}\left[n_i(T)\right]. \tag{15}$$

# E    INSTANCE DEPENDENT LOWER BOUNDS FOR MAB-CS

## LOWER BOUNDS FOR THE KNOWN REFERENCE ARM SETTING

We consider the case where the reference arm, denoted by arm $\ell$, is known. Let $\nu = \{\nu_i\}_{i=1,\ldots,K,\ell}$ denote the vector of reward distributions associated with all arms $i = 1, \ldots, K, \ell$, where the index of the reference arm can be thought of as $\ell = K + 1$. Let $\mathcal{M}$ denote the *model* under consideration that is a collection of all possible instances $\nu$. Recall that without loss of generality we assume the arm indices are cost ordered, i.e., $c_1 < c_2 < \cdots < c_K < c_\ell$, and that $a^*$ denotes the optimal action defined as the cheapest arm whose mean reward meets the *feasibility* threshold $\mu_{\mathrm{CS}}$. In the known reference arm setting, the feasibility threshold is defined as $\mu_{\mathrm{CS}} = (1 - \alpha)\mu_\ell$ for some $\alpha \in [0, 1)$ where $\mu_\ell$ is the mean reward of the reference arm $\ell$, which yields $a^* = \arg\min_{\{i:\mu_i \geq (1-\alpha)\mu_\ell\}} c_i$.

**Definition E.1** (Consistent Policies). *A policy $\pi$ is consistent if for all bandit instances $\nu$ and for all arms $i \neq a^*$, $\mathbb{E}\left[n_i(T)\right] = o(T^\gamma)$ for all $0 < \gamma \leq 1$.*

**Definition E.2** (The key quantity $D_{\mathrm{inf}}$). *Let KL denote the Kullback-Leibler divergence between two probability distributions. Given a distribution $\nu_i \in \mathcal{M}$ and a real number $x$, we define*

$$D_{inf}(\nu_i, x) = \inf\left\{KL(\nu_i, \nu_i') : \ \nu_i' \in \mathcal{M} \quad and \quad E[\nu_i'] > x\right\},$$

*where $E[\nu_i']$ denotes the mean of the distribution $\nu_i'$. We also need a slightly modified version of this key quantity defined as follows:*

$$\tilde{D}_{inf}(\nu_i, x) = \inf\left\{KL(\nu_i, \nu_i') : \ \nu_i' \in \mathcal{M} \quad and \quad E[\nu_i'] \leq x\right\}.$$

*Although not always explicitly stated, we will be using $D_{inf}(\nu_i, x)$ for distributions $\nu_i$ with $\mu_i < x$ and $\tilde{D}_{inf}(\nu_i, x)$ for distributions $\nu_i$ with $\mu_i > x$.*

Proofs of both Theorem E.2 and Theorem E.1 follow arguments similar to those used in establishing the well-known lower bounds for the classical multi-armed bandit setting. Here, we adopt the approach presented in Garivier et al. (2019) that is based on the following fundamental inequality (Garivier et al. (2019)[F]): For any $\nu, \nu'$,

$$\sum_{i \in \{1,\ldots,K,\ell\}} \mathbb{E}_\nu[n_i(T)]\mathrm{KL}(\nu_i, \nu_i') \geq \mathrm{kl}(\mathbb{E}_\nu[Z], \mathbb{E}_{\nu'}[Z]) \tag{16}$$

where kl denotes the Kullback-Leibler divergence for Bernoulli distributions, i.e.,

$$\forall p, q \in [0, 1]^2, \qquad \mathrm{kl}(p, q) = p \log \frac{p}{q} + (1 - p) \log \frac{1-p}{1-q},$$

and where $Z$ is any random variable with values in $[0, 1]$ (that is measurable with respect to the probability space that all reward random variables are defined on).

We will use (16) with $Z = \frac{n_i(T)}{T}$ for some arm $i$, and will also rely on the bound

$$\mathrm{kl}(p, q) = p \log \frac{p}{q} + (1 - p) \log \frac{1-p}{1-q} \geq (1-p) \log \frac{1}{1-q} - \log 2 \tag{17}$$

**Theorem E.1** (Lower bound for the number of pulls of *low-cost* arms). *For any bandit instance $\nu$ over horizon $T$, for all consistent strategies, we have*

$$\liminf_{T \to \infty} \frac{\mathbb{E}_\nu\left[n_i(T)\right]}{\log T} \geq \frac{1}{D_{inf}(\nu_i, (1-\alpha)\mu_\ell)}, \qquad for\ arms\ \ i = 1, \ldots, a^* - 1.$$

*Proof.* Given any bandit instance $\nu$ and any arm $i = 1, \ldots, a^* - 1$, we consider a modified instance $\nu'$ where $i$ is the unique optimal action: $\nu_j' = \nu_j$ for all $j \in \{1, \ldots, K, \ell\} - \{i\}$ and $\nu_i'$ is any distribution in $\mathcal{M}$ such that its expectation $\mu_i'$ satisfies $\mu_i' \geq (1-\alpha)\mu_\ell$. The key idea here is that any consistent policy should have $\mathbb{E}_\nu[n_i(T)] = o(T^\gamma)$ while $\mathbb{E}_{\nu'}[n_i(T)] = T - o(T^\gamma)$. We now apply the fundamental inequality (16) with $Z = \frac{n_i(T)}{T}$. Noting that all KL terms except that for arm $i$ is zero on the left hand side and using (17), we get

$$\mathbb{E}_\nu[n_i(T)]\mathrm{KL}(\nu_i, \nu_i') \ \geq \ \mathrm{kl}\left(\frac{\mathbb{E}_\nu[n_i(T)]}{T}, \frac{\mathbb{E}_{\nu'}[n_i(T)]}{T}\right)$$

$$\geq \left(1 - \frac{\mathbb{E}_\nu[n_i(T)]}{T}\right) \log\left(\frac{T}{T - \mathbb{E}_{\nu'}[n_i(T)]}\right) - \log 2. \qquad (18)$$

Given that arm $i \neq a^*$ in instance $\nu$ and $i = a^*$ in $\nu'$, any consistent policy must have $\mathbb{E}_\nu[n_i(T)] = o(T^\gamma)$ and $\mathbb{E}_{\nu'}[n_i(T)] = T - o(T^\gamma)$ which leads to,

$$\liminf_{T \to \infty} \frac{1}{\log T} \left(1 - \frac{\mathbb{E}_\nu[n_i(T)]}{T}\right) \log\left(\frac{T}{T - \mathbb{E}_{\nu'}[n_i(T)]}\right) \geq 1 - \gamma,$$

for all $0 < \gamma \leq 1$. Thus, we get

$$\liminf_{T \to \infty} \frac{\mathbb{E}_\nu[n_i(T)]}{\log T} \geq \frac{1}{\mathrm{KL}(\nu_i, \nu_i')} \qquad (19)$$

and taking the supremum in the right-hand side over all distributions $\mu_i' \geq (1 - \alpha)\mu_\ell$, we get the bound of Theorem E.1.

**Remark E.1.** *The line of arguments used in the proof given above would not yield a similar lower bound for high-cost arms, i.e., arms $i = a^* + 1, \ldots, K$. This is because even if the reward distribution of an arm $i \in \{a^* + 1, \ldots, K\}$ is modified to make its mean exceed the feasibility threshold $(1 - \alpha)\mu_\ell$, arm $a^*$ would remain to be the optimal action and thus we would not be able to state that $\mathbb{E}_{\nu'}[n_i(T)] = T - o(T^\gamma)$. This is consistent with the upper bounds established in Lemma F.7 for our PE algorithm where the number of samples for high-cost arms was seen to be upper bounded by a $O(1)$ term. In fact, the upper bounds for the PE algorithm will be seen to match within a constant factor the lower bounds for all sub-optimal arms; see Remark E.2 below.*

**Theorem E.2** (Lower bound for the number of pulls of the reference arm $\ell$). *For any bandit instance $\nu$ over horizon $T$, for all consistent strategies, we have*

$$\liminf_{T \to \infty} \frac{\mathbb{E}[n_\ell(T)]}{\log T} \geq \max\left\{ \frac{1}{D_{inf}\left(\nu_\ell, \frac{\mu_{a^*}}{1-\alpha}\right)}, \max_{i=1,\ldots,a^*-1} \frac{1}{\tilde{D}_{inf}\left(\nu_\ell, \frac{\mu_i}{1-\alpha}\right)} \right\}. \qquad (20)$$

*Proof.* Proof of Theorem E.2 follows similar arguments to that of Theorem E.1 but requires a more subtle application of the fundamental inequality (16). We will first establish the second part of (20) (associated with modifications that make an arm $i < a^*$ optimal) followed by the first part (that arises from modifications that make an arm $i > a^*$ optimal).

Consider any bandit instance $\nu$ with arm $a^*$ being the optimal action. If $a^* = \ell$, then any consistent policy must choose it at least $T - o(T^\gamma)$ times and the bound given in the Theorem follows immediately. Next, we assume $a^* \neq \ell$. Consider any arm $i = 1, \ldots, a^* - 1$ and note that we must have $\mu_i < \mu_{a^*}$ for all such arms. Now, consider a modified instance $\nu'$, where $\nu_j' = \nu_j$ for all $j = 1, \ldots, K$ and $\nu_\ell'$ is any distribution in $\mathcal{M}$ such that its expectation $\mu_\ell'$ satisfies $(1 - \alpha)\mu_\ell' \leq \mu_i$. This means that arm $i$ is feasible in the modified instance $\nu'$, and given that it has lower cost than $a^*$, $a^*$ is no longer the optimal action. In fact, in $\nu'$, the optimal action is guaranteed to be among the arms $\{1, \ldots, i\}$. Let $j \in \{1, \ldots, i\}$ be the new optimal action. We will apply the fundamental inequality (16) with $Z = \frac{n_j(T)}{T}$ and note that any consistent policy should have $\mathbb{E}_\nu[n_j(T)] = o(T^\gamma)$ while $\mathbb{E}_{\nu'}[n_j(T)] = T - o(T^\gamma)$. Proceeding similarly with the proof of Theorem E.1, we get

$$\mathbb{E}_\nu[n_\ell(T)]\mathrm{KL}(\nu_\ell, \nu_\ell') \geq \mathrm{kl}\left(\frac{\mathbb{E}_\nu[n_j(T)]}{T}, \frac{\mathbb{E}_{\nu'}[n_j(T)]}{T}\right)$$

$$\geq \left(1 - \frac{\mathbb{E}_\nu[n_j(T)]}{T}\right) \log\left(\frac{T}{T - \mathbb{E}_{\nu'}[n_j(T)]}\right) - \log 2. \qquad (21)$$

Noting that $\mathbb{E}_\nu[n_j(T)] = o(T^\gamma)$ and $\mathbb{E}_{\nu'}[n_j(T)] = T - o(T^\gamma)$ and using similar arguments as in the proof of Theorem E.1 we get

$$\liminf_{T \to \infty} \frac{\mathbb{E}_\nu[n_\ell(T)]}{\log T} \geq \frac{1}{\mathrm{KL}(\nu_\ell, \nu_\ell')} \qquad (22)$$

Taking the supremum in the right-hand side over all distributions $\nu'_\ell \in \mathcal{M}$ with $\mu'_\ell \leq \frac{\mu_i}{1-\alpha}$, we get

$$\liminf_{T\to\infty} \frac{\mathbb{E}_\nu\left[n_\ell(T)\right]}{\log T} \geq \frac{1}{\tilde{D}_{\inf}\left(\nu_\ell, \frac{\mu_i}{1-\alpha}\right)} \tag{23}$$

Finally, since these arguments can be applied for any arm $i = 1, \ldots, a^* - 1$, we get

$$\liminf_{T\to\infty} \frac{\mathbb{E}\left[n_\ell(T)\right]}{\log T} \geq \max_{i=1,\ldots,a^*-1} \frac{1}{\tilde{D}_{\inf}\left(\nu_\ell, \frac{\mu_i}{1-\alpha}\right)}. \tag{24}$$

To complete the proof of (20), we now consider a modified distribution that would make $a^*$ sub-optimal. Consider any bandit instance $\nu$ with arm $a^* \neq \ell$ being the optimal action. Consider a modified instance $\nu'$, where $\nu'_j = \nu_j$ for all $j = 1, \ldots, K$ and $\nu'_\ell$ is any distribution in $\mathcal{M}$ such that its expectation $\mu'_\ell$ satisfies $(1-\alpha)\mu'_\ell > \mu_{a^*}$. With this modification, the arm $a^*$ is no longer feasible and thus not optimal. The optimal action should then be in the set $\{a^* + 1, \ldots, K, \ell\}$. Let $j \in \{a^* + 1, \ldots, K, \ell\}$ be the new optimal action. We will apply the fundamental inequality (16) with $Z = \frac{n_j(T)}{T}$ and note that any consistent policy should have $\mathbb{E}_\nu[n_j(T)] = o(T^\gamma)$ (since in the original instance $a^* \neq j$ was optimal) while $\mathbb{E}_{\nu'}[n_j(T)] = T - o(T^\gamma)$. Using the same ideas as before, we then get

$$\liminf_{T\to\infty} \frac{\mathbb{E}_\nu\left[n_\ell(T)\right]}{\log T} \geq \frac{1}{\mathrm{KL}(\nu_\ell, \nu'_\ell)} \tag{25}$$

and taking the supremum in the right-hand side over all distributions $\nu'_\ell \in \mathcal{M}$ with $\mu'_\ell > \frac{\mu_{a^*}}{1-\alpha}$, we get

$$\liminf_{T\to\infty} \frac{\mathbb{E}_\nu\left[n_\ell(T)\right]}{\log T} \geq \frac{1}{D_{\inf}\left(\nu_\ell, \frac{\mu_{a^*}}{1-\alpha}\right)} \tag{26}$$

Finally, we combine 24 and 26 to complete the proof of 20.

**Remark E.2.** *When the reward distributions are Gaussian, i.e., the family $\mathcal{M}$ of distributions contain all Gaussian distributions with a common variance $\sigma^2$, the key terms $D_{inf}(\nu_i, x)$ and $\tilde{D}_{inf}(\nu_i, x)$ take the form of*

$$D_{inf}(\nu_i, x) = \tilde{D}_{inf}(\nu_i, x) = \frac{(\mu_i - x)^2}{2\sigma^2}. \tag{27}$$

*More generally, it was shown in Lattimore & Szepesvári (2020) that $D_{inf}(\nu_i, x) = O((\mu_i - x)^2)$ for most distribution families. Using this with $\sigma^2 = 1$, we obtain from Theorem E.1 and Theorem E.2 that*

$$\liminf_{T\to\infty} \frac{\mathbb{E}\left[n_i(T)\right]}{\log T} \geq \frac{2}{\Delta_{Q,i}^2}, \qquad \text{for arms} \ \ i = 1, \ldots, a^* - 1..$$

*and*

$$\liminf_{T\to\infty} \frac{\mathbb{E}\left[n_\ell(T)\right]}{\log T} \geq \max_{i\leq a^*} \frac{2(1-\alpha)^2}{\Delta_{Q,i}^2}.$$

*where $\Delta_{Q,i}$ is defined as before, i.e., $\Delta_{Q,i} = (1-\alpha)\mu_\ell - \mu_i$. These results match within a constant factor the upper bounds of our PE algorithm established in Lemmas F.2 and F.3 showing that the PE algorithm is order-wise optimal.*

## LOWER BOUNDS FOR THE SUBSIDIZED BEST REWARD SETTING

We now present lower bounds for the subsidized best reward setting. The core ideas in establishing these results are the same with those used in the *known reference arm* case. Namely, to establish a lower bound on the number of pulls needed from a given arm, we are seeking a modification in that arm's reward distribution that leads to a change in the optimal arm, i.e., arm $a^*$ being no

longer optimal. Among the modified distributions that change the optimal arm, the one that has the minimum KL-divergence with the original distribution leads to the tightest lower bound on that arm's number of pulls. We will repeatedly use this argument in the following discussion but the repetitive parts of the proofs are omitted for brevity.

Let $\mathcal{M}$ denote the *model* under consideration that contains all bandit instances $\nu$, where $\nu$ specifies the reward distributions $\nu_i$ of all arms $i = 1, \ldots, K$. Recall that the arm indices are cost ordered, i.e., $c_1 < \cdots < c_{a^*} < \cdots < c_{i^*} < \cdots < c_K$, where $a^*$ is the optimal action and $i^*$ is the arm with the largest mean reward; namely $i^* = \arg\max_{i \in \{1, \ldots, K\}} \mu_i$ and $a^* = \arg\min_{\{i : \mu_i \geq (1-\alpha)\mu^*\}} c_i$ where $\mu^* = \max_{i \in \{1, \ldots, K\}} \mu_i = \mu_{i^*}$. The definition of a consistent policy and key quantities $D_{\text{inf}}$, $\tilde{D}_{\text{inf}}$ remain as in the previous section.

LOWER BOUND ON THE NUMBER OF SAMPLES FOR ARMS $i < a^*$

**Theorem E.3** (Lower bound for the number of pulls of *low-cost* arms)**.** *For any bandit instance $\nu$ over horizon $T$, for all consistent strategies, we have*

$$\liminf_{T \to \infty} \frac{\mathbb{E}_\nu \left[ n_i(T) \right]}{\log T} \geq \frac{1}{D_{\text{inf}}(\nu_i, (1-\alpha)\mu^*)}, \qquad \text{for arms } i = 1, \ldots, a^* - 1.$$

*Proof.* For arms cheaper than the optimal action $a^*$, i.e., arms $i = 1, \ldots, a^* - 1$, our approach is to modify their reward distribution so that their mean reward is now larger than the feasibility threshold $(1-\alpha)\mu^*$. Since these arms have a lower cost than $a^*$, $a^*$ will no longer be optimal in the modified instance, paving the way to establishing a lower bound for the number of arms $i = 1, \ldots, a^* - 1$. Formally, given any bandit instance $\nu$ and any arm $i = 1, \ldots, a^* - 1$, we consider a modified instance $\nu'$ where $i$ is the unique best action: $\nu'_j = \nu_j$ for all $j \in \{1, \ldots, K\} - \{i\}$ and $\nu'_i$ is any distribution in $\mathcal{M}$ such that its expectation $\mu'_i$ satisfies $\mu'_i > (1-\alpha)\mu_{i^*}$. In the modified instance $\nu'$, the feasibility threshold is either $(1-\alpha)\mu_{i^*}$ as before, or it changes to $(1-\alpha)\mu'_i$ by virtue of arm $i$ being the arm with the largest mean reward. In either case, $\mu'_i$ exceeds this threshold and since $c_i < c_{a^*}$, arm $a^*$ becomes a sub-optimal action. This proves that

$$\liminf_{T \to \infty} \frac{\mathbb{E}_\nu \left[ n_i(T) \right]}{\log T} \geq \frac{1}{\text{KL}(\nu_i, \nu'_i)} \tag{28}$$

and taking the supremum in the right-hand side over all distributions $\nu'_i \in \mathcal{M}$ with $\mu'_i > (1-\alpha)\mu^*$, we get the bound of Theorem E.3. $\qquad \blacksquare$

**Remark E.3.** *For arms $i = 1, \ldots, a^* - 1$, recall that the quality gaps are defined as $\Delta_{Q,i} = (1-\alpha)\mu^* - \mu_i$. Recalling Remark E.2, we see from Theorem E.3 that when the reward distributions are Gaussian with $\sigma^2 = 1$*

$$\liminf_{T \to \infty} \frac{\mathbb{E} \left[ n_i(T) \right]}{\log T} \geq \frac{2}{\Delta_{Q,i}^2}, \qquad \text{for arms } i = 1, \ldots, a^* - 1.$$

*matching the upper bound of our PE-CS algorithm given in Lemma G.3.*

LOWER BOUND ON THE NUMBER OF SAMPLES FOR ARMS $i > a^*, i \neq i^*$

**Theorem E.4** (Lower bound for the number of pulls of *high-cost* arms)**.** *For any bandit instance $\nu$ over horizon $T$, for all consistent strategies, we have*

$$\liminf_{T \to \infty} \frac{\mathbb{E}_\nu \left[ n_i(T) \right]}{\log T} \geq \frac{1}{D_{\text{inf}}\left(\nu_i, \frac{\mu_{a^*}}{(1-\alpha)}\right)}, \qquad \text{for arms } i \in \{a^* + 1, \ldots, K\} - \{i^*\}.$$

*Proof.* For arms more expensive than $a^*$ but with mean reward less than $\mu^*$, the only modification we can make in their reward distribution to render $a^*$ sub-optimal would be to increase their mean reward to a level where they will be the highest-mean arm and $(1-\alpha)$ times their new mean exceeds $\mu_{a^*}$ (so that $a^*$ is no longer feasible and thus can not be optimal). Formally, given any bandit instance

$\nu$ and any arm $i \in \{a^* + 1, \ldots, K\} - \{i^*\}$, we consider a modified instance $\nu'$ such that $\nu'_j = \nu_j$ for all $j \in \{1, \ldots, K\} - \{i\}$ and $\nu'_i$ is any distribution in $\mathcal{M}$ such that its expectation $\mu'_i$ satisfies $(1 - \alpha)\mu'_i > \mu_{a^*}$. In the modified instance $\nu'$, we have $\mu'_i > \mu_{a^*}/(1 - \alpha) \geq \mu_{i^*}$ with the second inequality following from $\mu_{a^*} \geq (1 - \alpha)\mu_{i^*}$ since $a^*$ was feasible in the original instance. Thus, in the modified instance arm $i$ has the highest mean and the feasibility threshold becomes $(1 - \alpha)\mu'_i$ which exceeds the mean reward $\mu_{a^*}$ of arm $a^*$ by construction. Thus, in the modified instance $\nu'$, arm $a^*$ is no longer the optimal action leading to

$$\liminf_{T \to \infty} \frac{\mathbb{E}_\nu \left[ n_i(T) \right]}{\log T} \geq \frac{1}{\mathrm{KL}(\nu_i, \nu'_i)} \tag{29}$$

and taking the supremum in the right-hand side over all distributions $\nu'_i \in \mathcal{M}$ with $\mu'_i > \frac{\mu_{a^*}}{1-\alpha}$, we get the bound of Theorem E.3.

**Remark E.4.** *As before, when reward distributions are Gaussian with $\sigma^2 = 1$, Theorem E.4 gives*

$$\liminf_{T \to \infty} \frac{\mathbb{E} \left[ n_i(T) \right]}{\log T} \geq \frac{2}{\left( \frac{\mu_{a^*}}{1-\alpha} - \mu_i \right)^2}, \qquad \text{for arms } i \in \{a^* + 1, \ldots, K\} - \{i^*\}.$$

*The corresponding upper bound for the number of samples from the high-cost arms for our PE-CS algorithm, given in Lemma G.5, scales as $\frac{1}{\Delta_i^2}$. Since arm $a^*$ is feasible, we have*

$$\frac{\mu_{a^*}}{1 - \alpha} - \mu_i \; > \; \mu^* - \mu_i = \Delta_i.$$

*This suggests that there is potential for improvement in the PE-CS algorithm regarding the number of samples drawn from high-cost arms. For instance, if the two arms with the highest mean rewards have a small gap, it may not be critical to determine which one is superior (making $\Delta_i$ less relevant), provided that there exists a cheaper arm whose mean reward significantly exceeds the feasibility threshold corresponding to either of the two arms in contention to be the one with the largest mean reward.*

LOWER BOUND ON THE NUMBER OF SAMPLES FOR ARM $i^*$

**Theorem E.5** (Lower bound on the expected number of samples of the best arm)**.** *For any bandit instance $\nu$ over horizon $T$, for all consistent strategies, we have*

$$\liminf_{T \to \infty} \frac{\mathbb{E}_\nu \left[ n_{i^*}(T) \right]}{\log T} \geq \frac{1}{D_{inf}\left( \nu_{i^*}, \frac{\mu_{a^*}}{(1-\alpha)} \right)}. \tag{30}$$

*Also, with $\mu_j = \max_{i \in \{1, \ldots, a^*-1\}} \mu_i$ and $\mu_2 = \max_{i \in \{1, \ldots, K\} - i^*} \mu_i$, we have*

$$\liminf_{T \to \infty} \frac{\mathbb{E}_\nu \left[ n_{i^*}(T) \right]}{\log T} \geq \frac{1}{\tilde{D}_{inf}\left( \nu_{i^*}, \frac{\mu_j}{(1-\alpha)} \right)} \qquad \text{if} \qquad \mu_j \geq (1-\alpha)\mu_2. \tag{31}$$

*Proof.* For the arm $i^*$ that has the largest mean reward, establishing a lower bound is more subtle since we need to consider modifications in two different manners. First, we shall consider modifications to its reward distribution to make its mean larger such that arm $a^*$ is no longer feasible and thus not optimal. Formally, given any bandit instance $\nu$, we consider a modified instance $\nu'$ such that $\nu'_i = \nu_i$ for all $i \in \{1, \ldots, K\} - \{i^*\}$ and $\nu'_{i^*}$ is any distribution in $\mathcal{M}$ such that its expectation $\mu'_{i^*}$ satisfies $(1 - \alpha)\mu'_{i^*} > \mu_{a^*}$. In the modified instance $\nu'$, arm $i^*$ is still the one with the largest mean reward and the feasibility threshold has changed to $(1 - \alpha)\mu'_{i^*}$. Since $\mu_{a^*} < (1 - \alpha)\mu'_{i^*}$ by construction, arm $a^*$ is no longer feasible and thus can not be optimal. This leads to

$$\liminf_{T \to \infty} \frac{\mathbb{E}_\nu \left[ n_{i^*}(T) \right]}{\log T} \geq \frac{1}{\mathrm{KL}(\nu_{i^*}, \nu'_{i^*})} \tag{32}$$

and taking the supremum in the right-hand side over all distributions $\nu'_{i^*} \in \mathcal{M}$ with $\mu'_{i^*} > \frac{\mu_{a^*}}{1-\alpha}$, we get the first bound 30 of Theorem E.5.

Next, we consider ways of reducing the mean reward of arm $i^*$ that would render $a^*$ sub-optimal by making one of the cheaper arms $i = 1, \ldots, a^* - 1$ feasible. Here, we shall notice that by reducing the mean reward of arm $i^*$, we can reduce the feasibility threshold to no lower than $(1-\alpha)\mu_2$ where $\mu_2$ is the second largest mean reward among all arms, i.e., $\mu_2 = \max_{i \in \{1, \ldots, K\} - \{i^*\}} \mu_i$. Even then, this modification would make $a^*$ sub-optimal *only if* one of the arms $i = 1, \ldots, a^* - 1$ (that all have smaller cost than $a^*$) has a mean reward that would make it feasible at this threshold. Namely, with $\mu_j = \max_{i \in \{1, \ldots, a^*-1\}} \mu_i$ (i.e., $\mu_j$ is the largest mean reward among arms $1, \ldots, a^* - 1$, and second largest among arms $1, \ldots, a^*$), this would be possible only if $\mu_j \geq (1-\alpha)\mu_2$. If $\mu_j < (1-\alpha)\mu_2$, then no matter how much the mean reward of arm $i^*$ is reduced, the arms $i = 1, \ldots, a^* - 1$ would not be feasible (with feasibility threshold being at least $(1-\alpha)\mu_2$), and arm $a^*$ would remain optimal. In such cases, the only lower bound we can establish for the number of samples of arm $i^*$ is that given by (30).

Now, consider any bandit instance $\nu$ for which $\mu_j \geq (1-\alpha)\mu_2$. Then, consider a modified instance $\nu'$ such that $\nu_i' = \nu_i$ for all $i \in \{1, \ldots, K\} - \{i^*\}$ and $\nu_{i^*}'$ is any distribution in $\mathcal{M}$ such that its expectation $\mu_{i^*}'$ satisfies $(1-\alpha)\mu_{i^*}' \leq \mu_j$. Since arm $j$ was not feasible in the original instance $\nu$, we must have $(1-\alpha)\mu_{i^*} > \mu_j$, so $\mu_{i^*}' < \mu_{i^*}$. In the modified instance $\nu'$, the arm with the largest mean reward is either still $i^*$ or it becomes the arm with mean $\mu_2$. Thus, the feasibility threshold changes to $(1-\alpha)\max\{\mu_{i^*}', \mu_2\}$. By construction $\mu_j \geq (1-\alpha)\mu_{i^*}'$ and due to our initial assumption on the instance $\nu$ we have $\mu_j \geq (1-\alpha)\mu_2$ (which holds in the instance $\nu'$ since the distribution of all arms but $i^*$ remain the same). Thus, we get $\mu_j \geq (1-\alpha)\max\{\mu_{i^*}', \mu_2\}$, meaning that the arm with mean reward $\mu_j$ is feasible in instance $\nu'$. Since this arm is in $\{1, \ldots, a^* - 1\}$, its cost is less than the cost of arm $a^*$ meaning that arm $a^*$ is not optimal in $\nu'$; instead the arm with mean reward $\mu_j$ is optimal. This leads to

$$\liminf_{T \to \infty} \frac{\mathbb{E}_\nu[n_{i^*}(T)]}{\log T} \geq \frac{1}{\mathrm{KL}(\nu_{i^*}, \nu_{i^*}')} \tag{33}$$

and taking the supremum in the right-hand side over all distributions $\nu_{i^*}' \in \mathcal{M}$ with $\mu_{i^*}' \leq \frac{\mu_j}{1-\alpha}$, we get the second bound 31 of Theorem E.5.

**Remark E.5.** *As before, when reward distributions are Gaussian, Theorem E.5 gives us*

$$\liminf_{T \to \infty} \frac{\mathbb{E}[n_{i^*}(T)]}{\log T} \geq \max\left\{ \frac{2(1-\alpha)^2}{((1-\alpha)\mu_{i^*} - \mu_{a^*})^2}, \mathbf{1}[\mu_j \geq (1-\alpha)\mu_2]\frac{2(1-\alpha)^2}{((1-\alpha)\mu_{i^*} - \mu_j)^2} \right\}$$

$$= 2(1-\alpha)^2 \max\left\{ \frac{1}{\Delta_{Q,a^*}^2}, \max_{i<a^*} \frac{1}{\Delta_{Q,i}^2} \mathbf{1}[\mu_j \geq (1-\alpha)\mu_2] \right\}$$

$$= 2(1-\alpha)^2 \max\left\{ \frac{1}{\Delta_{Q,a^*}^2}, \max_{i<a^*} \frac{1}{\Delta_{Q,i}^2} \mathbf{1}[\min_{i<a^*} \Delta_{Q,i} \leq (1-\alpha)\Delta_{min}] \right\}$$

*where $\Delta_{Q,i} = (1-\alpha)\mu^* - \mu_i$ is the quality gap of arm $i$, $\Delta_{min} = \mu^* - \mu_2$ is the smallest of the regular gaps, and the last equality in the indicator function follows from the fact that*

$$\min_{i<a^*} \Delta_{Q,i} = (1-\alpha)\mu^* - \mu_j = (1-\alpha)\Delta_{min} + (1-\alpha)\mu_2 - \mu_j$$

**Remark E.6.** *The lower bound established in Theorem E.5, particularly the need to consider the case $\mu_j \geq (1-\alpha)\mu_2$ separately can be explained as follows. For a policy to determine that $a^*$ is the optimal action, it needs to be able to decide that arms $1, \ldots, a^* - 1$ are not feasible but arm $a^*$ is. For the latter, the gap between $\mu_{a^*}$ and the feasibility threshold $(1-\alpha)\mu_{i^*}$ is important and affects the number of samples needed from the arm $i^*$, and this is seen in the bound (30). For the former, i.e., to decide that arms $1, \ldots, a^* - 1$ are not feasible, a policy may not need any samples from arm $i^*$ if it can determine their non-feasibility through another arm. For example, if the mean reward of arms $1, \ldots, a^* - 1$ are all less than $(1-\alpha)\mu_2$, then they can be determined to be not feasible based on the samples from the arm with mean reward $\mu_2$ (i.e., the arm with the second largest mean reward among all). This is what gives rise to (30) being the only lower bound on the samples from arm $i^*$ when $\mu_j < (1-\alpha)\mu_2$. On the other hand, when $\mu_j \geq (1-\alpha)\mu_2$, then at least one arm in $1, \ldots, a^* - 1$ can not be deemed non-feasible by comparing it with the arm with mean reward $\mu_2$ and samples from $i^*$ would be necessary. The number of samples needed from $i^*$ would be tied to the gaps between the mean reward of arms $1, \ldots, a^* - 1$ and the feasibility threshold $(1-\alpha)\mu^*$ and the smallest of these gaps is given by $(1-\alpha)\mu_{i^*} - \mu_j$ which is consistent with the bound (31).*

**Remark E.7.** *We now compare the lower bound established in Theorem E.5 with the upper bound for our PE-CS algorithm given in Lemma G.4. We can see that if $\mu_j \geq (1 - \alpha)\mu_2$, the dominant gap term in the established lower bound will be given by $\min_{i \leq a^*} |\Delta_{Q,i}|$, which in this case is also smaller than $\Delta_{min} = \mu^* - \mu_2$. Thus, the upper bound of the PE-CS algorithm given in Lemma G.4 matches the lower bound in this case within a constant factor. When $\mu_j < (1 - \alpha)\mu_2$, as also explained in the previous remark, the lower bound on the number of samples from $i^*$ is only governed by $\Delta_{Q,a^*}$, while the PE-CS upper bound depends additionally on $\min_{i < a^*} \Delta_{Q,i}$ and $\Delta_{min}$.*

## LOWER BOUNDS FOR THE FIXED THRESHOLD SETTING

Finally, we provide lower bounds for the fixed threshold setting where $\mu_{CS} = \mu_0$ and $\mu_0 > 0, \mu_0 \in \mathbb{R}$ is a known threshold. The result is stated in Theorem E.1 and the definitions for various terms remain the same as those at the beginning of Appendix E. The key difference is that the optimal arm $a^*$ is now $a^* = \arg\min_{i:\mu_i \geq \mu_0} c_i$. Further, in the fixed threshold setting, there are only low-cost arms and high-cost arms and no special reference arm.

**Theorem E.6** (Lower bound for the number of pulls of *low-cost* arms). *For any bandit instance $\nu$ over horizon $T$, for all consistent strategies, we have*

$$\liminf_{T \to \infty} \frac{\mathbb{E}[n_i(T)]}{\log T} \geq \frac{1}{D_{inf}(\nu_i, \mu_0)}, \quad \text{for arms } i = 1, \ldots, a^* - 1.$$

*Proof.* The proof follows along similar lines to the proof of Theorem E.1. Given any bandit instance $\nu$ and any arm $i = 1, \ldots, a^* - 1$, we construct a modified instance $\nu'$. The instance $\nu'$ satisfies $\nu'_j = \nu_j, \forall j \in [K] \setminus \{i\}$. Since $i < a^*$ is the optimal action, for instance $\nu'$, for the reward distribution $\nu'_i$ of arm $i$ we must have that $\nu'_i$ be drawn from set $\mathcal{M}$ and the expected reward $\mu'_i \geq \mu_0$ so in the new environment the optimal arm changes. As in the proof of Theorem E.1 we shall leverage that any consistent policy must simultaneously satisfy $\mathbb{E}_\nu[n_i(T)] = o(T^\gamma)$ and $\mathbb{E}_{\nu'}[n_i(T)] = T - o(T^\gamma) \, \forall \gamma \in (0, 1]$. Selecting $Z = \frac{n_i(T)}{T}$ and following the steps in the proof of Theorem E.1 precisely we get,

$$\liminf_{T \to \infty} \frac{\mathbb{E}_\nu[n_i(T)]}{\log T} \geq \frac{1}{\text{KL}(\nu_i, \nu'_i)}$$

$$\geq \frac{1}{D_{\inf}(\nu_i, \mu_0)} \quad \text{(taking supremum over all permissible } \nu'_i).$$

**Remark E.8** (No lower bound for high cost arms). *As remarked earlier for the known reference arm setting, there is no lower bound corresponding to Theorem E.6 for high cost arms $i = a^*+1, \ldots, K$. In the modified instance $\nu'$ these arms shall not become optimal even if $\mu_i \geq \mu_0$ since they are more expensive than action $a^*$. This fact prevents developing a lower bound for their samples.*

**Remark E.9** (Lower bound in terms of gap $\Delta_{Q,i}$). *As worked out in previous remarks, when $\mathcal{M}$ is the family of Gaussian reward distributions with variance $\sigma^2 = 1$, we have $D_{inf}(\nu_i, x) = \frac{(\mu_i - x)^2}{2}$ and therefore,*

$$\liminf_{T \to \infty} \frac{\mathbb{E}[n_i(T)]}{\log T} \geq \frac{2}{(\mu_i - \mu_0)^2} = \frac{2}{\Delta_{Q,i}^2}, \quad \text{for arms } i = 1, \ldots, a^* - 1.$$

*We shall later see in Appendix H, the lower bound matches the upper bound for our FT-UCB algorithm designed for this setting.*

$\square$

# F    Analysis for PE in the Known Reference Arm Setting

In this Section we build up to the proof of Theorem 3.2 by upper bounding the expected number of samples of low-cost infeasible arms, the reference arm $\ell$, and of high-cost arms under the Asymmetric-PE setting with maximum round-deviation $\kappa$. We then particularize these results to the $\kappa = 0$ case corresponding to conventional PE to obtain an upper bound on the expected regret for Algorithm 1 PE.

## F.1    Definitions and Setup Required for Analysis

As discussed in the main paper, the PE algorithm is inspired by the Improved-UCB successive elimination approach where sampling of arms occurs in un-interrupted batches called rounds. In Improved-UCB, a set of active arms is maintained and at the end of every round, arms in the active set are re-tested for their candidacy using an elimination criteria. Since in Pairwise-Elimination, we inherit the un-interrupted round based sampling scheme and elimination-criteria first used in Improved-UCB, to prove Theorem 3.2 we build on the analysis from Auer & Ortner (2010).

**Definition F.1** (Round $\rho_i$). *For $i \leq a^*$, define round number $\rho_i$ to be,*

$$\rho_i = \min \left\{ \omega_i \mid \tilde{\Delta}_{\omega_i} < \frac{|\Delta_{Q,i}|}{2} \right\}. \tag{34}$$

Intuitively, $\rho_i$ is the PE round number during episode $i$ by which we expect to either identify low-cost infeasible arm $i < a^*$ as being unsuitable or identify the best action $a^*$ as being suitable.

**Definition F.2** (Samples associated with a round $\tau$). *From Function 4, we know that for any arm the required number of samples to be drawn by round $\omega_i$ is given by,*

$$\tau_{\omega_i} = \left\lceil \frac{2 \log \left( T \tilde{\Delta}_{\omega_i}^2 \right)}{\tilde{\Delta}_{\omega_i}^2} \right\rceil. \tag{35}$$

Table 1: Probabilistic Events Descriptions and Symbols for PE Analysis

| Symbol | Event Description |
|---|---|
| $G_{1,i}, \forall i \leq a^*$ | Episode $i$ is executed to evaluate arm $i$ |
| $G_{2,i}, \forall i < a^*$ | Arm $i$ is eliminated by arm $\ell$ by when round $\omega_i = \rho_i$, during episode $i$ |
| $G_{2,a^*}$ | Arm $\ell$ is eliminated by arm $a^*$ by when $\omega_{a^*} = \rho_{a^*}$, during episode $a^*$ |
| $G_{3,i}, \forall i < a^*$ | Arm $\ell$ is not eliminated by arm $i$ by when round $\omega_i = \rho_i - 1$, during episode $i$ |
| $G_{3,a^*}$ | Arm $a^*$ is not eliminated by arm $\ell$ by when round $\omega_{a^*} = \rho_{a^*} - 1$, during episode $a^*$ |
| $E_i, \forall i \leq a^*$ | Available samples ran out during episode $i$ before the sampling for round $\rho_i$ could conclude and before an arm could be eliminated |

**Definition F.3** (Samples between a time interval $n_i(t_1, t_2)$). *For the forthcoming analysis we introduce notation $n_i(t_1, t_2)$ for the random variable denoting the number of samples of arm $i$ accrued between time steps $t_1$ and $t_2$ both inclusive.*

**Definition F.4** (Final time-step in episode $i$). *We use $t_i$ to denote the final time-step in episode $i$.*

Combining definitions F.3 and F.4 the variable $n_\ell(1, t_i)$ denotes the number of samples of reference arm $\ell$ accrued by the end of episode $i$. At the outset of our analysis, we define a large collection of probabilistic events needed for developing Theorem 3.2's intermediate results in Table 1. Each event in Table 1 is a subset of the sample space $\Omega$ associated with a run of PE. In the descriptions

of events in Table 1, when we say that an elimination event occurs *by a round*, we are including the round being mentioned. For example: "Arm $i$ is eliminated by round $\rho_i$" means that the arm $i$ was eliminated in round $\omega_i$ such that $\omega_i \leq \rho_i$.

**Remark F.1.** *In the event descriptions, whenever we refer to the reference arm $\ell$ available to the algorithm, we are really referring to the action whose expected return is $(1 - \alpha)\mu_\ell$, where $\alpha$ is the subsidy factor.*

Arm $\ell$ not being eliminated by arm $i$ during round $\rho_i$ is covered by $G_{2,i}$, hence the rounds range up to $\rho_i - 1$ in the definition of $G_{3,i}$. Unlike the episodes $i < a^*$ where the nominal outcome is for the reference arm to win over the candidate arm, for episode $a^*$, nominally the arm $a^*$ shall be the winner and therefore the events $G_{2,a^*}, G_{3,a^*}$ are defined separately to account for this reality.

Lastly, we define compound event $G_i$ using events in Table 1 as,

$$G_i = G_{1,i} \cap \left( (G_{2,i} \cap G_{3,i}) \cup E_i \right). \tag{36}$$

In English the event $G_i, i \leq a^*$ is the event that episode $i$ occurs (event $G_{1,i}$) and that either an accurate and timely elimination of one arm by another is made (event $G_{2,i} \cap G_{3,i}$), or we run out of samples before a decision could be made and prior to the conclusion of round $\rho_i$ (event $E_i$). Conditioning the samples $n_i(T)$ for PE on event $G_i$ gives us the following key result,

$$\Pr\left( n_i(T) \leq \tau_{\rho_i} \mid G_i \right) = 1. \tag{37}$$

We leverage Equation 37 in conjunction with Lemma D.3 to prove Theorem 3.2. To use this procedure, we require Lemma F.1 that partitions the probability space $\Omega$ into mutually exclusive and exhaustive events including $G_i$.

**Lemma F.1** (Partition of $\Omega$ with $G_i$). *The events $G_i$, $B_{1,i} = G_{1,i}^c$, and $B_i = G_{1,i} \cap \left( G_{2,i}^c \cup G_{3,i}^c \right) \cap E_i^c$ are exhaustive and pairwise exclusive $\forall i \leq a^*$.*

*Proof.* We can prove that the events stated in Lemma F.1 are mutually exclusive and exhaustive by showing that $G_i^c = B_{1,i} \cup B_i$. Since $G_i$ and $G_i^c$ are exhaustive showing so will show that the three events are exhaustive. Moreover, since $G_i$ and $G_i^c$ are mutually exclusive, and since $B_{1,i}$ and $B_i$ are mutually exclusive by construction, we would also have all the events being pairwise mutually exclusive in addition to being exhaustive.

$$B_{1,i} \cup B_i = G_{1,i}^c \cup \left( G_{1,i} \cap \left( G_{2,i}^c \cup G_{3,i}^c \right) \cap E_i^c \right) \tag{38}$$
$$= \left( G_{1,i}^c \cup G_{1,i} \right) \cap \left( G_{1,i}^c \cup \left( \left( G_{2,i}^c \cup G_{3,i}^c \right) \cap E_i^c \right) \right) \quad (\cup \text{ distributes over } \cap) \tag{39}$$
$$= G_{1,i}^c \cup \left( \left( G_{2,i}^c \cup G_{3,i}^c \right) \cap E_i^c \right) \tag{40}$$
$$= G_i^c. \tag{41}$$

Where Equation 41 follows from the expression obtained using De Morgan's laws for $G_i^c$ using the definition of $G_i$ in Equation 36. $\qquad \square$

BOUND SAMPLES FOR THE CASE $i < a^*$

**Lemma F.2** (Bound on the number of samples of a low-cost infeasible arm under Pairwise-Elimination). *When the maximum round deviation $\kappa = 0$, the expected number of samples of a low-cost arm with index $i < a^*$ over horizon $T$ is upper bounded by,*

$$\mathbb{E}\left[ n_i(T) \right] < 1 + \frac{32 \log \left( T \Delta_{Q,i}^2 \right)}{\Delta_{Q,i}^2} + \frac{43}{\Delta_{Q,i}^2}.$$

*Proof.* The expected number of pulls $\mathbb{E}\left[ n_i(T) \right]$ are bound by using the Iterated Expectation Lemma D.3 and conditioning on the event collection $G_i, B_{1,i}, B_i$ which are mutually exclusive and exhaustive per Lemma F.1.

$$\mathbb{E}\left[ n_i(T) \right] = \mathbb{E}\left[ n_i(T) \mid G_i \right] \cdot \Pr(G_i) + \mathbb{E}\left[ n_i(T) \mid B_{1,i} \right] \cdot \Pr(B_{1,i})$$
$$+ \mathbb{E}\left[ n_i(T) \mid B_i \right] \cdot \Pr(B_i) \tag{42}$$

$$\leq \mathbb{E}\left[n_i\left(T\right) \mid G_i\right] + T \cdot \Pr\left(B_i\right) \tag{43}$$

$$= \mathbb{E}\left[n_i\left(T\right) \mid G_i\right] + T \cdot \Pr\left(G_{1,i} \cap \left(G_{2,i}^c \cup G_{3,i}^c\right) \cap E_i^c\right) \quad (B_i \text{ from Lemma F.1}) \tag{44}$$

$$\leq \mathbb{E}\left[n_i\left(T\right) \mid G_i\right] + T \cdot \Pr\left(G_{1,i} \cap \left(G_{2,i}^c \cup G_{3,i}^c\right)\right) \tag{45}$$

$$= \mathbb{E}\left[n_i\left(T\right) \mid G_i\right] + T \cdot \Pr\left(\left(G_{1,i} \cap G_{2,i}^c\right) \cup \left(G_{1,i} \cap G_{3,i}^c\right)\right) \quad (\text{distributivity of } \cap) \tag{46}$$

$$\leq \mathbb{E}\left[n_i\left(T\right) \mid G_i\right] + T \cdot \Pr\left(B_{2,i} \cup B_{3,i}\right) \quad (\text{simplifying notation}) \tag{47}$$

$$= \mathbb{E}\left[n_i\left(T\right) \mid G_i\right] + T \cdot \left(\Pr\left(B_{2,i}\right) + \Pr\left(B_{3,i}\right)\right) \quad (\text{using the union bound}). \tag{48}$$

Where Equation 43 follows from the fact that $n_i\left(T\right) \mid B_{1,i} = 0$ since there can be no samples of arm $i$ if episode $i$ never occurs, and we define $G_{1,i} \cap G_{2,i}^c$ and $G_{1,i} \cap G_{3,i}^c$ as $B_{2,i}$ and $B_{3,i}$ respectively for notational simplicity.

First we bound the number of samples of arm $i$ conditioned on the good event $G_i$. Since during episode $i < a^*$, we either make the correct decision of eliminating arm $i$ by episode $\omega_i = \rho_i$ as captured by $G_{2,i} \cap G_{3,i}$. Alternatively, under $G_i$ we run out of samples as captured by $E_i$. In either case we will not have more than $\tau_{\rho_i}$ samples of arm $i$, where $\tau_{\rho_i}$ is given by,

$$\tau_{\rho_i} = \left\lceil \frac{2\log(T\tilde{\Delta}_{\rho_i}^2)}{\tilde{\Delta}_{\rho_i}^2} \right\rceil. \tag{49}$$

By construction of the round $\rho_i$, for all $i \leq a^*$, we have,

$$\frac{|\Delta_{Q,i}|}{4} \leq \tilde{\Delta}_{\rho_i} < \frac{|\Delta_{Q,i}|}{2}. \tag{50}$$

Plugging in the bounds in Equation 50,

$$\tau_{\rho_i} \leq \left\lceil \frac{32\log\left(\frac{T\Delta_{Q,i}^2}{4}\right)}{\Delta_{Q,i}^2} \right\rceil \tag{51}$$

$$< 1 + \frac{32\log(T\Delta_{Q,i}^2)}{\Delta_{Q,i}^2}. \tag{52}$$

Therefore, we have,

$$\mathbb{E}[n_i(T) \mid G_i] \leq \tau_{\rho_i} < 1 + \frac{32\log(T\Delta_{Q,i}^2)}{\Delta_{Q,i}^2}. \tag{53}$$

Next we bound $\Pr\left(B_{2,i}\right)$ and $\Pr\left(B_{3,i}\right)$ in order. Since $B_{2,i} = G_{1,i} \cap G_{2,i}^c$, from the specification of $G_{2,i}^c$ and the fact that intersecting with $G_{1,i}$ puts us in the sub-space of $\Omega$ where episode $i$ occurs, $B_{2,i} \forall i < a^*$ is the event: "Arm $i$ is **not** eliminated by arm $\ell$ by when round $\omega_i = \rho_i$, during episode $i$". Similarly $B_{3,i} \forall i < a^*$ is the event that "Arm $\ell$ **is eliminated** by arm $i$ by when round $\omega_i = \rho_i - 1$, during episode $i$"

Along the lines of the proof composition in Auer & Ortner (2010) for the Improved UCB algorithm, we construct three clauses on the empirical returns of arms $i$ and $\ell$ in Equations 54, 55, and 56.

$$\hat{\mu}_i \leq \mu_i + \sqrt{\frac{\log\left(T\tilde{\Delta}_{\omega_i}^2\right)}{2\tau_{\omega_i}}} \tag{54}$$

$$\hat{\mu}_\ell \geq \mu_\ell - \sqrt{\frac{\log\left(T\tilde{\Delta}_{\omega_i}^2\right)}{2\tau_{\omega_i}}} \tag{55}$$

$$\hat{\mu}_\ell \geq \mu_\ell - \sqrt{\frac{\log\left(T\tilde{\Delta}_{\omega_\ell}^2\right)}{2\tau_{\omega_\ell}}}. \tag{56}$$

Clauses 54 and 55 holding when $\omega_i = \rho_i$ lead to the elimination of arm $i$ by arm $\ell$ as is shown in the work that follows,

$$\sqrt{\log\left(T\tilde{\Delta}_{\rho_i}^2\right)/2\tau_{\rho_i}} \leq \tilde{\Delta}_{\rho_i}/2 < \Delta_{Q,i}/4. \tag{57}$$

Therefore,

$$\hat{\mu}_i + \sqrt{\frac{\log\left(T\tilde{\Delta}_{\rho_i}^2\right)}{2\tau_{\rho_i}}} \leq \mu_i + 2\sqrt{\frac{\log\left(T\tilde{\Delta}_{\rho_i}^2\right)}{2\tau_{\rho_i}}} \quad \text{(From clause 54, and } \omega_i = \rho_i) \tag{58}$$

$$< \mu_i + \Delta_{Q,i} - 2\sqrt{\frac{\log\left(T\tilde{\Delta}_{\rho_i}^2\right)}{2\tau_{\rho_i}}} \quad \text{(From the ordering 57)} \tag{59}$$

$$= (1-\alpha)\mu_\ell - 2\sqrt{\frac{\log\left(T\tilde{\Delta}_{\rho_i}^2\right)}{2\tau_{\rho_i}}} \tag{60}$$

$$\leq (1-\alpha)\hat{\mu}_\ell - (1-\alpha)\sqrt{\frac{\log\left(T\tilde{\Delta}_{\rho_i}^2\right)}{2\tau_{\rho_i}}} \quad \text{(From clause 55, and } \omega_i = \rho_i) \tag{61}$$

$$\leq (1-\alpha)\hat{\mu}_\ell - (1-\alpha)\sqrt{\frac{\log\left(T\tilde{\Delta}_{\omega_\ell}^2\right)}{2\tau_{\omega_\ell}}} \quad \text{(Since } \omega_\ell \geq \omega_i). \tag{62}$$

Here, Equation 62 is the criteria for arm $i$ being eliminated by arm $\ell$ in PE. We upper bound the probability of the arm $i$ not being eliminated by union bounding the probability of the complements of the Clauses 54 and 55 using Lemma D.2 (Hoeffding's Inequality). In addition, we include the bound on the complement of Clause 56 which shall be useful later in bounding $\Pr\left(B_{3,i}\right)$.

$$\Pr\left(\hat{\mu}_i > \mu_i + \sqrt{\frac{\log\left(T\tilde{\Delta}_{\omega_i}^2\right)}{2\tau_{\omega_i}}}\right) \leq \frac{1}{T\tilde{\Delta}_{\omega_i}^2} \tag{63}$$

$$\Pr\left(\hat{\mu}_\ell < \mu_\ell - \sqrt{\frac{\log\left(T\tilde{\Delta}_{\omega_i}^2\right)}{2\tau_{\omega_i}}}\right) \leq \exp\left(-\frac{\tau_{\omega_\ell}}{\tau_{\omega_i}}\log\left(T\tilde{\Delta}_{\omega_i}^2\right)\right) \leq \frac{1}{T\tilde{\Delta}_{\omega_i}^2} \quad \text{(Since } \tau_{\omega_\ell} \geq \tau_{\omega_i}) \tag{64}$$

$$\Pr\left(\hat{\mu}_\ell < \mu_\ell - \sqrt{\frac{\log\left(T\tilde{\Delta}_{\omega_\ell}^2\right)}{2\tau_{\omega_\ell}}}\right) \leq \frac{1}{T\tilde{\Delta}_{\omega_\ell}^2} \leq \frac{4^\kappa}{T\tilde{\Delta}_{\omega_i}^2} \quad \text{(Since } \omega_\ell \leq \omega_i + \kappa, \text{ and } \tilde{\Delta}_m = 2^{-m}). \tag{65}$$

If either of the two clauses 54 or 55 are violated, then elimination of arm $i$ will not occur. Therefore, we can bound,

$$\Pr\left(B_{2,i}\right) \leq \Pr\left(\hat{\mu}_i > \mu_i + \sqrt{\frac{\log\left(T\tilde{\Delta}_{\omega_i}^2\right)}{2\tau_{\omega_i}}}\right) + \Pr\left(\hat{\mu}_\ell < \mu_\ell - \sqrt{\frac{\log\left(T\tilde{\Delta}_{\omega_i}^2\right)}{2\tau_{\omega_i}}}\right) \tag{66}$$

$$\leq \frac{2}{T\tilde{\Delta}_{\omega_i}^2}. \tag{67}$$

Plugging in round number $\omega_i = \rho_i$ in Equation 67, and then plugging in the lower bound on $\tilde{\Delta}_{\rho_i}$ from Ordering 50, we have,

$$\Pr\left(B_{2,i}\right) \leq \frac{2}{T\tilde{\Delta}_{\rho_i}^2} \leq \frac{32}{T\Delta_{Q,i}^2}. \tag{68}$$

Finally, we wish to bound $\Pr\left(B_{3,i}\right)$. Say that the actual elimination of arm $\ell$ by arm $i$ occurs in some round $\omega_i = \rho < \rho_i$. To bound the probability of this clause of the event $G_i^c$, we note that the

clauses in Equations 54 and 56 holding simultaneously preclude arm $\ell$ from being removed by arm $i$ regardless of the round number $\rho$ in question. Therefore, using the results in Equations 63 and 65, the probability of a round $\rho$, where $\ell$ is removed, existing, can be found by plugging in $\omega_i = \rho$, and is upper bounded by $\frac{4^\kappa + 1}{T\tilde{\Delta}_\rho^2}$ [3]. While there is no definitive round number associated with $\rho$, from the clause itself we know that we must have $\rho < \rho_i$.

$$\Pr(B_{3,i}) \leq \sum_{\rho=0}^{\rho_i - 1} \frac{4^\kappa + 1}{T\tilde{\Delta}_\rho^2} = \sum_{\rho=0}^{\rho_i - 1} \frac{(4^\kappa + 1) \cdot 4^\rho}{T} \text{ (Using } \tilde{\Delta}_m = 2^{-m}) \tag{69}$$

$$< \frac{4^\kappa + 1}{3T} \cdot 4^{\rho_i} \text{ (Using the formula for the sum of a Geometric Series)} \tag{70}$$

$$= \frac{4^\kappa + 1}{3T\tilde{\Delta}_{\rho_i}^2} \text{ (Since } \tilde{\Delta}_m = 2^{-m}) \tag{71}$$

$$= \frac{16(4^\kappa + 1)}{3T\Delta_{Q,i}^2} \text{ (Because } \tilde{\Delta}_{\rho_i} \geq \frac{\Delta_{Q,i}}{4}) \tag{72}$$

$$< \frac{11}{T\Delta_{Q,i}^2} \text{ (Since we impose in Lemma F.2).} \tag{73}$$

Plugging in the bounds in Expressions 53, 68 and 73 into Equation 48 we get the overall bound on the number of samples stated in Lemma F.2. $\qquad\square$

BOUNDING SAMPLES OF REFERENCE ARM $\ell$

Let the random variable $Z$ denote the final episode in the run of PE. Then $\{Z = z\}$ constitutes a measurable event in the sample space $\Omega$ indicating that the terminal PE episode was $z$. This construction lets us establish F.3.

**Lemma F.3** (Bound on Samples of Reference arm $\ell$ under Pairwise-Elimination). *Samples of reference arm $\ell$ emanate from the episodes of candidate arms being compared to arm $\ell$. When $\kappa = 0$, we show the bound,*

$$\mathbb{E}[n_\ell(T)] < 1 + \max_{i \leq a^*} \frac{32 \log(T\Delta_{Q,i}^2)}{\Delta_{Q,i}^2} + \sum_{i=1}^{a^*} \frac{43}{\Delta_{Q,i}^2} + \mathbb{E}[n_\ell(T) \mid \{Z > a^*\}]\Pr(Z > a^*).$$

*Where $Z$ is the terminal episode of PE.*

*Proof.* Under Pairwise-Elimination the nominal outcome is for episodes $i = 1, \ldots, a^* - 1$ to result in the candidate arm $i$ being eliminated by arm $\ell$, followed arm $a^*$ eliminating arm $\ell$ during episode $a^*$. To prove Lemma F.3 we condition on this nominal sequence and upper bound the probability of the outcome deviating from this sequence by a factor proportional to $\frac{1}{T}$. Throughout the episodes $i = 1, \ldots, a^*$ the number of samples $n_\ell(T)$ are equal to the number of samples of the most sampled candidate arm $i \leq a^*$. This is because in PE we re-use samples of arm $\ell$ across episodes and only further sample $\ell$ to keep up with the samples of a candidate arm. Motivated by this fact about $n_\ell$ we begin by defining a compound high-probability good event $G$.

**Definition F.5** (Event $G$). *The compound good event $G$ is the event that for each episode $i \leq a^*$ that was executed, the episode satisfied the episode-wise good event $G_i$. Mathematically we define,*

$$G = \bigcup_{z=1}^{a^*} \left( \{Z = z\} \cap \bigcap_{i=1}^{z} G_i \right). \tag{74}$$

The complement of $G$, namely $G^c$ can be written out using the definition of $G$ and De Morgan's laws as,

$$G^c = \bigcap_{z=1}^{a^*} \left( \{Z = z\}^c \cup \bigcup_{i=1}^{z} G_i^c \right) \tag{75}$$

---

[3]Because for any events $A$, $B$, and $C$, $\Pr(A \cap B) \leq \Pr(C^c) \implies \Pr(C) \leq \Pr(A^c \cup B^c)$.

$$\subseteq \bigcup_{i=1}^{a^*} G_i^c \cup \{Z \neq a^*\} \quad \text{(picking } z = a^* \text{ from the iterated intersection)} \tag{76}$$

$$= \bigcup_{i=1}^{a^*} G_i^c \cup \{Z < a^*\} \cup \{Z > a^*\} \tag{77}$$

$$= \bigcup_{i=1}^{a^*} (B_{1,i} \cup B_i) \cup \bigcup_{i=1}^{a^*} B_{1,i} \cup \{Z > a^*\} \tag{78}$$

$$= \bigcup_{i=1}^{a^*} (B_{1,i} \cup B_i) \cup \{Z > a^*\} \tag{79}$$

$$= \bigcup_{i=1}^{a^*} G_i^c \cup \{Z > a^*\}. \tag{80}$$

Where the first term in Equation 78 follows from the equivalence between $G_i^c$ and $B_{1,i} \cup B_i$ shown in the proof of Lemma F.1. The second term in Equation 78 is based on the equivalence between the event $\{Z < a^*\}$, meaning that the final episode precedes $a^*$, and the event $\bigcup_{i=1}^{a^*} B_{1,i}$ which means that some episode $i = 1, \ldots, a^*$ was not executed.

We leverage event $G$ to bound the expected number of samples of arm $\ell$. In particular we consider three events $G, G^c \cap \{Z \leq a^*\}$, and $G^c \cap \{Z > a^*\}$ that are mutually exclusive and exhaustive by construction. We decompose $\mathbb{E}[n_\ell(T)]$ by conditioning on this collection.

$$\begin{aligned}
&\mathbb{E}[n_\ell(T)] \\
&= \mathbb{E}[n_\ell(T) \mid G] \Pr(G) + \mathbb{E}[n_\ell(T) \mid G^c \cap \{Z \leq a^*\}] \Pr(G^c \cap \{Z \leq a^*\}) \\
&\quad + \mathbb{E}[n_\ell(T) \mid G^c \cap \{Z > a^*\}] \Pr(G^c \cap \{Z > a^*\}) \tag{81} \\
&\leq \mathbb{E}[n_\ell(T) \mid G] + T \cdot \Pr(G^c \cap \{Z \leq a^*\}) + \mathbb{E}[n_\ell(T) \mid \{Z > a^*\}] \Pr(Z > a^*). \tag{82}
\end{aligned}$$

Where bound 82 follows from the simplified form of $G^c$ developed in Equation 80.

Since samples of arm $\ell$ are reused between episodes with further sampling of arm $\ell$ only occurring to match the demand from a higher round number, we have,

$$\mathbb{E}[n_\ell(T) \mid G] = \mathbb{E}\left[\left(\max_{i \leq Z} n_i(1, t_Z)\right) \mid G\right] \quad \text{(using definitions F.3 and F.4)} \tag{83}$$

$$\leq \mathbb{E}\left[\max_{i \leq a^*} n_i(1, t_{a^*}) \mid G\right] \quad (\because Z \mid G \leq a^*) \tag{84}$$

$$\leq \max_{i \leq a^*} \tau_{\rho_i} \quad \text{(using Equation 37, and by construction of } G \text{ as } \cap G_i) \tag{85}$$

$$= \max_{i \leq a^*} \left\lceil \frac{2 \log\left(T \tilde{\Delta}_{\rho_i}^2\right)}{\tilde{\Delta}_{\rho_i}^2} \right\rceil \tag{86}$$

$$< 1 + \max_{i \leq a^*} \frac{2 \log\left(T \tilde{\Delta}_{\rho_i}^2\right)}{\tilde{\Delta}_{\rho_i}^2} \tag{87}$$

$$\leq 1 + \max_{i \leq a^*} \frac{32 \log\left(T \Delta_{Q,i}^2 / 4\right)}{\Delta_{Q,i}^2} \quad \text{(Using the ordering in Relation 50)} \tag{88}$$

$$< 1 + \max_{i \leq a^*} \frac{32 \log\left(T \Delta_{Q,i}^2\right)}{\Delta_{Q,i}^2}. \tag{89}$$

To complete the bound $\mathbb{E}[n_\ell(T)]$, we develop a bound on the term $\Pr(G^c \cap \{Z \leq a^*\})$ in Lemma F.6. However, to prove Lemma F.6 we first require two intermediate results in the form of Lemmas F.4 and F.5.

**Lemma F.4.** *In all the outcomes contained in $G^c$ ending in some episode $Z = i \leq a^*$, some $B_j, j \leq i$ must have held.*

$$G^c \cap \{Z = i\} \subseteq \bigcup_{j=1}^{i} B_j \quad \forall i \leq a^*. \tag{90}$$

*Proof.*

$$G^c \cap \{Z = i\} = ((G_i^c \cup G_i) \cap \{Z = i\}) \cap G^c \tag{91}$$

$$= ((G_i^c \cap \{Z = i\}) \cup (G_i \cap \{Z = i\})) \cap G^c \quad (\cap \text{ distributes over } \cup) \tag{92}$$

$$= (G_i^c \cap \{Z = i\}) \cup (G_i \cap \{Z = i\} \cap G^c) \quad (\because G_i^c \cap \{Z = i\} \subseteq G^c) \tag{93}$$

$$= ((B_{1,i} \cup B_i) \cap \{Z = i\}) \cup (G_i \cap \{Z = i\} \cap G^c) \quad (\because G_i^c = B_{1,i} \cup B_i) \tag{94}$$

$$\subseteq B_i \cup (G_i \cap \{Z = i\} \cap G^c) \quad (\because B_{1,i} \cap \{Z = i\} = \phi). \tag{95}$$

Now consider just the event $G_i \cap \{Z = i\} \cap G^c$ from Equation 95. We can find an event it is subsumed within in the following way,

$$G_i \cap \{Z = i\} \cap G^c = G_i \cap \left( \bigcap_{j=1}^{i-1} G_j \cup \left( \bigcap_{j=1}^{i-1} G_j \right)^c \right) \cap \{Z = i\} \cap G^c \tag{96}$$

$$= \left( \bigcap_{j=1}^{i} G_j \cap \{Z = i\} \cap G^c \right) \cup \left( G_i \cap \bigcup_{j=1}^{i-1} G_j^c \cap \{Z = i\} \cap G^c \right) \tag{97}$$

$$= G_i \cap \bigcup_{j=1}^{i-1} \left( G_j^c \cap \{Z = i\} \right) \cap G^c \quad (\bigcap_{j=1}^{i} G_j \cap \{Z = i\} \subseteq G, \forall i \leq a^*) \tag{98}$$

$$= G_i \cap \bigcup_{j=1}^{i-1} \left( (B_{1,j} \cup B_j) \cap \{Z = i\} \right) \cap G^c \quad \text{(from Lemma F.1)} \tag{99}$$

$$= G_i \cap \bigcup_{j=1}^{i-1} (B_j \cap \{Z = i\}) \cap G^c \quad (\because B_{1,j} \cap \{Z = i\} = \phi, \forall j \leq i) \tag{100}$$

$$\subseteq \bigcup_{j=1}^{i-1} B_j. \tag{101}$$

Plugging in Equation 101 into Equation 95 gives us the result stated in Lemma F.4. $\qquad \square$

**Lemma F.5.** *Within the space of events that constitute $G^c$, if an episode $i \leq a^*$ does not occur, then there exists $j < i$ such that the event $B_j$ occurred. Mathematically,*

$$B_{1,i} \cap G^c \subseteq \bigcup_{j=1}^{i-1} B_j, \forall i \leq a^*. \tag{102}$$

*Proof.* We can prove the result by induction. First note that $B_{1,1} = \phi$ since Episode 1 always occurs. Let $i = 2$ represent the base case. Then,

$$B_{1,2} \cap G^c = \{Z = 1\} \cap G^c \tag{103}$$

$$\subseteq B_1 \quad \text{(Using Lemma F.4)}. \tag{104}$$

Now say that the statement in Lemma F.5 holds true for some $i = k < a^*$. That is,

$$B_{1,k} \cap G^c \subseteq \bigcup_{j=1}^{k-1} B_j. \tag{105}$$

To prove Lemma F.5 we need to show this result for $i = k + 1$.

$$B_{1,k+1} \cap G^c = (B_{1,k} \cup \{Z = k\}) \cap G^c \tag{106}$$

$$= (B_{1,k} \cap G^c) \cup (\{Z = k\} \cap G^c) \quad \cap \text{ over } \cup \tag{107}$$

$$\subseteq \bigcup_{j=1}^{k-1} B_j \cup \bigcup_{j=1}^{k} B_j \quad \text{(using the induction hypothesis and Lemma F.4).} \tag{108}$$

$$= \bigcup_{j=1}^{k} B_j. \tag{109}$$

Where Equation 106 uses the identity $B_{1,k+1} = B_{1,k} \cup \{Z = k\}$ which breaks down the event of episode $k + 1$ not occurring into the event that episode $k$ did not occur, or the event that episode $k$ was the final episode $Z$. Equation 109 is the required result from the statement of Lemma F.5. $\square$

**Lemma F.6** (Bound on $\Pr(G^c \cap \{Z \leq a^*\})$). *We can upper bound* $\Pr(G^c \cap \{Z \leq a^*\})$ *as,*

$$\Pr(G^c \cap \{Z \leq a^*\}) \leq \sum_{i=1}^{a^*} \Pr(B_j) \tag{110}$$

$$\leq \sum_{i=1}^{a^*-1} \frac{43}{T\Delta_{Q,i}^2} + \Pr(B_{a^*}) \ . \tag{111}$$

*Where $B_i, \forall i \leq a^*$ is as defined in Lemma F.1, and using the bound on $\Pr(B_i)$ shown in the proof of Lemma F.2.*

*Proof.* We begin by introducing the expanded expression for $G^c$ developed in Equation 80.

$$\Pr(G^c \cap \{Z \leq a^*\}) = \Pr(G^c \cap \{Z \leq a^*\} \cap G^c) \tag{112}$$

$$\leq \Pr\left(\left(\bigcup_{i=1}^{a^*} G_i^c \cup \{Z > a^*\}\right) \cap \{Z \leq a^*\} \cap G^c\right) \quad \text{(from Equation 80)} \tag{113}$$

$$\leq \Pr\left(\left(\bigcup_{i=1}^{a^*} G_i^c \cap G^c\right)\right) \quad (\cap \text{ distributes over } \cup) \tag{114}$$

$$\leq \Pr\left(\bigcup_{i=1}^{a^*} (G_i^c \cap G^c)\right) \quad \text{(union bound and } \Pr(A, B) \leq \Pr(A)) \tag{115}$$

$$= \Pr\left(\bigcup_{i=1}^{a^*} ((B_{1,i} \cup B_i) \cap G^c)\right) \quad \text{(by definition of } G_i) \tag{116}$$

$$= \Pr\left(\bigcup_{i=1}^{a^*} ((B_{1,i} \cap G^c) \cup (B_i \cap G^c))\right) \quad (\cap \text{ distributes over } \cup) \tag{117}$$

$$\leq \Pr\left(\bigcup_{i=1}^{a^*} \left(B_i \cup \bigcup_{j=1}^{i-1} B_j\right)\right) \quad \text{(using Lemma F.5)} \tag{118}$$

$$= \Pr\left(\bigcup_{i=1}^{a^*} B_i\right) \tag{119}$$

$$\leq \sum_{i=1}^{a^*} \Pr(B_j) \quad \text{(Union bound, and } \{Z > a^*\} \implies B_{a^*}). \tag{120}$$

$\square$

To complete the upper bound on the expected number of samples $\mathbb{E}[n_\ell(T)]$ from Equation 82, we upper bound $\Pr(B_{a^*})$ along the same lines as $\Pr(B_i), i < a^*$ in the proof of Lemma F.2.

The key difference being that the roles of candidate arm $a^*$ and reference arm $\ell$ are reversed as compared to the earlier procedure since $\mu_{a^*} \geq \mu_\ell$. Moreover, just like the earlier proof we can define $B_{2,a^*} = G_{1,a^*} \cap G_{2,i}^c$ and $B_{3,a^*} = G_{1,a^*} \cap G_{3,i}^c$. In words $B_{2,a^*}$ is the event that "Arm $\ell$ is **not** eliminated by arm $a^*$ by when $\omega_{a^*} = \rho_{a^*}$, during episode $a^*$". Similarly, $B_{3,a^*}$ is the event "Arm $a^*$ **is eliminated** by arm $\ell$ by when round $\omega_{a^*} = \rho_{a^*} - 1$, during episode $a^*$".

We bound $\Pr\left(B_{2,a^*}\right)$ and $\Pr\left(B_{3,a^*}\right)$ separately by constructing clauses on $\hat{\mu}_\ell$, and $\hat{\mu}_{a^*}$ as before.

$$\hat{\mu}_\ell \leq \mu_\ell + \sqrt{\frac{\log\left(T\tilde{\Delta}_{\omega_{a^*}}^2\right)}{2\tau_{\omega_{a^*}}}} \tag{121}$$

$$\hat{\mu}_\ell \leq \mu_\ell + \sqrt{\frac{\log\left(T\tilde{\Delta}_{\omega_\ell}^2\right)}{2\tau_{\omega_\ell}}} \tag{122}$$

$$\hat{\mu}_{a^*} \geq \mu_{a^*} - \sqrt{\frac{\log\left(T\tilde{\Delta}_{\omega_{a^*}}^2\right)}{2\tau_{\omega_{a^*}}}}. \tag{123}$$

Clauses 121 and 123 holding when $\omega_{a^*} = \rho_{a^*}$ lead to the elimination of arm $\ell$ by arm $a^*$ as is shown in the following steps,

$$\sqrt{\log\left(T\tilde{\Delta}_{\rho_{a^*}}^2\right)/2\tau_{\rho_{a^*}}} < \tilde{\Delta}_{\rho_{a^*}}/2 < |\Delta_{Q,a^*}|/4. \tag{124}$$

Therefore using $\omega_{a^*} = \rho_{a^*}$ ,

$$(1-\alpha)\left(\hat{\mu}_\ell + \sqrt{\frac{\log\left(T\tilde{\Delta}_{\omega_\ell}^2\right)}{2\tau_{\omega_\ell}}}\right) \leq (1-\alpha)\left(\hat{\mu}_\ell + \sqrt{\frac{\log\left(T\tilde{\Delta}_{\rho_{a^*}}^2\right)}{2\tau_{\rho_{a^*}}}}\right) \tag{125}$$

$$\leq (1-\alpha)\left(\mu_\ell + 2\sqrt{\frac{\log\left(T\tilde{\Delta}_{\rho_{a^*}}^2\right)}{2\tau_{\rho_{a^*}}}}\right) \quad \text{(Equation 121)} \tag{126}$$

$$\leq (1-\alpha)\mu_\ell + 2\sqrt{\frac{\log\left(T\tilde{\Delta}_{\rho_{a^*}}^2\right)}{2\tau_{\rho_{a^*}}}} \tag{127}$$

$$< (1-\alpha)\mu_\ell + |\Delta_{Q,a^*}| - 2\sqrt{\frac{\log\left(T\tilde{\Delta}_{\rho_{a^*}}^2\right)}{2\tau_{\rho_{a^*}}}} \quad \text{(From 124)} \tag{128}$$

$$= \mu_{a^*} - 2\sqrt{\frac{\log\left(T\tilde{\Delta}_{\rho_{a^*}}^2\right)}{2\tau_{\rho_{a^*}}}} \tag{129}$$

$$\leq \hat{\mu}_{a^*} - \sqrt{\frac{\log\left(T\tilde{\Delta}_{\rho_{a^*}}^2\right)}{2\tau_{\rho_{a^*}}}} \quad \text{(From Equation 123) .} \tag{130}$$

Here Equation 130 is the criteria for arm $\ell$ being eliminated by arm $a^*$ in PE. Since Clause 121 and Clause 123 being true and applicable at round $\omega_i = a^*$ imply arm $\ell$ being eliminated, we can upper bound $\Pr\left(B_{2,a^*}\right)$ as,

$$\Pr\left(B_{2,a^*}\right) \leq \Pr\left(\hat{\mu}_\ell > \mu_\ell + \sqrt{\frac{\log\left(T\tilde{\Delta}_{\omega_{a^*}}^2\right)}{2\tau_{\omega_{a^*}}}}\right) + \Pr\left(\hat{\mu}_{a^*} < \mu_{a^*} - \sqrt{\frac{\log\left(T\tilde{\Delta}_{\omega_{a^*}}^2\right)}{2\tau_{\omega_{a^*}}}}\right) \tag{131}$$

$$\leq \frac{2}{T\tilde{\Delta}^2_{\omega_{a^*}}} \quad \text{(Similar to the bounds in 63 and 64 ).} \tag{132}$$

Plugging in round number $\omega_{a^*} = \rho_{a^*}$ in Equation 132, and then plugging in the lower bound on $\tilde{\Delta}_{\rho_{a^*}}$ from Ordering 50, we have,

$$\Pr\left(B_{2,a^*}\right) \leq \frac{2}{T\tilde{\Delta}^2_{\rho_{a^*}}} \leq \frac{32}{T\Delta^2_{Q,a^*}}. \tag{133}$$

To complete the bound on $\Pr\left(B_{a^*}\right)$ we must bound $\Pr\left(B_{3,a^*}\right)$ which is the the probability of arm $a^*$ being eliminated by arm $\ell$ by round $\omega_{a^*} = \rho_{a^*}$. Similar to the arguments in the Proof of Lemma F.2, the clauses in Equations 121 and 123 holding simultaneously preclude arm $a^*$ from being removed by arm $\ell$ regardless of the round number, and we shall have,

$$\Pr\left(B_{3,a^*}\right) < \frac{16\left(4^\kappa + 1\right)}{3T\Delta^2_{Q,a^*}} \tag{134}$$

$$< \frac{11}{T\Delta^2_{Q,a^*}} \quad \text{(When we impose } \kappa = 0\text{).} \tag{135}$$

Using steps identical to those that lead up to Equation 48 we shall have,

$$\Pr\left(B_{a^*}\right) \leq \Pr\left(B_{2,a^*}\right) + \Pr\left(B_{3,a^*}\right) \tag{136}$$

$$< \frac{43}{T\Delta^2_{Q,a^*}} \quad \text{(from upper bounds in Equations 133 and 135).} \tag{137}$$

Combining the upper bound on the expected number of samples of arm $\ell$ in Equation 89, with the bound on $\Pr\left(G^c \cap \{Z \leq a^*\}\right)$ in Lemma F.6, and the bound on $\Pr\left(B_{a^*}\right)$ in Equation 137 all into 82 we reach the upper bound stated in Lemma F.3. $\qquad\square$

BOUNDING SAMPLES FOR HIGH COST ARMS $a^* < i < \ell$

**Lemma F.7** (Bound on the expected number of samples of all high cost arms under Pairwise-Elimination). *When $\kappa = 0$ the expected number of samples of all the higher cost, non-reference arms, that is, arms with index in the range $a^* < i < \ell$ is upper bounded by,*

$$\sum_{a^* < i < \ell} \mathbb{E}\left[n_i(T)\right] \leq \mathbb{E}\left[\sum_{a^* < i < \ell} n_i(T) \mid \{Z > a^*\}\right] \cdot \Pr\left(Z > a^*\right).$$

*Where $Z$ is the terminal PE episode. We shall later see that this amounts to a constant.*

*Proof.* We can upper bound the expected number of samples $\sum_{a^* < i < \ell} \mathbb{E}\left[n_i(T)\right]$ by conditioning on the event $\{Z \leq a^*\}$ and its complement.

$$\sum_{a^* < i < \ell} \mathbb{E}\left[n_i(T)\right] = \mathbb{E}\left[\sum_{a^* < i < \ell} n_i(T)\right] \quad \text{(from the linearity of Expectation operator)} \tag{138}$$

$$= \mathbb{E}\left[\sum_{a^* < i < \ell} n_i(T) \mid \{Z \leq a^*\}\right] \cdot \Pr\left(\{Z \leq a^*\}\right)$$

$$+ \mathbb{E}\left[\sum_{a^* < i < \ell} n_i(T) \mid \{Z > a^*\}\right] \cdot \Pr\left(\{Z > a^*\}\right) \quad \text{(from Lemma D.3).} \tag{139}$$

Where $\mathbb{E}\left[\sum_{i=a^*+1}^{\ell} n_i(T) \mid \{Z \leq a^*\}\right] = 0$ follows from the fact that there cannot be any samples of an arm $i > a^*$ in the case when the final episode $Z$ is less than $a^*$. $\qquad\square$

Finally, we are in a position to Prove Theorem 3.2.

*Proof of Theorem 3.2.* First, we apply Regret decomposition in Lemma D.5 to Cost Regret,

$$\mathbb{E}\left[\text{Cost\_Reg}(T,\nu)\right] \leq \sum_{i=a^*+1}^{\ell} \Delta_{C,i}^+ \mathbb{E}\left[n_i(T)\right] \quad \text{(because } \Delta_{C,i} \leq 0 \text{ for } i \leq a^*) \tag{140}$$

$$= \Delta_{C,\ell}^+ \mathbb{E}\left[n_\ell(T)\right] + \sum_{a^*<i<\ell} \Delta_{C,i}^+ \mathbb{E}\left[n_i(T)\right] \tag{141}$$

$$\leq \Delta_{C,\ell}^+ \mathbb{E}\left[n_\ell(T)\right] + \max_{a^*<i<\ell} \Delta_{C,i}^+ \sum_{a^*<i<\ell} \mathbb{E}\left[n_i(T)\right] \tag{142}$$

$$< \left(1 + \max_{i\leq a^*} \frac{32\log\left(T\Delta_{Q,i}^2\right)}{\Delta_{Q,i}^2} + \sum_{i=1}^{a^*} \frac{43}{\Delta_{Q,i}^2}\right)\Delta_{C,\ell}^+$$
$$+ \Delta_{C,\ell}^+ \mathbb{E}\left[n_\ell(T) \mid \{Z > a^*\}\right] \cdot \Pr\left(Z > a^*\right)$$
$$+ \max_{a^*<i<\ell} \Delta_{C,i}^+ \mathbb{E}\left[\sum_{a^*<i<\ell} n_i(T) \mid \{Z > a^*\}\right] \cdot \Pr\left(Z > a^*\right) \tag{143}$$

$$\leq \left(1 + \max_{i\leq a^*} \frac{32\log\left(T\Delta_{Q,i}^2\right)}{\Delta_{Q,i}^2} + \sum_{i=1}^{a^*} \frac{43}{\Delta_{Q,i}^2}\right)\Delta_{C,\ell}^+$$
$$+ \max_{a^*<i\leq\ell} \Delta_{C,i}^+ \cdot T \cdot \Pr\left(Z > a^*\right) \tag{144}$$

$$\leq \left(1 + \max_{i\leq a^*} \frac{32\log\left(T\Delta_{Q,i}^2\right)}{\Delta_{Q,i}^2} + \sum_{i=1}^{a^*} \frac{43}{\Delta_{Q,i}^2}\right)\Delta_{C,\ell}^+$$
$$+ \max_{a^*<i\leq\ell} \Delta_{C,i}^+ \cdot T \cdot \Pr\left(B_{a^*}\right) \; (\because \{Z > a^*\} \implies B_{a^*}) \tag{145}$$

$$\leq \left(1 + \max_{i\leq a^*} \frac{32\log\left(T\Delta_{Q,i}^2\right)}{\Delta_{Q,i}^2} + \sum_{i=1}^{a^*} \frac{43}{\Delta_{Q,i}^2}\right)\Delta_{C,\ell}^+ + \frac{43}{\Delta_{Q,a^*}^2} \max_{a^*<i\leq\ell} \Delta_{C,i}^+. \tag{146}$$

Where the final inequality follows from the bound in 137. Similarly for Quality Regret we have,

$$\mathbb{E}\left[\text{Quality\_Reg}(T,\nu)\right] = \sum_{i=1}^{\ell} \Delta_{Q,i}^+ \mathbb{E}\left[n_i(T)\right] \tag{147}$$

$$= \sum_{i=1}^{a^*-1} \Delta_{Q,i} \mathbb{E}\left[n_i(T)\right] + \sum_{i=a^*+1}^{\ell-1} \Delta_{Q,i}^+ \mathbb{E}\left[n_i(T)\right] \tag{148}$$

$$\leq \sum_{i=1}^{a^*-1} \Delta_{Q,i} \mathbb{E}\left[n_i(T)\right] + \max_{a^*<i<\ell} \Delta_{Q,i}^+ \sum_{i=a^*+1}^{\ell-1} \mathbb{E}\left[n_i(T)\right] \tag{149}$$

$$< \sum_{i=1}^{a^*-1} \left(\Delta_{Q,i} + \frac{32\log\left(T\Delta_{Q,i}^2\right)}{\Delta_{Q,i}} + \frac{43}{\Delta_{Q,i}}\right) + \frac{43}{\Delta_{Q,a^*}^2} \max_{a^*<i<\ell} \Delta_{Q,i}^+. \tag{150}$$

Which are the bounds stated in Theorem 3.2. $\hfill\square$

For an improved understanding of these upper bounds, we provide a description of the terms.

First for Cost Regret,

$$\underbrace{\left(1 + \max_{i\leq a^*} \frac{32\log\left(T\Delta_{Q,i}^2\right)}{\Delta_{Q,i}^2}\right)\Delta_{C,\ell}^+}_{\substack{\text{Contribution from } \ell \text{ under nominal} \\ \text{termination in PE episode } a^*}} + \underbrace{\left(\sum_{i=1}^{a^*} \frac{43}{\Delta_{Q,i}^2}\right)\Delta_{C,\ell}^+}_{\substack{\text{Contribution from } \ell \text{ under} \\ \text{mis-termination in PE episode } \leq a^*}} + \underbrace{\frac{43}{\Delta_{Q,a^*}^2} \max_{a^*<i\leq\ell} \Delta_{C,i}^+}_{\substack{\text{Contribution from episodes } >a^* \\ \text{in case of mis-termination during ep } a^*}}.$$

Next for Quality Regret,

$$\underbrace{\sum_{i=1}^{a^*-1} \left( \Delta_{Q,i} + \frac{32 \log \left( T \Delta_{Q,i}^2 \right)}{\Delta_{Q,i}} \right)}_{\substack{\text{Contribution from } i < a^* \\ \text{under nominal termination in PE episode } a^*}} + \underbrace{\sum_{i=1}^{a^*-1} \frac{43}{\Delta_{Q,i}}}_{\substack{\text{Contribution from } i < a^* \text{ under} \\ \text{mis-termination in PE episode } \le a^*}} + \underbrace{\frac{43}{\Delta_{Q,a^*}^2} \max_{a^* < i < \ell} \Delta_{Q,i}^+}_{\substack{\text{Contribution from episodes } > a^* \\ \text{in case of mis-termination during ep } a^*}} \quad .$$

## G   ANALYSIS FOR PE-CS IN THE FULL COST-SUBSIDY SETTING

We now turn towards proving Theorem 3.4 that establishes an upper bound on expected cumulative cost and quality regret for PE-CS. The PE-CS algorithm operates in the subsidized best reward setting Sinha et al. (2021). We have already shown upper bounds on cost and quality regret for Pairwise-Elimination (PE) for its operation in the known reference arm setting. The principle hurdle in generalizing the PE analysis to an analysis for PE-CS is working through the uncertainty associated with the identification of the best arm in the BAI stage of PE-CS. To perform the PE-CS analysis, not only do we need the definitions, notation, and setup from the analysis of PE, but also we require some additional constructs spelled out in the following section.

### G.1   PE-CS AND UNKNOWN REFERENCE ARM SETTING SPECIFIC DEFINITIONS

As described in Algorithm 2, PE-CS operates in two phases, a Best-Arm-Identification (BAI) phase and a Pairwise Elimination (PE) phase. As discussed in Section 3, the BAI phase of PE-CS is the Improved UCB algorithm Auer & Ortner (2010) terminated once there is a single active arm remaining. The single remaining arm is assigned to be the empirical reference arm denoted by $\ell$. As we show in the subsequent work, by the manner in which PE-CS is setup, the event that the identified reference arm gets arm $i^*$ in the line $\ell \leftarrow \mathcal{A}[0]$ (Line 6, Algorithm 2) is a high probability event. The core idea behind the analysis is to condition the expected number of samples on this desirable and likely outcome occurring during the BAI stage.

In addition to the notation defined in the main paper, we define more constructs that are specific to the analysis of PE-CS. For arm $i$, define round number $\sigma_i$ within the Best-Arm-Identification (BAI) phase of PE-CS to be,

$$\sigma_i = \min \left\{ m \,|\, \tilde{\Delta}_m < \frac{\Delta_i}{2} \right\}. \tag{151}$$

Intuitively, round $\sigma_i$ is the round number by which we expect arm $i$ to be eliminated by arm $i^*$ in the BAI stage of PE-CS.

We let the final round during which arm $i$ was sampled during the BAI stage be denoted by the random variable $\Sigma_i$. To apportion the contributions of the BAI stage and the PE stage to the total number of samples $n_i(T)$, we introduce the variable $t_{\text{BAI}}$ to denote the final time-step $t$ in the BAI stage. As a consequence of these two definitions $n_i(1, t_{\text{BAI}}) \leq \tau_{\Sigma_i}$. The highest round number reached during the BAI stage overall for any and all active arms is denoted,

$$\Sigma_f = \max_{i \neq i^*} \Sigma_i. \tag{152}$$

And to denote the set of active arms at the last time-step of the BAI stage we use,

$$\mathcal{A}_f = \mathcal{A}(t_{\text{BAI}}). \tag{153}$$

Next, we define a collection of useful events in Table 2 for the analysis that follows. These events are contingent on outcomes occurring during the BAI stage alone.

Table 2: Probabilistic Events Descriptions and Symbols for PE-CS Analysis

| Symbol | Event Description |
|---|---|
| $\Gamma_i, \forall i \neq i^*$ | Arm $i$ is eliminated by when round $m = \sigma_i$ during the BAI-stage |
| $\beta_{1,i}, \forall i \neq i^*$ | Arm $i$ is not eliminated by when round $m = \sigma_i$ and Arm $i^* \in \mathcal{A}_{\sigma_i}$ during the BAI-stage |
| $\beta_{2,i}, \forall i \neq i^*$ | Arm $i$ is not eliminated by when round $m = \sigma_i$ and Arm $i^* \notin \mathcal{A}_{\sigma_i}$ during the BAI-stage |
| $F_i, \forall i \neq i^*$ | Final BAI round $\Sigma_f < \sigma_i$ and Arm $i \in \mathcal{A}_f$ |

**Remark G.1** (The event $F_i$). *The event $F_i$ in words is the event that samples run out before the validity of the event $\Gamma_i$ could be checked and before the arm $i$ could be eliminated. Equivalent to the description in Table 2, we can write $F_i = \{\Sigma_f < \sigma_i\} \cap \{i \in \mathcal{A}_f\}$. Using De Morgan's rules, its complement is given by $F_i^c = \{\Sigma_f \geq \sigma_i\} \cup \{i \notin \mathcal{A}_f\}$. Consequently, $\Gamma_i \subseteq F_i^c$, and $\Gamma_i = \Gamma_i \cap F_i^c$. Since event $F_i$ requires that $i \in \mathcal{A}_f$, the event $\{t_{BAI} = T\} \subset F_i \; \forall \, i \neq i^*$.*

**Remark G.2** (Events are proper subsets of $\Omega$). *Similar to the events defined in Table 1 for the analysis of PE, the events in Table 2 are proper subsets of the sample space $\Omega$. From the setup inherited from Improved-UCB, $\Gamma_i$ is a desirable and likely fate for Arm $i$ during the BAI-stage.*

To analyze the outcome of the BAI stage of the PE-CS algorithm we intuit and validate a three way partition of the sample space $\Omega$ into the events $\Gamma$, $\beta$, and $F$. $\Gamma$ is the event conditioning on which ensures that $\ell = i^*$ by requiring that the events $\Gamma_i$ held for each arm $i \neq i^*$. Additionally we require that there are samples remaining at the end of the BAI stage by intersecting with $\{t_{\text{BAI}} < T\}$. This structure makes the downstream analysis of the PE-stage tractable.

$$\Gamma = \{t_{\text{BAI}} < T\} \cap \bigcap_{i \neq i^*} \Gamma_i. \tag{154}$$

Let $A$ denote the set of arms other than the maximum reward arm $i^*$. We use $\mathcal{P}(A)$ to denote the power set of $A$, that is the collection of all the possible sub-sets of $A$. Let $S \in \mathcal{P}(A)$ denote an arbitrary subset of A. We define an event $F(S) \subseteq \Omega$ parameterized by the set $S$ as,

$$F(S) = \{t_{\text{BAI}} = T\} \cap \left( \bigcap_{i \in S} F_i \cap \bigcap_{j \in A \setminus S} \left( \Gamma_j \cap F_j^c \right) \right) \tag{155}$$

Intuitively, the set $S$ consists of arms $i \in S$ for which $F_i$ held thereby making $\Gamma_i$ unverifiable. In contrast the arms $j$ contained in $j \in A \setminus S$ are those for which $\Gamma_j$ held. An inspection of the definition of $F(S)$ in Equation 155 reveals that $F(S_1) \cap F(S_2) = \phi \; \forall \, S_1, S_2 \in \mathcal{P}(A), S_1 \neq S_2$. Taking a union over all possible $F(S)$ we get the compound event $F$,

$$F = \bigcup_{S \in \mathcal{P}(A)} F(S). \tag{156}$$

Finally, the event $\beta$ is the event that $\Gamma_i$ did not hold for some arm $i \neq i^*$ despite it being verifiable ($F_i^c$ holding).

$$\beta = \bigcup_{i \neq i^*} \left( \Gamma_i^c \cap F_i^c \right). \tag{157}$$

**Lemma G.1** (A partition of $\Omega$ using $\Gamma$). *The events $\Gamma$, $F$, and $\beta$ as defined in Equations 154, 156, and 157 respectively, form a mutually exclusive and exhaustive partition of the sample space $\Omega$.*

*Proof.* First we show that the sets are mutually exclusive by showing that their pairwise intersections namely $\Gamma \cap F$, $F \cap \beta$, and $\beta \cap \Gamma$ are all $\phi$. Starting off, it is easy to see that $\Gamma \cap F = \phi$ since,

$$\Gamma \cap F \subseteq \{t_{\text{BAI}} < T\} \cap \{t_{\text{BAI}} = T\} = \phi \text{ (definitions from Equations 154 and 156)}. \tag{158}$$

Next, to show that $F \cap \beta = \phi$, it is sufficient to show that $F(S) \cap \beta = \phi$ for arbitrary $S \in \mathcal{P}(A)$.

$$F(S) \cap \beta = \{t_{\text{BAI}} = T\} \cap \left( \bigcap_{i \in S} F_i \cap \bigcap_{j \in A \setminus S} \left( \Gamma_j \cap F_j^c \right) \right) \cap \bigcup_{k \neq i^*} \left( \Gamma_k^c \cap F_k^c \right) \tag{159}$$

Since both $F_i \cap (\Gamma_i^c \cap F_i^c) = \phi$ and $(\Gamma_i \cap F_i^c) \cap (\Gamma_i^c \cap F_i^c) = \phi$, we shall have $F(S) \cap \beta = \phi$.

Lastly, for the pair $\beta \cap \Gamma$ we have,

$$\beta \cap \Gamma = \bigcup_{i \neq i^*} \left( \Gamma_i^c \cap F_i^c \right) \cap \bigcap_{j \neq i^*} \Gamma_j \quad \text{(since the indexing for $\beta$ and $\Gamma$ need not coincide)} \tag{160}$$

$$= \bigcup_{i \neq i^*} \left( \Gamma_i^c \cap F_i^c \right) \cap \bigcap_{j \neq i^*} \left( \Gamma_j \cap F_j^c \right) \quad (\text{since } \Gamma_j = F_j^c \cap \Gamma_j) \tag{161}$$

$$= \bigcup_{i \neq i^*} \bigcap_{j \neq i^*} \left( \left( \Gamma_i^c \cap F_i^c \right) \cap \left( \Gamma_j \cap F_j^c \right) \right) = \phi. \tag{162}$$

Next we show that $\Gamma \cup F \cup \beta = \Omega$, that is, the event collection considered in Lemma G.1 is exhaustive.

$$\Gamma \cup F \supset \Gamma \cup F(\phi) = \bigcap_{i \neq i^*} \left( \Gamma_i \cap F_i^c \right) \quad (\text{plugging } F(S) \text{ when } S = \phi \text{ per Equation 155}) \tag{163}$$

$$\implies \Gamma \cup F \cup \beta \supset \bigcap_{i \neq i^*} \left( \Gamma_i \cap F_i^c \right) \cup \bigcup_{j \neq i^*} \left( \Gamma_j^c \cap F_j^c \right) \tag{164}$$

$$= \bigcup_{j \neq i^*} \bigcap_{i \neq i^*} \left( \left( \Gamma_j^c \cap F_j^c \right) \cup \left( \Gamma_i \cap F_i^c \right) \right) \tag{165}$$

$$\supseteq \bigcap_{i \neq i^*} \left( \left( \Gamma_i^c \cap F_i^c \right) \cup \left( \Gamma_i \cap F_i^c \right) \right) = \bigcap_{i \neq i^*} F_i^c. \tag{166}$$

We shall now show that $F \cup \beta \supseteq \bigcup_{i \neq i^*} F_i$ which combined with Equation 166 completes the check on the exhaustive criteria. To do this we show that $F_i \subseteq F \cup \beta \ \forall i \neq i^*$.

PROOF THAT $F_i \subseteq F \cup \beta$:

To prove this result, we start with the event $F(\{i\})$,

$$F(\{i\}) = \{t_{\text{BAI}} = T\} \cap \left( F_i \cap \bigcap_{j \in A, j \neq i} \left( \Gamma_j \cap F_j^c \right) \right) \tag{167}$$

$$= F_i \cap \bigcap_{j \in A, j \neq i} \left( \Gamma_j \cap F_j^c \right) \quad (\because F_i \subset \{t_{\text{BAI}} = T\}). \tag{168}$$

Without loss of generality, let the set of remaining arms in $A$, i.e. $A \setminus \{i\} = \{p, q, \ldots\}$. The idea behind this proof is to identify sub-events $F(\cdot)$ such that iteratively taking their union with one another and with events lying in $\beta$ reveals that $F_i \subseteq F \cup \beta$. Since $\beta = \bigcup_{k \neq i^*} \left( F_k^c \cap \Gamma_k^c \right)$ we have,

$$\left( F_p^c \cap \Gamma_p^c \right) \subseteq \beta \implies F_i \cap \left( \Gamma_p^c \cap F_p^c \right) \cap \bigcap_{j \in A \setminus \{i,p\}} \left( \Gamma_j \cap F_j^c \right) \subseteq \beta \tag{169}$$

$$(\text{intersecting with sets keeps us inside } \beta)$$

$$\implies F(\{i\}) \cup F_i \cap \left( \Gamma_p^c \cap F_p^c \right) \cap \bigcap_{j \in A \setminus \{i,p\}} \left( \Gamma_j \cap F_j^c \right) \subseteq F \cup \beta \quad (\because F(\{i\}) \subseteq F) \tag{170}$$

$$\implies F_p^c \cap F_i \cap \bigcap_{j \in A \setminus \{i,p\}} \left( \Gamma_j \cap F_j^c \right) \subseteq F \cup \beta \tag{171}$$

$$(\because \left( \Gamma_p^c \cap F_p^c \right) \cup \left( \Gamma_p \cap F_p^c \right) \text{ from } \beta, F(\{i\}))$$

$$\implies F(\{i,p\}) \cup F_p^c \cap F_i \cap \bigcap_{j \in A \setminus \{i,p\}} \left( \Gamma_j \cap F_j^c \right) \subseteq F \cup \beta \tag{172}$$

$$\implies F_i \cap \bigcap_{j \in A \setminus \{i,p\}} \left( \Gamma_j \cap F_j^c \right) \subseteq F \cup \beta. \tag{173}$$

In reaching Equation 173, we have removed the dependence on Arm $p$ for the event on the left. With the next series of equations, we further remove the dependence on Arm $q$.

$$F_i \cap \bigcap_{j \in A \setminus \{i,p\}} \left( \Gamma_j \cap F_j^c \right) \subseteq F \cup \beta \tag{174}$$

$$\implies F_i \cap \left( \Gamma_q \cap F_q^c \right) \cap \bigcap_{j \in A \setminus \{i,p,q\}} \left( \Gamma_j \cap F_j^c \right) \subseteq F \cup \beta. \tag{175}$$

Similar to what we saw in the first iteration of this procedure, we have,

$$F_i \cap \left(\Gamma_q^c \cap F_q^c\right) \cap \bigcap_{j \in A \setminus \{i,p,q\}} \left(\Gamma_j \cap F_j^c\right) \subseteq \beta \tag{176}$$

$$\implies F_i \cap F_q^c \cap \bigcap_{j \in A \setminus \{i,p,q\}} \left(\Gamma_j \cap F_j^c\right) \subseteq F \cup \beta \quad \text{(combining with Equation 175).} \tag{177}$$

Similar to how we reached the result in Equation 173 in starting from $F(\{i\})$, if instead we had started with the set $F(\{i,q\})$, and then eliminated the dependence on $p$, we would have shown,

$$F_i \cap F_q \cap \bigcap_{j \in A \setminus \{i,p,q\}} \left(\Gamma_j \cap F_j^c\right) \subseteq F \cup \beta. \tag{178}$$

Combining Equations 177 and 178, we have,

$$F_i \cap \bigcap_{j \in A \setminus \{i,p,q\}} \left(\Gamma_j \cap F_j^c\right) \subseteq F \cup \beta. \tag{179}$$

Repeating this procedure iteratively, it is clear that the event on the left can be pruned down to simply $F_i$, and therefore,

$$F_i \subseteq F \cup \beta. \tag{180}$$

Since no assumptions were made on the choice of $i$, we have,

$$\bigcup_{i \neq i^*} F_i \subseteq F \cup \beta. \tag{181}$$

As mentioned earlier, we can combine $\Gamma \cup F \cup \beta \supset \bigcap_{i \neq i^*} F_i^c$ from Equation 166 and $F \cup \beta \supseteq \bigcup_{i \neq i^*} F_i$ from Equation 181 to obtain $\Gamma \cup F \cup \beta = \Omega$. □

**Corollary G.1** (Corollary to Lemma G.1). *The events $\Gamma, \{F(S)\}_{S \in \mathcal{P}(A)}, \beta$ form a mutually exclusive and exhaustive partition over the sample space $\Omega$. This result follows trivially from Lemma G.1 and the fact that $S_1 \neq S_2 \implies F(S_1) \cap F(S_2) = \phi$.*

Next we prove an upper bound on the probability of the event $\beta$ which we will need repeatedly in proving subsequent results.

**Lemma G.2** (Bound on $\Pr(\beta)$). *To bound the expected number of samples in all the cases pertinent to PE-CS we show the following bound on $\Pr(\beta)$,*

$$\Pr(\beta) < \frac{11}{T\Delta_{min}^2} + \sum_{i \neq i^*} \frac{32}{T\Delta_i^2}. \tag{182}$$

*Where $\Delta_{min} = \mu^* - \mu_2$ is the difference between the highest and second highest reward.*

*Proof.*

$$\Pr(\beta) = \Pr\left(\bigcup_{i \neq i^*} \left(\Gamma_i^c \cap F_i^c\right)\right) \tag{183}$$

$$= \Pr\left(\bigcup_{i \neq i^*} \left(\beta_{1,i} \cup \beta_{2,i}\right)\right) \quad \text{(since } \Gamma_i^c \cap F_i^c = \beta_{1,i} \cup \beta_{2,i}) \tag{184}$$

$$\leq \Pr\left(\bigcup_{i \neq i^*} \beta_{1,i}\right) + \Pr\left(\bigcup_{i \neq i^*} \beta_{2,i}\right) \quad \text{(Union Bound)} \tag{185}$$

$$\leq \sum_{i \neq i^*} \Pr(\beta_{1,i}) + \Pr\left(\bigcup_{i \neq i^*} \{\text{Arm } i^* \notin \mathcal{A}_{\sigma_i}\}\right) \quad \text{(Union Bound, Latter clause of } \beta_{2,i}) \tag{186}$$

$$= \sum_{i \neq i^*} \Pr\left(\beta_{1,i}\right) + \Pr\left(\{\text{ Arm } i^* \notin \mathcal{A}_{\sigma_{\max}} \}\right). \tag{187}$$

Where $\sigma_{\max} = \max_{i \neq i^*} \sigma_i$, and Equation 187 follows from the fact that $i^* \notin \mathcal{A}_{\sigma_1} \implies i^* \notin \mathcal{A}_{\sigma_2}$ when $\sigma_2 > \sigma_1$.

The event $\beta_{1,i}$ is the event that Arm $i$ is not eliminated by round $\sigma_i$ while Arm $i^*$ is active at the end of the sampling for round $\sigma_i$. The term $\Pr\left(\beta_{1,i}\right)$ therefore can be bounded in a manner analogous to the way the probability of low-cost infeasible arm $i$ not being eliminated by reference arm $\ell$ was bound in Equation 67. The difference being that the round number $\rho_i$ is replaced by the round $\sigma_i$, or equivalently, the gap $\Delta_{Q,i}$ is replaced by the gap $\Delta_i$. Therefore,

$$\Pr\left(\beta_{1,i}\right) \leq \frac{32}{T\Delta_i^2}. \tag{188}$$

The problem of analyzing $\Pr\left(\beta_{2,i}\right)$ is analogous to the analysis of $\Pr\left(B_{3,i}\right)$ in the proof of Lemma F.2 for the PE algorithm. By applying the Hoeffding bound (Lemma D.2) to the clauses in Equations 54 and 56 we were able to establish that the probability of the event $\{\text{Arm } \ell \text{ eliminated by infeasible arm } i \text{ after the sampling for an arbitrary round } \rho \text{ concludes}\}$ is upper bounded by $\frac{4^\kappa + 1}{T\tilde{\Delta}_\rho^2}$. Since for the BAI setting the samples of all the arms are always matched ($\kappa = 0$) here we shall have,

$$\Pr\left(\{\text{ Arm } i^* \notin \mathcal{A}_{\sigma_{\max}} \}\right) \leq \sum_{\rho=0}^{m_{\sigma_{\max}}-1} \frac{2}{T\tilde{\Delta}_\rho^2} \leq \frac{2}{T} \sum_{\rho=0}^{m_{\sigma_{\max}}-1} 4^\rho \tag{189}$$

$$< \frac{2 \cdot 4^{\sigma_{\max}}}{3T} \tag{190}$$

$$\leq \frac{2}{3T\tilde{\Delta}_{\sigma_{\max}}^2} \tag{191}$$

$$\leq \frac{32}{3T\Delta_{\min}^2} \tag{192}$$

$$< \frac{11}{T\Delta_{\min}^2}. \tag{193}$$

Combining the bounds shown in Equations 188 and 193 we obtain the overall bound stated in Lemma G.2. $\qquad \square$

We now move on to analyzing the evolution of samples in the PE stage of PE-CS. The pieces needed from the analysis of the BAI stage are the partition over $\Omega$ from Lemma G.1, the bound on $\Pr\left(\beta\right)$ shown in Lemma G.2, and the Iterated Expectation Lemma D.3. The key difference between the analysis of PE and the PE-stage in PE-CS is the possibility that the round number $\Sigma_i$ to which the samples of an arbitrary arm $i \neq i^*$ advance during the BAI-stage exceeds the round number $\rho_i$ defined in Equation 34. Our modular proof technique sequesters both the pathological (event $F$) and the unlikely (event $\beta$) outcomes of the BAI stage away from the PE stage. In our approach the $\Sigma_i > \rho_i$ case in the analysis of the PE-stage of PE-CS only surfaces for episode $a^*$. Moreover, analyzing samples accrued during episode $a^*$ is only called for when bounding the expected number of samples of the best arm $i^*$.

**Remark G.3.** *Similar to Remark F.1 we note here that Arm $i^*$ during the PE stage of PE-CS really refers to a hypothetical Bandit Arm with expected return $(1-\alpha)\mu_*$.*

Using all the definitions and constructs introduced in this section, we are now in a position to show an upper bound on the expected number of samples of low-cost arms in Lemma G.3, the maximum reward arm $i^*$ in Lemma G.4, and high-cost arms in Lemma G.5.

UPPER BOUND ON THE NUMBER OF SAMPLES FOR ARMS $i < a^*$

**Lemma G.3** (Bound on the expected number of samples of a low-cost arm under PE-CS). *For any low cost infeasible arm $i < a^*$, its expected number of samples accrued is upper bounded by,*

$$\mathbb{E}\left[n_i\left(T\right)\right] < 1 + \frac{32\log\left(T\Delta_{Q,i}^2\right)}{\Delta_{Q,i}^2} + \frac{43}{\Delta_{Q,i}^2} + \mathbb{E}\left[n_i\left(T\right)\mid\beta\right]\cdot\left(\frac{11}{T\Delta_{min}^2} + \sum_{j\neq i^*}\frac{32}{T\Delta_j^2}\right).$$

*Proof.* We begin the analysis by applying the Iterated Expectation Lemma D.3 to the partition developed in Lemma G.1.

$$\mathbb{E}\left[n_i\left(T\right)\right] = \mathbb{E}\left[n_i\left(T\right)\mid\Gamma\right]\Pr\left(\Gamma\right) + \sum_{S\in\mathcal{P}(A)}\mathbb{E}\left[n_i\left(T\right)\mid F(S)\right]\Pr\left(F(S)\right) + \mathbb{E}\left[n_i\left(T\right)\mid\beta\right]\Pr\left(\beta\right)$$

$$(194)$$

$$\leq \max\left\{\mathbb{E}\left[n_i\left(T\right)\mid\Gamma\right], \left\{\mathbb{E}\left[n_i\left(T\right)\mid F(S)\right]\right\}_{S\in\mathcal{P}(S)}\right\} + \mathbb{E}\left[n_i\left(T\right)\mid\beta\right]\Pr\left(\beta\right) \quad (195)$$

$$(\because \Pr\left(\Gamma\right) + \sum\Pr\left(F(S)\right) < 1)$$

$$\leq \max\left\{\mathbb{E}\left[n_i\left(T\right)\mid\Gamma\right], \tau_{\sigma_i}\right\} + \mathbb{E}\left[n_i\left(T\right)\mid\beta\right]\Pr\left(\beta\right). \quad (196)$$

Where Equation 196 follows from the fact that conditioned on any $F(S)$, the maximum round up to which arm $i$ can be sampled is $\sigma_i$ both in the case when $\Gamma_i$ holds ($i \notin S$), and in the case when $i \in S$, and $F_i = \{\Sigma_f < \sigma_i\} \cap \{i \in \mathcal{A}_f\}$ holds instead. We now proceed by separately bounding the $\mathbb{E}\left[n_i\left(T\right)\mid\Gamma\right]$ term by further conditioning on the cases where $\Sigma_i > \rho_i$ and $\Sigma_i \leq \rho_i$.

$$\mathbb{E}\left[n_i(T)\mid\Gamma\right] = \mathbb{E}\left[n_i(T)\mid\Sigma_i\leq\rho_i,\Gamma\right]\Pr\left(\Sigma_i\leq\rho_i\mid\Gamma\right)$$
$$+ \mathbb{E}\left[n_i(T)\mid\Sigma_i>\rho_i,\Gamma\right]\Pr\left(\Sigma_i>\rho_i\mid\Gamma\right) \quad \text{(using Lemma D.3 on } \{\Sigma_i\leq\rho_i\})$$
$$(197)$$

$$\leq \mathbb{E}\left[n_i(T)\mid\Sigma_i\leq\rho_i,\Gamma\right] = \mathbb{E}_1\left[n_i(T)\right] \quad \text{(introducing shorthand notation } \mathbb{E}_1). \quad (198)$$

Where $\rho_i$ is as defined in Equation 34. In writing Equation 198 we leverage the fact that for a low cost arm with index $i < a^*$, $\Delta_{Q,i} = (1-\alpha)\mu_* - \mu_i$ is necessarily a smaller gap than $\Delta_i = \mu_* - \mu_i$ since by construction, each of these low cost arms has a return $\mu_i < \mu_{\text{CS}} = (1-\alpha)\mu_*$. It follows that $\Pr\left(\Sigma_i > \rho_i, \Gamma\right) = \Pr\left(\Sigma_i > \rho_i \mid \Gamma\right) = 0$ because the largest value that the random variable $\Sigma_i$ can take under $\Gamma$ is $\sigma_i$, and $\Delta_{Q,i} < \Delta_i \implies \rho_i > \sigma_i$.

BOUND ON $\mathbb{E}_1\left[n_i\left(T\right)\right]$

Since we enter the PE stage of the algorithm with a round number $\Sigma_i \leq \rho_i$, we use the same event construction of the compound event $G_i \forall i < a^*$, defined and used in the Proof of Lemma F.2. Therefore, just as before we work towards a bound by conditioning on the partition with $G_i$ introduced in Lemma F.1[4]. Let $\Lambda_i$ be the random variable denoting the highest round number corresponding to which sampling was performed for arm $i$ during the run of PE-CS.

$$\mathbb{E}_1\left[n_i(T)\right] = \mathbb{E}_1\left[n_i(T)\mid G_i\right]\Pr\left(G_i\right) + \mathbb{E}_1\left[n_i(T)\mid B_{1,i}\right]\Pr\left(B_{1,i}\right) + \mathbb{E}_1\left[n_i(T)\mid B_i\right]\Pr\left(B_i\right)$$

$$(199)$$

$$\leq \max\left\{\mathbb{E}_1\left[n_i(T)\mid G_i\right], \mathbb{E}_1\left[n_i(T)\mid B_{1,i}\right]\right\} + T\cdot\Pr\left(B_i\right) \quad (200)$$

$$(\text{Since } \Pr\left(G_i\right) + \Pr\left(B_{1,i}\right) < 1)$$

$$= \max\left\{\mathbb{E}_1\left[n_i(T)\mid G_i\right], \mathbb{E}_1\left[n_i(1, t_{\text{BAI}}) + n_i(t_{\text{BAI}}+1, T)\mid B_{1,i}\right]\right\} + T\cdot\Pr\left(B_i\right) \quad (201)$$

$$\leq \max\left\{\mathbb{E}_1\left[\tau_{\Lambda_i}\mid G_i\right], \mathbb{E}_1\left[\tau_{\Sigma_i}+0\mid B_{1,i}\right]\right\} + T\cdot\Pr\left(B_i\right) \quad (202)$$

---

[4]The probability operator $\Pr$ in the work that follows is also for the conditional distribution conditioned on $\{\Sigma_i \leq \rho_i, \Gamma\}$.

$$\text{($\tau_m$ as defined in Equation 35)}$$

$$\leq \max\left\{\tau_{\rho_i}, \tau_{\sigma_i}\right\} + T \cdot \Pr\left(B_i\right) \tag{203}$$

$$= \tau_{\rho_i} + T \cdot \Pr\left(B_i\right) \text{ (since $\rho_i > \sigma_i$ $\forall i < a^*$)} \tag{204}$$

$$< 1 + \frac{32 \log\left(T\Delta_{Q,i}^2\right)}{\Delta_{Q,i}^2} + T \cdot \Pr\left(B_i\right) \quad \text{(Using bound on $\tau_{\rho_i}$ in Equation 53).} \tag{205}$$

Where the treatment of the random variable $n_i(t_{\text{BAI}} + 1, T)$ (from definition F.3) is based on the expression for the additional rounds for which arm $i$ is sampled during the PE stage of PE-CS. In Equation 202 conditioned on $G_i$ there may be more samples, however conditioned on $B_{1,i}$ there shall be no further samples since the episode corresponding to $i$ is never initiated. Now to bound $\Pr\left(B_i \mid \Sigma_i \leq \rho_i, \Gamma\right)$ we use arguments similar to the ones in Proof of Lemma F.2.

$$\Pr\left(B_i \mid \Sigma_i \leq \rho_i, \Gamma\right) = \Pr\left(B_{1,i} \cup B_{2,i} \mid \Sigma_i \leq \rho_i, \Gamma\right). \tag{206}$$

Where $B_{1,i} = G_{1,i} \cap G_{2,i}^c$ and $B_{2,i} = G_{1,i} \cap G_{3,i}^c$ as in the proof of Lemma F.2. The Probability $\Pr\left(B_{1,i}\right)$ in Lemma F.2 was bound by the probability of either Clause 54 or Clause 55 being violated during round $\rho_i$. Due to the parallel nature of the construction here, and the possibility of round $\rho_i$ being conducted, we can upper bound $\Pr\left(B_{1,i} \mid \Sigma_i \leq \rho_i, \Gamma\right)$ identically as,

$$\Pr\left(B_{1,i} \mid \Sigma_i \leq \rho_i, \Gamma\right) \leq \frac{32}{T\Delta_{Q,i}^2}. \tag{207}$$

Now we move on to bounding $\Pr\left(B_{2,i} \mid \Sigma_i \leq \rho_i, \Gamma\right)$ by bounding the probability of arm $\ell$ being eliminated in any round $\omega_i = \rho$ lying in the range $\Sigma_i \leq \rho \leq \rho_i$. Just like in the proof of Lemma F.2, even here Clauses 54 and 56 holding simultaneously preclude arm $\ell$ from being eliminated by arm $i$ regardless of round $\rho$. Therefore using work identical to the one that goes into establishing Equations 72 and 73 we have,

$$\Pr\left(B_{2,i} \mid \Sigma_i \leq \rho_i, \Gamma\right) \leq \sum_{\rho=\Sigma_i}^{\rho_i - 1} \frac{4^\kappa + 1}{T\tilde{\Delta}_\rho^2} \tag{208}$$

$$\leq \sum_{\rho=0}^{\rho_i - 1} \frac{4^\kappa + 1}{T\tilde{\Delta}_\rho^2} \text{ (Because $\Sigma_i \geq 0$)} \tag{209}$$

$$< \frac{11}{T\Delta_{Q,i}^2} \text{ (From Equation 73 in Lemma F.2, $\kappa = 0$).} \tag{210}$$

Applying the Union bound to Equation 206, and plugging in the bounds in 207 and 210, we have,

$$\Pr\left(B_i \mid \Sigma_i \leq \rho_i, \Gamma\right) < \frac{43}{T\Delta_{Q,i}^2}. \tag{211}$$

Substituting the bounds in Equations 211, 205, and Lemma G.2 into Equation 196 we obtain,

$$\mathbb{E}\left[n_i(T)\right] \leq \max\left\{\tau_{\rho_i} + T \cdot \Pr\left(B_i\right), \tau_{\sigma_i}\right\} + \mathbb{E}\left[n_i(T) \mid \beta\right] \Pr\left(\beta\right) \tag{212}$$

$$< 1 + \frac{32 \log\left(T\Delta_{Q,i}^2\right)}{\Delta_{Q,i}^2} + \frac{43}{\Delta_{Q,i}^2} + \mathbb{E}\left[n_i(T) \mid \beta\right] \Pr\left(\beta\right). \tag{213}$$

Which when combined with Lemma G.2 is the bound stated in Lemma G.3. □

BOUND ON THE NUMBER OF SAMPLES FOR ARM $i^*$

**Lemma G.4** (Bound on the expected number of samples of the maximum reward arm). *For the arm $i^* = \arg\max_{i \in [K]} \mu_i$, the expected number of samples accrued is upper bounded as,*

$$\mathbb{E}\left[n_{i^*}(T)\right] < 1 + \max_{\Delta \in \boldsymbol{\Delta}}\left\{\frac{32 \log\left(T\Delta^2\right)}{\Delta^2}\right\} + \sum_{i=1}^{a^*} \frac{43}{\Delta_{Q,i}^2} + \frac{32}{\Delta_{a^*}^2}$$

$$+ \mathbb{E}\left[n_{i^*}(T) \mid \{Z > a^*\}, \Gamma\right] \cdot \left(\frac{32}{T\Delta_{a^*}^2} + \frac{43}{T\Delta_{Q,a^*}^2}\right)$$

$$+ \mathbb{E}\left[n_{i^*}(T) \mid \beta\right] \left(\frac{11}{T\Delta_{\min}^2} + \sum_{j \neq i^*} \frac{32}{T\Delta_j^2}\right)$$

*Where* $\boldsymbol{\Delta} = \{\Delta_{min}\} \cup \{\Delta_{Q,j}\}_{j \leq a^*}$.

*Proof.* Just like in the proof of Lemma G.3 we sequester away outcomes of the BAI stage that make analysis of the PE stage intractable.

$$\mathbb{E}\left[n_{i^*}(T)\right] = \mathbb{E}\left[n_{i^*}(T) \mid \Gamma\right]\Pr(\Gamma) + \sum_{S \in \mathcal{P}(A)} \mathbb{E}\left[n_{i^*}(T) \mid F(S)\right]\Pr(F(S)) + \mathbb{E}\left[n_{i^*}(T) \mid \beta\right]\Pr(\beta)$$

(214)

$$\leq \max\left\{\mathbb{E}\left[n_{i^*}(T) \mid \Gamma\right], \{\mathbb{E}\left[n_{i^*}(T) \mid F(S)\right]\}_{S \in \mathcal{P}(S)}\right\} + \mathbb{E}\left[n_{i^*}(T) \mid \beta\right]\Pr(\beta)$$

(215)

$$(\because \Pr(\Gamma) + \Pr(F) < 1)$$

$$\leq \max\left\{\mathbb{E}\left[n_{i^*}(T) \mid \Gamma\right], \tau_{\sigma_{\max}}\right\} + \mathbb{E}\left[n_{i^*}(T) \mid \beta\right]\Pr(\beta).$$

(216)

Equation 216 results from the observation that the maximum round number up till which any arm is sampled during the BAI stage under $F(S) \; \forall \, S \in \mathcal{P}(A)$ is $\sigma_{\max}$. Conditioned on $\Gamma$, the reference arm $\ell$ is identified correctly to be $i^*$. Consequently, the expected number of samples $\mathbb{E}\left[n_{i^*}(T) \mid \Gamma\right]$ can be analyzed in a manner that closely parallels the analysis of $\mathbb{E}\left[n_\ell(T)\right]$ in the Proof of Lemma F.3.

The key difference from Lemma F.3 is that we grapple with the case $\Sigma_{a^*} > \rho_{a^*}$. This possibility arises because our bandit instance may have $\Delta_{a^*} < \Delta_{Q,a^*}$, which in turn implies that $\Pr\left(\rho_{a^*} < \Sigma_{a^*} < \sigma_{a^*} \mid \Gamma\right) > 0$. The outcome $\Sigma_{a^*} > \rho_{a^*}$ skips the checks associated with the events $G_{2,a^*}$ and $G_{3,a^*}$. Mathematically, this means that for the event $G_{a^*}$ as defined in Equation 36 we shall have $\Pr\left(G_{1,a^*} \cap ((G_{2,a^*} \cap G_{3,a^*}) \cup E_{a^*}) \mid \Sigma_{a^*} > \rho_{a^*}, \Gamma\right) = 0$. This motivates us to expand the scope of the good event $G_{a^*}$.

First we define new event $G_{4,a^*} \subset \Omega$ and then we augment the definition of $G_{a^*}$ using this new event.

$G_{4,a^*}$ : {Arm $\ell$ is eliminated by arm $a^*$ in round $\Sigma_{a^*}$, during ep. $a^*$ of the PE stage of PE-CS}

**Remark G.4.** *Since arm $a^*$ enters the PE stage of PE-CS with samples corresponding to $\Sigma_{a^*}$ number of rounds already accrued, checking whether the event $G_{4,a^*}$ holds does not involve any further sampling of arms.*

The definition of $G_{a^*}$ is now changed to the one in Equation 217 which supersedes the prior generic $G_{a^*}$ (Equation 36).

$$G_{a^*} = \underbrace{G_{1,a^*}}_{\text{Ep. } a^* \text{ is executed}} \cap \left(\underbrace{((G_{2,a^*} \cap G_{3,a^*}) \cup E_{a^*})}_{\text{Prior } G_{a^*} \text{ Clause}} \cup \underbrace{(G_{4,a^*} \cap \{\Sigma_{a^*} > \rho_{a^*}\})}_{\text{New Clause}}\right).$$

(217)

**Remark G.5** (Implicit event in Prior $G_{a^*}$ clause)**.** *We remark here that the event $\{\Sigma_{a^*} \leq \rho_{a^*}\}$ is implicit in the event $G_{1,a^*} \cap ((G_{2,a^*} \cap G_{3,a^*}) \cup E_{a^*})$ i.e. $G_{1,a^*} \cap ((G_{2,a^*} \cap G_{3,a^*}) \cup E_{a^*}) \subseteq \{\Sigma_{a^*} \leq \rho_{a^*}\}$. This is because the event is contingent on correct eliminations happening leading up to and during $\rho_{a^*}$.*

While retaining the definition of $G$ from Equation 74, and the definition of $B_{1,a^*}$ from Lemma F.1 we redefine $B_{a^*}$ so that the relation $G_{a^*}^c = B_{1,a^*} \cup B_{a^*}$ continues to hold.

$$G_{a^*} = G_{1,a^*} \cap (((G_{2,a^*} \cap G_{3,a^*}) \cup E_{a^*}) \cup (G_{4,a^*} \cap \{\Sigma_{a^*} > \rho_{a^*}\}))$$

(218)

$$= G_{1,a^*} \cap ((((G_{2,a^*} \cap G_{3,a^*}) \cup E_{a^*})) \cap \{\Sigma_{a^*} \leq \rho_{a^*}\} \cup (G_{4,a^*} \cap \{\Sigma_{a^*} > \rho_{a^*}\}))$$

(219)

(due to Remark G.5)

$$\implies G_{a^*}^c = G_{1,a^*}^c \cup \left( \underbrace{(((G_{2,a^*}^c \cup G_{3,a^*}^c) \cap E_{a^*}^c) \cup \{\Sigma_{a^*} > \rho_{a^*}\})}_{B_{a^*}^{\text{original}}} \cap \left( G_{4,a^*}^c \cup \{\Sigma_{a^*} \le \rho_{a^*}\} \right) \right)$$

(220)

$$= B_{1,a^*} \cup \left( \left( \left( B_{a^*}^{\text{original}} \cap \{\Sigma_{a^*} \le \rho_{a^*}\} \right) \cup \{\Sigma_{a^*} > \rho_{a^*}\} \right) \right.$$
$$\left. \cap \left( (G_{4,a^*}^c \cap \{\Sigma_{a^*} > \rho_{a^*}\}) \cup \{\Sigma_{a^*} \le \rho_{a^*}\} \right) \right)$$

(221)

$$= B_{1,a^*} \cup \underbrace{\left( \left( B_{a^*}^{\text{original}} \cap \{\Sigma_{a^*} \le \rho_{a^*}\} \right) \cup \left( G_{4,a^*}^c \cap \{\Sigma_{a^*} > \rho_{a^*}\} \right) \right)}_{B_{a^*}}.$$

(222)

Where Equation 221 follows from $A \cup B = (A \cap B^c) \cup B$ being applied to both the left and right clauses. Equation 222 gives us the updated definition of the event $B_{a^*}$.

To continue bounding $\mathbb{E}\left[n_{i^*}(T) \mid \Gamma\right]$ from Equation 216, we shall condition on the collection of mutually exclusive and exhaustive events $G, G^c \cap \{Z \le a^*\}$, and $G^c \cap \{Z > a^*\}$ earlier used in Appendix F. Armed with the updated event $B_{a^*}$, we can now develop a bound for $\Pr\left(G^c \cap \{Z \le a^*\} \mid \Gamma\right)$ using the result in Lemma F.6. By trivially generalizing the proof of Lemma F.6 to the scenario for this proof where we condition on $\Gamma$ we have,

$$\Pr\left(G^c \cap \{Z \le a^*\} \mid \Gamma\right) \le \sum_{i=1}^{a^*} \Pr\left(B_j \mid \Gamma\right)$$

(223)

$$\le \sum_{i=1}^{a^*-1} \frac{43}{T\Delta_{Q,i}^2} + \Pr\left(B_{a^*} \mid \Gamma\right),$$

(224)

from Equation 211, and $\Pr\left(\Sigma_i > \rho_i \mid \Gamma\right) = 0 \ \forall i < a^*$. We need now only develop a bound for $\Pr\left(B_{a^*} \mid \Gamma\right)$ under its definition in Equation 222.

$$\Pr\left(B_{a^*} \mid \Gamma\right) = \Pr\left( \left( B_{a^*}^{\text{original}} \cap \{\Sigma_{a^*} \le \rho_{a^*}\} \right) \cup \left( G_{4,a^*}^c \cap \{\Sigma_{a^*} > \rho_{a^*}\} \right) \mid \Gamma \right)$$

(225)

$$\le \Pr\left( B_{a^*}^{\text{original}} \cap \{\Sigma_{a^*} \le \rho_{a^*}\} \mid \Gamma \right) + \Pr\left( G_{4,a^*}^c \cap \{\Sigma_{a^*} > \rho_{a^*}\} \mid \Gamma \right)$$

(226)

(Union Bound)

$$\le \frac{43}{T\Delta_{Q,a^*}^2} + \Pr\left( G_{4,a^*}^c \cap \{\Sigma_{a^*} > \rho_{a^*}\} \mid \Gamma \right) \quad \text{(From Equation 137).}$$

(227)

From Equation 132 in the Proof of Lemma F.2, we have the probability of arm $\ell$ not being eliminated by arm $a^*$ during round $\omega_{a^*}$ of episode $a^*$ as being upper bounded by $\frac{2}{T\tilde{\Delta}_{\omega_{a^*}}^2}$. Therefore since elimination under $G_{4,a^*}$ happens during episode $a^*$ in round $\Sigma_{a^*}$,

$$\Pr\left( G_{4,a^*}^c \cap \{\Sigma_{a^*} > \rho_{a^*}\} \mid \Gamma \right) \le \frac{2}{T\tilde{\Delta}_{\sigma_{a^*}}^2}$$

(228)

(since $\Pr\left(\Sigma_{a^*} > \sigma_{a^*} \mid \Gamma\right) = 0$, and $\tilde{\Delta}_m$ decreases with $m$).

$$\le \frac{32}{T\Delta_{a^*}^2} \quad \text{(since by definition of } \sigma_i, \tilde{\Delta}_{\sigma_i} \ge \frac{\Delta_i}{4} \text{).}$$

(229)

Combining the bounds in Equations 227 and 229 with the Expression 224 we have,

$$\Pr\left(G^c \cap \{Z \le a^*\} \mid \Gamma\right) \le \sum_{i=1}^{a^*-1} \frac{43}{T\Delta_{Q,i}^2} + \left( \frac{43}{T\Delta_{Q,a^*}^2} + \frac{32}{T\Delta_{a^*}^2} \right).$$

(230)

We are now in a position to bound $\mathbb{E}\left[n_{i^*}(T) \mid \Gamma\right]$. For brevity of notation we use the symbols $\mathbb{E}_\Gamma$ and $\Pr_\Gamma$ to denote expectation and probability conditioned on $\Gamma$.

$$\mathbb{E}_\Gamma\left[n_{i^*}(T)\right] = \mathbb{E}_\Gamma\left[n_{i^*}(T) \mid G\right] \Pr_\Gamma\left(G\right) + \mathbb{E}_\Gamma\left[n_{i^*}(T) \mid G^c \cap \{Z \le a^*\}\right] \Pr_\Gamma\left(G^c \cap \{Z \le a^*\}\right)$$

$$+ \mathbb{E}_\Gamma \left[ n_{i^*}(T) \mid G^c \cap \{ Z > a^* \} \right] \Pr_\Gamma \left( G^c \cap \{ Z > a^* \} \right) \tag{231}$$

$$\leq \mathbb{E}_\Gamma \left[ n_{i^*}(T) \mid G \right] + T \cdot \Pr_\Gamma \left( G^c \cap \{ Z \leq a^* \} \right)$$
$$+ \mathbb{E}_\Gamma \left[ n_{i^*}(T) \mid \{ Z > a^* \} \right] \Pr_\Gamma \left( \{ Z > a^* \} \right) \quad (\because \{ Z > a^* \} \subseteq G^c) \tag{232}$$

$$\leq \mathbb{E}_\Gamma \left[ n_{i^*}(T) \mid G \right] + \sum_{i=1}^{a^*-1} \frac{43}{\Delta_{Q,i}^2} + \left( \frac{43}{\Delta_{Q,a^*}^2} + \frac{32}{\Delta_{a^*}^2} \right) +$$
$$+ \mathbb{E}_\Gamma \left[ n_{i^*}(T) \mid \{ Z > a^* \} \right] \Pr_\Gamma \left( \{ Z > a^* \} \right). \tag{233}$$

To bound $\mathbb{E}\left[ n_{i^*}(T) \mid G, \Gamma \right]$ we leverage the fact that due to the samples of the reference arm being $\Pr\left( n_{i^*}(T) > \max_{i \neq i^*} n_i(1, t_Z) \mid G, \Gamma \right) = 0^5$. Conditioned on $G, \Gamma$ there can be no further sampling of the best arm $i^*$ beyond time $t_Z$, since under $\Gamma$ the best arm $i^*$ is the reference arm. Consequently,

$$\mathbb{E}\left[ n_{i^*}(T) \mid G, \Gamma \right] \leq \mathbb{E}\left[ \max_{i \neq i^*} n_i(1, t_Z) \mid G, \Gamma \right] \tag{234}$$

$$\leq \mathbb{E}\left[ \max_{i \neq i^*} n_i(1, t_{a^*}) \mid G, \Gamma \right] \quad \text{(because } \Pr\left( Z > a^* \mid G, \Gamma \right) = 0) \tag{235}$$

$$= \mathbb{E}\left[ \max \left\{ \{ n_i(1, t_{a^*}) \}_{i < a^*}, n_{a^*}(1, t_{a^*}), \{ n_i(1, t_{a^*}) \}_{i > a^*} \right\} \mid G, \Gamma \right] \tag{236}$$

$$\leq \mathbb{E}\left[ \max \left\{ \{ \tau_{\Lambda_i} \}_{i < a^*}, \tau_{\Lambda_{a^*}}, \{ \tau_{\Sigma_i} \}_{i > a^*} \right\} \mid G, \Gamma \right] \tag{237}$$
$$\text{(using definitions of } \tau, \Lambda, \text{ and } \Sigma)$$

$$\leq \max \left\{ \{ \tau_{\rho_i} \}_{i < a^*}, \max \left\{ \tau_{\rho_{a^*}}, \tau_{\sigma_{a^*}} \right\}, \{ \tau_{\sigma_i} \}_{i > a^*} \right\} \tag{238}$$
$$\text{(because } \Pr\left( \Lambda_{a^*} > \max \{ \rho_{a^*}, \sigma_{a^*} \} \right) = 0)$$

$$\leq \max \left\{ \{ \tau_{\rho_i} \}_{i \leq a^*}, \tau_{\sigma_{\max}} \right\}. \tag{239}$$

Plugging Equation 239 into 233 and the subsequent result into Equation 216 we obtain the statement of Lemma G.4,

$$\mathbb{E}\left[ n_{i^*}(T) \right] \leq \max \left\{ \{ \tau_{\rho_i} \}_{i \leq a^*}, \tau_{\sigma_{\max}} \right\} + \sum_{i=1}^{a^*-1} \frac{43}{\Delta_{Q,i}^2} + \left( \frac{43}{\Delta_{Q,a^*}^2} + \frac{32}{\Delta_{a^*}^2} \right)$$
$$+ \mathbb{E}\left[ n_{i^*}(T) \mid \{ Z > a^* \}, \Gamma \right] \Pr\left( \{ Z > a^* \} \mid \Gamma \right) + \mathbb{E}\left[ n_{i^*}(T) \mid \beta \right] \Pr(\beta) \tag{240}$$

$$< 1 + \max \left\{ \frac{32 \log \left( T \Delta_{\min}^2 \right)}{\Delta_{\min}^2}, \left\{ \frac{32 \log \left( T \Delta_{Q,i}^2 \right)}{\Delta_{Q,i}^2} \right\}_{i \leq a^*} \right\} + \sum_{i=1}^{a^*} \frac{43}{\Delta_{Q,i}^2} + \frac{32}{\Delta_{a^*}^2}$$

$$+ \mathbb{E}\left[ n_{i^*}(T) \mid \{ Z > a^* \}, \Gamma \right] \cdot \left( \frac{32}{T \Delta_{a^*}^2} + \frac{43}{T \Delta_{Q,a^*}^2} \right)$$

$$+ \mathbb{E}\left[ n_{i^*}(T) \mid \beta \right] \left( \frac{11}{T \Delta_{\min}^2} + \sum_{j \neq i^*} \frac{32}{T \Delta_j^2} \right). \tag{241}$$

Where Equation 241 follows from from the bound on $\tau_{\rho_i}$ shown in Equation 52 and from Lemma G.2. In reaching the bound used for $\Pr\left( \{ Z > a^* \} \mid \Gamma \right)$ we use $\{ Z > a^* \} \subseteq B_{a^*}$, and the bound on $\Pr\left( B_{a^*} \mid \Gamma \right)$ shown leading up to Equation 230. $\qquad \square$

UPPER BOUND ON THE NUMBER OF SAMPLES FOR ARMS $i > a^*, i \neq i^*$

**Lemma G.5** (Bound on the expected number of samples of high-cost arms). *For any high cost arm with $i > a^*, i \neq i^*$, its expected number of samples are upper bounded as,*

$$\mathbb{E}\left[ n_i(T) \right] < 1 + \frac{32 \log \left( T \Delta_i^2 \right)}{\Delta_i^2} + \mathbb{E}\left[ n_i(T) \mid \{ Z > a^* \}, \Gamma \right] \cdot \left( \frac{32}{T \Delta_{a^*}^2} + \frac{43}{T \Delta_{Q,a^*}^2} \right)$$

---

[5]This is because in PE-CS all sampling of arm $i^*$, or of any arm for that matter, leading up to time $t_Z$ is always matched exactly by the number of samples of another arm.

$$+ \mathbb{E}\left[n_i\left(T\right) \mid \beta\right] \cdot \left(\frac{11}{T\Delta_{min}^2} + \sum_{j \neq i^*} \frac{32}{T\Delta_j^2}\right).$$

*Proof.* To prove the bound stated in Lemma G.5 we proceed with initial steps identical to the ones that go into proving Lemmas G.3 and G.4.

$$\mathbb{E}\left[n_i(T)\right] = \mathbb{E}\left[n_i(T) \mid \Gamma\right]\Pr\left(\Gamma\right) + \sum_{S \in \mathcal{P}(A)} \mathbb{E}\left[n_i(T) \mid F(S)\right]\Pr\left(F(S)\right) + \mathbb{E}\left[n_i(T) \mid \beta\right]\Pr\left(\beta\right) \tag{242}$$

$$\leq \max\left\{\mathbb{E}\left[n_i(T) \mid \Gamma\right], \max_{S \in \mathcal{P}(A)} \mathbb{E}\left[n_i(T) \mid F(S)\right]\right\} + \mathbb{E}\left[n_i(T) \mid \beta\right]\Pr\left(\beta\right) \tag{243}$$

$$\left(\because \Pr\left(\Gamma\right) + \Pr\left(F\right) < 1\right)$$

$$\leq \max\left\{\mathbb{E}\left[n_i(T) \mid \Gamma\right], \tau_{\sigma_i}\right\} + \mathbb{E}\left[n_i(T) \mid \beta\right]\left(\frac{11}{T\Delta_{\min}^2} + \sum_{j \neq i^*} \frac{32}{T\Delta_j^2}\right). \tag{244}$$

Where the final bound is from Lemma G.2. Now we must bound the expectation term $\mathbb{E}\left[n_i(T) \mid \Gamma\right]$. For this we recognize that during the PE-stage, the critical event which determines the number of samples further accrued for a high-cost arm is whether the final PE episode $Z > a^*$ or not. In the case when $Z \leq a^*$, there are no further samples of arm $i$ accrued beyond the BAI-stage.

$$\mathbb{E}\left[n_i(T) \mid \Gamma\right] = \mathbb{E}\left[n_i(T) \mid \{Z > a^*\}, \Gamma\right]\Pr\left(Z > a^* \mid \Gamma\right)$$
$$+ \mathbb{E}\left[n_i(T) \mid \{Z \leq a^*\}, \Gamma\right]\Pr\left(Z \leq a^* \mid \Gamma\right) \tag{245}$$
$$\leq \tau_{\sigma_i} + \mathbb{E}\left[n_i(T) \mid \{Z > a^*\}, \Gamma\right]\Pr\left(B_{a^*} \mid \Gamma\right) \quad \text{(since } \{Z > a^*\} \subseteq B_{a^*}\text{)} \tag{246}$$

$$\leq \tau_{\sigma_i} + \mathbb{E}\left[n_i(T) \mid \{Z > a^*\}, \Gamma\right] \cdot \left(\frac{32}{T\Delta_{a^*}^2} + \frac{43}{T\Delta_{Q,a^*}^2}\right). \tag{247}$$

Where the final line follows by using the bound developed in Equation 227. Returning to bounding $\mathbb{E}\left[n_i(T)\right]$ we obtain the bound stated in Lemma G.5,

$$\mathbb{E}\left[n_i(T)\right] \leq \tau_{\sigma_i} + \mathbb{E}\left[n_i(T) \mid \{Z > a^*\}, \Gamma\right] \cdot \left(\frac{32}{T\Delta_{a^*}^2} + \frac{43}{T\Delta_{Q,a^*}^2}\right)$$

$$+ \mathbb{E}\left[n_i(T) \mid \beta\right]\left(\frac{11}{T\Delta_{\min}^2} + \sum_{j \neq i^*} \frac{32}{T\Delta_j^2}\right) \tag{248}$$

$$< 1 + \frac{32\log\left(T\Delta_i^2\right)}{\Delta_i^2} + \mathbb{E}\left[n_i(T) \mid \{Z > a^*\}, \Gamma\right] \cdot \left(\frac{32}{T\Delta_{a^*}^2} + \frac{43}{T\Delta_{Q,a^*}^2}\right)$$

$$+ \mathbb{E}\left[n_i(T) \mid \beta\right]\left(\frac{11}{T\Delta_{\min}^2} + \sum_{j \neq i^*} \frac{32}{T\Delta_j^2}\right). \tag{249}$$

$$\square$$

Finally, we have all the pieces needed to prove Theorem 3.4. We combine the results obtained in Lemmas G.3, G.4, and G.5 by adding together the contributions to regret of the three categories of arms while coalescing terms.

*Proof of Theorem 3.4.* Using Equation 13 from the regret decomposition Lemma D.5 we can express and bound the expected cumulative cost regret as,

$$\mathbb{E}\left[\text{Cost\_Reg}\left(T, \nu\right)\right] \tag{250}$$

$$= \sum_{i=1}^{K} \Delta_{C,i}^{+} \mathbb{E}\left[n_i\left(T\right)\right] \tag{251}$$

$$= \sum_{i>a^*, i\neq i^*} \Delta_{C,i}^{+} \mathbb{E}\left[n_i\left(T\right)\right] + \Delta_{C,i^*}^{+} \mathbb{E}\left[n_{i^*}\left(T\right)\right] \quad \text{(because } \Delta_{C,i}^{+} = 0, \ \forall i \leq a^*\text{)} \tag{252}$$

$$< \sum_{i>a^*, i\neq i^*} \Delta_{C,i}^{+} \left(1 + \frac{32\log\left(T\Delta_i^2\right)}{\Delta_i^2} + \mathbb{E}\left[n_i\left(T\right) \mid \{Z > a^*\}, \Gamma\right] \cdot \left(\frac{32}{T\Delta_{a^*}^2} + \frac{43}{T\Delta_{Q,a^*}^2}\right)\right.$$
$$\left. + \mathbb{E}\left[n_i\left(T\right) \mid \beta\right] \cdot \left(\frac{11}{T\Delta_{\min}^2} + \sum_{j\neq i^*} \frac{32}{T\Delta_j^2}\right)\right)$$
$$+ \Delta_{C,i^*}^{+} \left(1 + \max_{\Delta \in \mathbf{\Delta}} \left\{\frac{32\log\left(T\Delta^2\right)}{\Delta^2}\right\} + \sum_{i=1}^{a^*} \frac{43}{\Delta_{Q,i}^2} + \frac{32}{\Delta_{a^*}^2}\right.$$
$$+ \mathbb{E}\left[n_{i^*}(T) \mid \{Z > a^*\}, \Gamma\right] \cdot \left(\frac{32}{T\Delta_{a^*}^2} + \frac{43}{T\Delta_{Q,a^*}^2}\right)$$
$$\left. + \mathbb{E}\left[n_{i^*}\left(T\right) \mid \beta\right] \cdot \left(\frac{11}{T\Delta_{\min}^2} + \sum_{j\neq i^*} \frac{32}{T\Delta_j^2}\right)\right) \tag{253}$$

$$= \sum_{i>a^*} \Delta_{C,i}^{+} \mathbb{E}\left[n_i\left(T\right) \mid \beta\right] \cdot \left(\frac{11}{T\Delta_{\min}^2} + \sum_{j\neq i^*} \frac{32}{T\Delta_j^2}\right) + \sum_{i>a^*, i\neq i^*} \Delta_{C,i}^{+} \left(1 + \frac{32\log\left(T\Delta_i^2\right)}{\Delta_i^2}\right)$$
$$+ \Delta_{C,i^*}^{+} \left(1 + \max_{\Delta \in \mathbf{\Delta}} \left\{\frac{32\log\left(T\Delta^2\right)}{\Delta^2}\right\} + \sum_{i=1}^{a^*} \frac{43}{\Delta_{Q,i}^2} + \frac{32}{\Delta_{a^*}^2}\right)$$
$$+ \sum_{i>a^*} \Delta_{C,i}^{+} \mathbb{E}\left[n_i\left(T\right) \mid \{Z > a^*\}, \Gamma\right] \cdot \left(\frac{32}{T\Delta_{a^*}^2} + \frac{43}{T\Delta_{Q,a^*}^2}\right) \tag{254}$$

$$\leq \max_{i\in[K]} \Delta_{C,i}^{+} \left(\frac{11}{\Delta_{\min}^2} + \sum_{j\neq i^*} \frac{32}{\Delta_j^2}\right) + \sum_{i>a^*, i\neq i^*} \Delta_{C,i}^{+} \left(1 + \frac{32\log\left(T\Delta_i^2\right)}{\Delta_i^2}\right)$$
$$+ \Delta_{C,i^*}^{+} \left(1 + \max_{\Delta \in \mathbf{\Delta}} \left\{\frac{32\log\left(T\Delta^2\right)}{\Delta^2}\right\} + \sum_{i=1}^{a^*} \frac{43}{\Delta_{Q,i}^2} + \frac{32}{\Delta_{a^*}^2}\right)$$
$$+ \max_{i>a^*, i\in[K]} \Delta_{C,i}^{+} \left(\frac{32}{\Delta_{a^*}^2} + \frac{43}{\Delta_{Q,a^*}^2}\right). \tag{255}$$

Where the final expression stated in Equation 255 is derived using $\sum_{i>a^*} \Delta_{C,i}^{+} \mathbb{E}\left[n_i(T) \mid X\right] \leq \max_{i\in[K]} \Delta_{C,i}^{+} \cdot T$. Here $X$ is used as a placeholder for conditioning on $\beta$ or $\{Z > a^*\}, \Gamma$.

Proceeding identically, for Quality Regret we shall have,

$$\mathbb{E}\left[\text{Quality\_Reg}\left(T, \nu\right)\right] \tag{256}$$

$$= \sum_{i=1}^{K} \Delta_{Q,i}^{+} \mathbb{E}\left[n_i\left(T\right)\right] \tag{257}$$

$$= \sum_{i<a^*} \Delta_{Q,i} \mathbb{E}\left[n_i\left(T\right)\right] + \sum_{i>a^*, i\neq i^*} \Delta_{Q,i}^{+} \mathbb{E}\left[n_i\left(T\right)\right] \tag{258}$$

$$< \sum_{i<a^*} \Delta_{Q,i} \left(1 + \frac{32\log\left(T\Delta_{Q,i}^2\right)}{\Delta_{Q,i}^2} + \frac{43}{\Delta_{Q,i}^2} + \mathbb{E}\left[n_i\left(T\right) \mid \beta\right] \cdot \left(\frac{11}{T\Delta_{\min}^2} + \sum_{j\neq i^*} \frac{32}{T\Delta_j^2}\right)\right) \tag{259}$$

$$+ \sum_{i>a^*, i\neq i^*} \Delta_{Q,i}^+ \left(1 + \frac{32\log\left(T\Delta_i^2\right)}{\Delta_i^2} + \mathbb{E}\left[n_i\left(T\right) \mid \{Z > a^*\}, \Gamma\right] \cdot \left(\frac{32}{T\Delta_{a^*}^2} + \frac{43}{T\Delta_{Q,a^*}^2}\right)\right.$$

$$\left. + \mathbb{E}\left[n_i\left(T\right) \mid \beta\right] \cdot \left(\frac{11}{T\Delta_{\min}^2} + \sum_{j\neq i^*} \frac{32}{T\Delta_j^2}\right)\right)$$

$$\leq \sum_{i<a^*} \left(\frac{32\log\left(T\Delta_{Q,i}^2\right)}{\Delta_{Q,i}} + \frac{43}{\Delta_{Q,i}}\right) + \left(\frac{11}{T\Delta_{\min}^2} + \sum_{j\neq i^*} \frac{32}{T\Delta_j^2}\right) \max_{i\in[K]} \Delta_{Q,i}^+ \sum_{i=1}^K \mathbb{E}\left[n_i(T) \mid \beta\right]$$

$$+ \left(\frac{32}{T\Delta_{a^*}^2} + \frac{43}{T\Delta_{Q,a^*}^2}\right) \max_{i>a^*} \Delta_{Q,i}^+ \sum_{i>a^*, i\neq i^*} \mathbb{E}\left[n_i(T) \mid \{Z > a^*\}, \Gamma\right] + \sum_{i=1}^K \Delta_{Q,i}^+$$

$$+ \sum_{i>a^*, i\neq i^*} \Delta_{Q,i}^+ \frac{32\log\left(T\Delta_i^2\right)}{\Delta_i^2} \tag{260}$$

$$\leq \sum_{i<a^*} \left(\frac{32\log\left(T\Delta_{Q,i}^2\right)}{\Delta_{Q,i}} + \frac{43}{\Delta_{Q,i}}\right) + \sum_{i>a^*, i\neq i^*} \Delta_{Q,i}^+ \frac{32\log\left(T\Delta_i^2\right)}{\Delta_i^2}$$

$$+ \max_{i>a^*} \Delta_{Q,i}^+ \left(\frac{32}{\Delta_{a^*}^2} + \frac{43}{\Delta_{Q,a^*}^2}\right) + \max_{i\in[K]} \Delta_{Q,i}^+ \left(\frac{11}{\Delta_{\min}^2} + \sum_{j\neq i^*} \frac{32}{\Delta_j^2}\right) + \sum_{i=1}^K \Delta_{Q,i}^+. \tag{261}$$

Where Equation 261 is the upper bound on expected cumulative quality regret stated in Theorem 3.4 and follows from the linearity of the expectation operator and the total sample budget being $T$. $\square$

Similar to the description that followed the proof of Theorem3.2, we rearrange the various terms in the regret bound and provide their interpretation as underbraces.

$$\underbrace{\Delta_{C,i^*}^+ \left(1 + \max_{\Delta\in\mathbf{\Delta}} \left\{\frac{32\log\left(T\Delta^2\right)}{\Delta^2}\right\}\right)}_{\substack{\text{Contribution from } i^* \text{ under nominal} \\ \text{termination in PE-stage episode } a^*}} + \underbrace{\Delta_{C,i^*}^+ \left(\sum_{i=1}^{a^*-1} \frac{43}{\Delta_{Q,i}^2} + \left(\frac{32}{\Delta_{a^*}^2} + \frac{43}{\Delta_{Q,a^*}^2}\right)\right)}_{\substack{\text{Contribution from } i^* \text{ under} \\ \text{mis-termination in PE-stage episode } \leq a^*}}$$

$$+ \underbrace{\sum_{i>a^*, i\in[K]\setminus\{i^*\}} \Delta_{C,i}^+ \left(1 + \frac{32\log\left(T\Delta_i^2\right)}{\Delta_i^2}\right)}_{\substack{\text{Contribution from high-cost arms} \\ \text{with a proper end to the BAI-stage}}} + \underbrace{\max_{i>a^*, i\in[K]} \Delta_{C,i}^+ \left(\frac{32}{\Delta_{a^*}^2} + \frac{43}{\Delta_{Q,a^*}^2}\right)}_{\substack{\text{Contribution from PE-stage episodes } >a^* \\ \text{in case of mis-termination during ep } a^*}}$$

$$+ \underbrace{\max_{i\in[K]} \Delta_{C,i}^+ \left(\frac{11}{\Delta_{\min}^2} + \sum_{j\neq i^*} \frac{32}{\Delta_j^2}\right)}_{\substack{\text{Contribution from improper} \\ \text{end to BAI stage}}}.$$

And for quality regret,

$$\underbrace{\sum_{i=1}^{a^*-1} \left(\Delta_{Q,i} + \frac{32\log\left(T\Delta_{Q,i}^2\right)}{\Delta_{Q,i}}\right)}_{\substack{\text{Contribution from } i < a^* \text{ under nominal} \\ \text{termination in PE-stage episode } a^*}} + \underbrace{\sum_{i=1}^{a^*-1} \frac{43}{\Delta_{Q,i}}}_{\substack{\text{Contribution from} \\ i < a^* \text{ under} \\ \text{mis-termination in} \\ \text{PE-stage episode } \leq a^*}} + \underbrace{\max_{i>a^*, i\in[K]} \Delta_{Q,i}^+ \left(\frac{32}{\Delta_{a^*}^2} + \frac{43}{\Delta_{Q,a^*}^2}\right)}_{\substack{\text{Contribution from PE-stage episodes } >a^* \\ \text{in case of mis-termination during ep } a^*}}$$

$$+ \underbrace{\sum_{i>a^*, i\in[K]\setminus\{i^*\}} \Delta_{Q,i}^+ \left(1 + \frac{32\log\left(T\Delta_i^2\right)}{\Delta_i^2}\right)}_{\substack{\text{Contribution from PE-stage episodes } >a^* \\ \text{in case of mis-termination during ep } a^*}} + \underbrace{\max_{i\in[K]} \Delta_{Q,i}^+ \left(\frac{11}{\Delta_{\min}^2} + \sum_{j\neq i^*} \frac{32}{\Delta_j^2}\right)}_{\substack{\text{Contribution from improper} \\ \text{end to BAI stage}}}.$$

Where $\Delta_i = \mu^* - \mu_i$ are conventional gaps, $\Delta_{\min} = \mu^* - \max_{i \neq i^*} \mu_i$ is the smallest conventional gap, and $\boldsymbol{\Delta} = \{\Delta_{\min}\} \cup \{\Delta_{Q,j}\}_{j \leq a^*}$ is defined especially to bound samples of arm $i^*$ under expectation.

# H    ALGORITHMS AND ANALYSIS FOR THE FIXED THRESHOLD SETTING

As described in the main paper the fixed threshold setting is the variant of the cost subsidy framework that imposes a reward constraint in the form of a fixed known reward threshold $\mu_0$. Below in Algorithm 6 we present a pricipled approach for regret minimization in the fixed threshold MAB-CS setting.

---

**Algorithm 6:** FIXED THRESHOLD UCB (FT-UCB)

---

**Inputs:** Bandit instance $\nu$, Fixed threshold $\mu_0$.
**Initialize:** Samples $n_k = 0$, Empirical Means $\hat{\mu}_k = 0 \ \forall k \in [K]$, Time $t = 1$ .
1 **while** $t \leq T$ **do**
2      **if** $t \leq K$ **then**
3          $k_t \leftarrow t$
4      **else**
5          $C_t \leftarrow \left\{ k : \hat{\mu}_k(t) + \sqrt{\frac{2 \log t}{n_k(t)}} \geq \mu_0 \right\}$      // Identify all arms with UCB satisfying reward threshold
6          **if** $C_t \neq \phi$ **then**
7              $k_t \leftarrow \arg\min_{i \in C_t} c_i$
8          **else**
9              $k_t \leftarrow \text{Uniform}(K)$
10      $\hat{\boldsymbol{\mu}}(t+1), \boldsymbol{n}(t+1), t \leftarrow \text{sample\_and\_update}(k_t, \hat{\boldsymbol{\mu}}(t), \boldsymbol{n}(t), t)$      // in Appendix B

---

ANALYSIS FOR FT-UCB

FT operates in the *fixed threshold* setting with $\mu_{\text{CS}} = \mu_0$, where $\mu_0 > 0, \mu_0 \in \mathbb{R}$ is a known threshold. Although the setup for the fixed threshold setting remains similar to the known reference arm and subsidized best reward settings discussed in the main paper, the key difference is in the definition of optimal arm $a^*$. In this section, optimal action $a^* = \arg\min_{i:\mu_i \geq \mu_0} c_i$. We highlight that while in the *known reference arm* and subsidized best reward settings, the structure of the problem ensured that a feasible arm always existed. In the fixed threshold setting, we assume that there is at least one feasible arm satisfying the reward constraint.

**Theorem H.1** (Instance dependent upper bound on cost and quality regret for FT-UCB). *For bandit instance $\nu$, over horizon $T$, the expected cumulative cost regret $\mathbb{E}\left[Cost\_Reg\left(T, \nu\right)\right]$ and quality regret $\mathbb{E}\left[Quality\_Reg\left(T, \nu\right)\right]$ of the FT-UCB algorithm are upper bounded respectively as,*

$$\mathbb{E}\left[Cost\_Reg\left(T, \nu\right)\right] \leq \sum_{i=a^*+1}^{K} \Delta_{C,i}^+ + \frac{\pi^2}{6} \max_{i > a^*} \Delta_{C,i}^+,$$

$$\mathbb{E}\left[Quality\_Reg\left(T, \nu\right)\right] \leq \sum_{i=1}^{a^*-1} \frac{8 \log T}{\Delta_{Q,i}^+} + \left(1 + \frac{\pi^2}{6}\right) \sum_{i=1}^{a^*-1} \Delta_{Q,i}^+ + \frac{\pi^2}{6} \max_{i \in [K]} \Delta_{Q,i}^+.$$

Along the lines of the analysis for the expected cumulative regret for PE in Appendix F, to prove Theorem H.1 we first bound the expected number of samples of sub-optimal arms and then combine the results together into a regret bound using the regret decomposition lemma.

UPPER BOUND ON THE NUMBER OF SAMPLES OF ARMS $i > a^*$

**Lemma H.1** (Upper bound on the sum of the expected number of samples of all high cost arms). *The sum of the expected number of samples of all high cost arms accrued after the initial sampling of each arm is upper bounded as,*

$$\sum_{a^*+1}^{K} \mathbb{E}\left[n_i(K+1, T)\right] \leq \frac{\pi^2}{6}.$$

*Proof.* We prove Lemma H.1 by first writing an expression for the samples of any high-cost arm. Then we upper bound the expression by the number of times the optimal arm $a^*$ was excluded from the set of empirically satisfactory arms $\mathcal{C}_t$. We can bound this way because at time $t$ an arm $i > a^*$ can only be picked if the arm $a^*$ is not inside the set $\mathcal{C}_t$.

$$\sum_{i=a^*+1}^{K} n_i(K+1,T) = \sum_{i=a^*+1}^{K}\sum_{t=K+1}^{T} \mathbf{1}\left\{k_t = i\right\} = \sum_{t=K+1}^{T} \mathbf{1}\left\{k_t \in \{a^*+1,\ldots,K\}\right\} \tag{262}$$

$$\leq \sum_{t=K+1}^{T} \mathbf{1}\left\{a^* \notin \mathcal{C}_t\right\} \tag{263}$$

$$\leq \sum_{t=K+1}^{T} \mathbf{1}\left\{\hat{\mu}_{a^*}\left(n_{a^*}(t-1)\right) + \sqrt{\frac{2\log(t-1)}{n_{a^*}(t-1)}} < \mu_0\right\} \tag{264}$$

$$= \sum_{t=K+1}^{T} \mathbf{1}\left\{\hat{\mu}_{a^*}\left(n_{a^*}(t-1)\right) - \mu_{a^*} < \mu_0 - \mu_{a^*} - \sqrt{\frac{2\log(t-1)}{n_{a^*}(t-1)}}\right\} \tag{265}$$

$$\leq \sum_{t=K+1}^{T}\sum_{s=1}^{t-1} \mathbf{1}\left\{\hat{\mu}_{a^*}(s) - \mu_{a^*} < \mu_0 - \mu_{a^*} - \sqrt{\frac{2\log(t-1)}{s}}\right\}. \tag{266}$$

Taking expectations, and perform a shift of 1 in time-indexing, we have,

$$\mathbb{E}\left[\sum_{i=a^*+1}^{K} n_i(K+1,T)\right] = \sum_{i=a^*+1}^{K} \mathbb{E}\left[n_i(K+1,T)\right] \tag{267}$$

$$\leq \sum_{t=1}^{\infty}\sum_{s=1}^{t} \Pr\left(\hat{\mu}_{a^*}(s) - \mu_{a^*} < \mu_0 - \mu_{a^*} - \sqrt{\frac{2\log t}{s}}\right) \tag{268}$$

$$\leq \sum_{t=1}^{\infty}\sum_{s=1}^{t} \exp\left(-2s\frac{2\log t}{s}\right) \quad \text{(using Hoeffding Inequality D.2)} \tag{269}$$

$$= \sum_{t=1}^{\infty}\sum_{s=1}^{t} \frac{1}{t^4} \tag{270}$$

$$\leq \frac{\pi^2}{6}. \tag{271}$$

$\square$

UPPER BOUND ON THE NUMBER OF SAMPLES OF ARMS $i < a^*$

**Lemma H.2** (Bound on the expected number of samples of low-cost arms under FT-UCB). *For any low cost infeasible arm $i < a^*$, its expected number of samples accrued is upper bounded by,*

$$\mathbb{E}[n_i(T)] \leq \frac{8\log T}{(\Delta_{Q,i})^2} + 1 + \frac{\pi^2}{6} + \mathbb{E}\left[\sum_{t=1}^{\infty} \mathbf{1}\left\{k_t = i, C_t = \phi\right\}\right].$$

*Where the last expectation term represents any samples of $i$ accrued by uniform random sampling when we find $\mathcal{C}_t$ empty.*

*Proof.* We begin by writing an expression for the random variable $n_i(T)$ denoting the number of samples of arm $i$.

$$n_i(T) = 1 + \sum_{t=K+1}^{T} \mathbf{1}\left\{k_t = i, C_t \neq \phi\right\} + \sum_{t=K+1}^{T} \mathbf{1}\left\{k_t = i, C_t = \phi\right\} \tag{272}$$

$$\leq 1 + \sum_{t=K+1}^{T} \mathbf{1}\{k_t = i, C_t \neq \phi\} + \sum_{t=K+1}^{T} \mathbf{1}\{k_t = i, C_t = \phi\} \tag{273}$$

$$= m_0 + \sum_{t=K+1}^{T} \mathbf{1}\{k_t = i, C_t \neq \phi, n_i(t-1) \geq m_0\} + \sum_{t=K+1}^{T} \mathbf{1}\{k_t = i, C_t = \phi\} \tag{274}$$

$$= m_0 + \sum_{t=K+1}^{T} \mathbf{1}\left\{\hat{\mu}_i(n_i(t-1)) + \sqrt{\frac{2\log(t-1)}{n_i(t-1)}} \geq \mu_0, n_i(t-1) \geq m_0\right\}$$

$$+ \sum_{t=K+1}^{T} \mathbf{1}\{k_t = i, C_t = \phi\} \tag{275}$$

$$\leq m_0 + \sum_{t=1}^{\infty}\sum_{s=m_0}^{t} \mathbf{1}\left\{\hat{\mu}_i(s) + \sqrt{\frac{2\log t}{s}} \geq \mu_0\right\} + \sum_{t=1}^{\infty}\mathbf{1}\{k_t = i, C_t = \phi\}. \tag{276}$$

Taking expectations on both sides, we have,

$$\mathbb{E}[n_i(T)] \leq m_0 + \sum_{t=1}^{\infty}\sum_{s_i=m_0}^{t} \Pr\left(\hat{\mu}_i(s_i) - \mu_i \geq \Delta_{Q,i} - \sqrt{\frac{2\log t}{s_i}}\right)$$

$$+ \mathbb{E}\left[\sum_{t=1}^{\infty}\mathbf{1}\{k_t = i, C_t = \phi\}\right]. \tag{277}$$

Applying Hoeffding Inequality, we have,

$$\Pr\left(\hat{\mu}_i - \mu_i \geq \Delta_{Q,i} - \sqrt{\frac{2\log t}{s_i}}\right) \leq \exp\left(-2s_i\left(\Delta_{Q,i} - \sqrt{\frac{2\log t}{s_i}}\right)^2\right) \tag{278}$$

$$\leq \exp\left(-2s_i\left((\Delta_{Q,i})^2 + \frac{2\log t}{s_i} - 2\Delta_{Q,i}\sqrt{\frac{2\log t}{s_i}}\right)\right). \tag{279}$$

Plugging in the bound in Equation 279 into Equation 277 we have,

$$\mathbb{E}[n_i(T)] \leq m_0 + \sum_{t=1}^{\infty}\sum_{s_i=m_0}^{t} \exp\left(-2s_i\left(\frac{2\log t}{s_i} + (\Delta_{Q,i})^2 - 2\Delta_{Q,i}\sqrt{\frac{2\log t}{s_i}}\right)\right) +$$

$$+ \mathbb{E}\left[\sum_{t=1}^{\infty}\mathbf{1}\{k_t = i, C_t = \phi\}\right]. \tag{280}$$

We pick $m_0 = \left\lceil\frac{8\log T}{(\Delta_{Q,i})^2}\right\rceil$ as in the proof technique in Auer et al. (2002).

$$\mathbb{E}[n_i(T)] \leq m_0 + \sum_{t=1}^{\infty}\sum_{s_i=m_0}^{t} \exp\left(-4\log t + 2s_i(\Delta_{Q,i})^2\left(\sqrt{\frac{\log t}{\log T}} - 1\right)\right) + \frac{\pi^2}{6} \tag{281}$$

Since $t < T$, we have,

$$\mathbb{E}[n_i(T)] \leq \frac{8\log T}{(\Delta_{Q,i})^2} + 1 + \sum_{t=1}^{T-1}\sum_{s_i=m_0}^{t-1} \exp\left(-4\log t\right) + \mathbb{E}\left[\sum_{t=1}^{\infty}\mathbf{1}\{k_t = i, C_t = \phi\}\right] \tag{282}$$

$$\leq \frac{8\log T}{(\Delta_{Q,i})^2} + 1 + \sum_{t=1}^{\infty}\frac{1}{t^3} + \mathbb{E}\left[\sum_{t=1}^{\infty}\mathbf{1}\{k_t = i, C_t = \phi\}\right] \tag{283}$$

$$\leq \frac{8 \log T}{(\Delta_{Q,i})^2} + 1 + \frac{\pi^2}{6} + \mathbb{E}\left[\sum_{t=1}^{\infty} \mathbf{1}\{k_t = i, C_t = \phi\}\right]. \tag{284}$$

$\square$

We are now in a position to prove Theorem H.1 by combining regret decomposition (Lemma D.5) with the results of Lemmas H.1 and H.2.

*Proof of Theorem H.1.* First for cost regret we have,

$$\text{Cost\_Reg}(T, \nu) = \sum_{i=a^*+1}^{K} \Delta_{C,i}^+ \mathbb{E}[n_i(T)] \quad (\text{because } \Delta_{C,i}^+ = 0 \,\forall\, i \leq a^*) \tag{285}$$

$$= \sum_{i=a^*+1}^{K} \Delta_{C,i}^+ \mathbb{E}[n_i(1, K) + n_i(K+1, T)] \tag{286}$$

$$= \sum_{i=a^*+1}^{K} \Delta_{C,i}^+ + \max_{i > a^*} \Delta_{C,i}^+ \sum_{i=a^*+1}^{K} \mathbb{E}[n_i(K+1, T)] \tag{287}$$

$$\leq \sum_{i=a^*+1}^{K} \Delta_{C,i}^+ + \frac{\pi^2}{6} \max_{i > a^*} \Delta_{C,i}^+ \quad (\text{plugging in Lemma H.1}). \tag{288}$$

Next for quality regret,

$$\text{Quality\_Reg}(T, \nu) = \sum_{i=1}^{a^*-1} \Delta_{Q,i}^+ \mathbb{E}[n_i(T)] + \sum_{i=a^*+1}^{K} \Delta_{Q,i}^+ \mathbb{E}[n_i(T)] \quad (\text{because } \Delta_{Q,a^*}^+ = 0) \tag{289}$$

$$\leq \sum_{i=1}^{a^*-1} \Delta_{Q,i}^+ \left(\frac{8 \log T}{(\Delta_{Q,i})^2} + 1 + \frac{\pi^2}{6} + \mathbb{E}\left[\sum_{t=1}^{\infty} \mathbf{1}\{k_t = i, C_t = \phi\}\right]\right)$$
$$+ \sum_{i=a^*+1}^{K} \Delta_{Q,i}^+ \left(\mathbb{E}[n_i(1, K)] + \mathbb{E}[n_i(K+1, T)]\right) \tag{290}$$

$$\leq \sum_{i=1}^{a^*-1} \Delta_{Q,i}^+ \left(\frac{8 \log T}{(\Delta_{Q,i})^2} + 1 + \frac{\pi^2}{6} + \mathbb{E}\left[\sum_{t=1}^{\infty} \mathbf{1}\{k_t = i, C_t = \phi\}\right]\right)$$
$$+ \sum_{i=a^*+1}^{K} \Delta_{Q,i}^+ + \max_{i > a^*} \Delta_{Q,i}^+ \sum_{t=K+1}^{T} \mathbb{E}[\mathbf{1}\{k_t \in \{a^*+1, \ldots, K\}\}]$$
$$(\text{reused from Equation 262}) \tag{291}$$

$$\leq \sum_{i=1}^{a^*-1} \frac{8 \log T}{\Delta_{Q,i}^+} + \left(1 + \frac{\pi^2}{6}\right) \sum_{i=1}^{a^*-1} \Delta_{Q,i}^+$$
$$+ \max_{i < a^*} \Delta_{Q,i}^+ \mathbb{E}\left[\sum_{t=1}^{\infty} \mathbf{1}\left\{\underbrace{k_t \in \{1, \ldots, a^*-1\}, C_t = \phi}_{\text{Clause A}}\right\}\right]$$
$$+ \sum_{i=a^*+1}^{K} \Delta_{Q,i}^+ + \max_{i > a^*} \Delta_{Q,i}^+ \sum_{t=K+1}^{T} \mathbb{E}\left[\mathbf{1}\left\{\underbrace{k_t \in \{a^*+1, \ldots, K\}}_{\text{Clause B}}\right\}\right] \tag{292}$$

$$\leq \sum_{i=1}^{a^*-1} \frac{8 \log T}{\Delta_{Q,i}^+} + \left(1 + \frac{\pi^2}{6}\right) \sum_{i=1}^{a^*-1} \Delta_{Q,i}^+$$

$$+ \max_{i \in [K]} \Delta_{Q,i}^+ \mathbb{E}\left[\sum_{t=1}^{\infty} \mathbf{1}\{\text{Clause A or Clause B}\}\right] \tag{293}$$

$$\leq \sum_{i=1}^{a^*-1} \frac{8 \log T}{\Delta_{Q,i}^+} + \left(1 + \frac{\pi^2}{6}\right) \sum_{i=1}^{a^*-1} \Delta_{Q,i}^+$$

$$+ \max_{i \in [K]} \Delta_{Q,i}^+ \mathbb{E}\left[\sum_{t=1}^{\infty} \mathbf{1}\{a^* \notin \mathcal{C}_t\}\right] \quad (\text{because } a^* \notin \mathcal{C}_t \supseteq \text{Clause A, Clause B})$$

$$\tag{294}$$

$$\leq \sum_{i=1}^{a^*-1} \frac{8 \log T}{\Delta_{Q,i}^+} + \left(1 + \frac{\pi^2}{6}\right) \sum_{i=1}^{a^*-1} \Delta_{Q,i}^+ + \frac{\pi^2}{6} \max_{i \in [K]} \Delta_{Q,i}^+. \tag{295}$$

Where Equation 295 follows from the bound on $\sum_{t=1}^{\infty} \Pr(a^* \notin \mathcal{C}_t)$ shown in the proof of Lemma H.1. $\qquad \square$

**Remark H.1** (Comparison with lower bound). *Theorem H.1 and its analysis reveal that FT-UCB is an order-optimal algorithm for the fixed threshold MAB-CS setting. The lower bound of Theorem E.6 is matched precisely by FT-UCB's $O(1)$ bound on expected cumulative cost regret and its $O(\log T)$ bound with dependence only on the quality gaps of arms $i < a^*$ for its expected cumulative quality regret.*

