# OpenReview forum: "Pairwise Elimination with Instance-Dependent Guarantees for Bandits with Cost Subsidy"
_ICLR.cc/2025/Conference — ICLR 2025 Poster_

### Official Review · Reviewer_CGLm · 2024-10-28

**Soundness:** 2
**Presentation:** 2
**Contribution:** 2
**Rating:** 5
**Confidence:** 4

**Summary:**

This paper proposes the pairwise elimination (PE) algorithm for bandits with a cost subsidy model. First, it develops a simple PE algorithm that addresses two simplified versions of the MAB-CS model with a known reference arm. Then, it extends the PE algorithm with a BAI subroutine to address the MAB-CS model without knowing the reference arm. Theoretical analysis verifies that the algorithms have instance-dependent logarithmic regret/cost upper bounds. Experiments also verify the PE-CS’s performance.

**Strengths:**

1. New theoretical results. This paper proposes the first algorithm with instance-dependent logarithmic regret/cost guarantees for the MAB-CS model.

**Weaknesses:**

1. The algorithm design and analysis of the paper are not novel, and the theoretical results from the proposed algorithm are not interesting (i.e., expected). The algorithmic technique is very similar to prior literature, and the analytical approaches are also similar to other related works. The reviewer would suggest the author look into some challenging parts of the topic, like the lower bound for the MAB-CS model and proposing near-optimal algorithms (e.g., how to revise the UCB-CS algorithm so as it also works for the MAB-CS model).
2. The writing of this paper is not easy to follow. For example, when introducing the three types of model settings (Lines 93—101), which are supposed to be mathematically rigorous (e.g., for clear math notations and definitions), the paper uses an example to explain these three settings vaguely. Another example is the inconsistency of the notations. The notation $\mu_{\text{CS}}$ was used much earlier before it was formally defined. The two gaps $\Delta_C$ and $\Delta_Q^+$ should both have a plus on the superscript, but one is defined explicitly, and the other implicitly, which is inconsistent and confusing.

**Questions:**

### Minor Comments

- For the `sample_and_update` subroutine, it would be better to give the details, possibly in the appendix, if there is no space in the main paper.
- In Line 321, the `omega` should be $\omega$.
- Why, in the experiment, the regret and cost are always reported in a summation form. Would it be interesting to plot one as the x-axis and the other as the y-axis and check whether there is a trade-off?

---

> ### Author Response · Authors · 2024-11-27
> **Response to reviewer CGLm**
>
> Response to reviewer CGLm
>
> Response Key:
> Wi: Direct response to weakness number i
>
> Qi: Direct response to question number i
>
>  #################
>
> W1
>
> We respectfully disagree with the reviewers objections about a lack of novelty and with their comment about the theoretical results being expected. To make our case for the novelty in algorithm design and analysis we would like to draw attention to specific original contributions that we believe are significant.
> 1. We introduce two new structured bandit problem settings: fixed threshold, and known reference arm within MAB-CS. While these new settings are natural extensions of the work in sinha et al. 2021, they significantly enhance the utility of MAB-CS to practitioners.
> 2. Our PE algorithm is the first approach to MAB-CS that evaluates feasibility of arms in cost order. It builds upon improved-ucb (Auer and Ortner, 2011) but has significant additional structure that is novel and original. A feature is sample re-use where the same samples of the reference arm $\ell$ are leveraged to evaluate the feasibility of candidate arms across episodes of PE. (Lines 275-282)
> 3. A practical extension to PE is asymmetric-PE (the details of which we present in Appendix C). Asymmetric PE uses a completely different arm
>  elimination criteria than any prior works. In particular it uses different exploration bonus terms $\beta_{\ell}$ and $\beta_i$ for the reference arm $\ell$ and candidate arm $i$ under evaluation. (Lines 311-319).
> 4. We provide an analysis of PE that conditions both on events at the intra and inter-episode levels. We show that the probability of improper outcomes at both the intra and the inter episode levels is small. (Lines 298-303)
> 5. We develop PE-CS by cascading BAI() and PE() and analyze it in part by characterizing the probability of our best arm identification subroutine making an error which is an original style of analyzing an algorithm. (Lines 399-405)
> 6. Through experiments we show that PE-CS significantly outperforms the only other algorithm from the MAB-CS literature that has a regret guarantee (namely ETC-CS). Moreover we argue that even including other baselines that lack any guarantee, PE-CS offers the best balance between reliability and performance (Figures 2, 3).
> 7. Lower bounds which have been included in the rebuttal revisions (see below).
>
> Lower Bounds:
> - The reviewer's suggestion about working on other challenging topics like the lower bounds is well taken. We have worked out lower bounds for the known reference arm and subsidized best reward cases and these are now stated in Theorems 3.1 and 3.3 respectively (line 216, 322). Extended discussion and L.B. proofs are available in Appendix E.
>
> - In particular we would like to highlight that PE is now shown to be an order optimal algorithm for the known reference arm setting (lines 305-310). Moreover, we find that while PE-CS is not order optimal, there is a class of bandit instances for which it is order optimal (Lines 407-410, and Remark E.7 in Appendix E).
>
> - Finally, we argue that the while the general form of the upper bounds can be considered expected, working out the details is a significant contribution and represents an important step towards characterizing the MAB-CS problem formally. We have updated the presentation of the upper bound result in Theorem 3.2 for PE and in lines 399-405 for PE-CS to better reveal the saliency.
>
> - The reviewers suggestion about proposing near-optimal algorithms is well taken. We would like them to consider that PE and PE-CS do in fact satisfy this requirement (PE-CS only for a certain problem class).
>
> - We are not sure what the reviewer's suggestion was when they suggested that work be done on revising the UCB-CS algorithm for the MAB-CS model. The UCB-CS algorithm (Sinha et al. 2021) was in fact proposed for the MAB-CS model. Moreover, through experiments in Section 4.2 we make the case that the strategy of the algorithm is flawed and has a fixed probability of permanently incorporating an infeasible arm in its empirically feasible set. We may have misunderstood the reviewer's intent with their feedback and would really appreciate a clarification.

---

> ### Author Response · Authors · 2024-11-27
>
> Response to reviewer CGLm
>
> #################
>
> W2
>
> The reviewer's feedback on issues with the writing are well received. While we have used the rebuttal period to significantly improve the presentation in many ways we highlight specific corrections made in response to W2.
>
> - The three types of model settings are now introduced more formally in lines 96-102 of the revised version
> - The notation $\mu_{\text{CS}}$ is now defined before its first use in line 66-67.
> - We now have made both cost and quality gaps explicitly zero-clipped, and have applied this change through the paper. (Lines 75, 78, 283 (Thm 3.2 ), 377 (Thm 3.4)). We thank the reviewer for this suggestion as it improves readability.
>
> #################
>
> - Q1: We apologize for the oversight in not including `sample_and_update()`. It is now included in Appendix B and a reference as such has been provided as an inline comment in blocks Algorithm 1 and Algorithm 2. (Lines 263, 358)
>
> - Q2: Corrections to change text omega to symbol ${\omega}$ made
>
> - Q3: The reviewers feedback on the regret metrics being summed together and not plotted separately is well taken. The reason we chose to do this is that we did not find the differences in the trade off's made by different algorithms (we test 4 for the subsidized best reward setting) to be particularly insightful. However, please see our post tomorrow when we shall share a revised Experiments section with new results addressing this and other experiments requested by reviewer fMRh.

---

> > ### Author Response · Authors · 2024-11-28
> > **Response to reviewer CGLm**
> >
> > Response to reviewer CGLm
> >
> > #################
> >
> > Q3 (Follow Up)
> >
> > To perform the cost vs. quality regret trade off visualization we use the same data as our main (new, post rebuttal) experiments.
> >
> > - We have revised the presentation of the experiments to be simulations that vary parameters $\ell$, $\alpha$ for the known reference arm setting and parameter $\alpha$ for the subsidized best reward setting. These experiments visualize the terminal regret on a scatter plot with the value of the varied hyper-parameter on the x-axis. Plots are available in Section 4.
> >
> > - We use the same terminal regret data as the above mentioned parameter varying experiments and plot the trade off between cost and quality regret by plotting cost and quality on the y and x axes respectively. Since the insights from these visualizations are tangential to the aim of Section 4 (which validating PE and PE-CS performance), they are available as Figures 5 and 6 under Supplemental Experiments in Appendix A.
> >
> > #################
> >
> > Figures 5 and 6,
> >
> > Comments on Appendix Figures 5 and 6 (Figure captions incomplete in submitted version due to oversight, however from the point of view of understanding trade-offs the source of the experiments is not important)
> >
> > While trade off result can be looked at from many angles, the one we choose to highlight is that our method PE-CS in the subsidized best reward setting. PE-CS never falls into the trap of a linear regret trend obtained from over sampling a sub optimal arm $i \neq a^*$. Linear regret trends can be caught on the terminal regret plot based on the terminal values being orders of magnitude larger than the other trend lines.
> >
> > #################
> >
> > - We would greatly appreciate further discussion and questions from the reviewer and would like them to consider increasing their score if our responses have already addressed their concerns.

---

> > > ### Author Response · Authors · 2024-12-01
> > > **Response to reviewer CGLm**
> > >
> > > ## Follow up response to reviewer CGLm
> > >
> > > ---
> > >
> > > ### Q3 (Further follow up)
> > >
> > > As we get close to the discussion end deadline we wanted to share a follow up response to review CGLm in light of concerns raised by new reviewer wsgZ.
> > >
> > > Reviewer wsgZ questioned why the metric in experiments Section 4 was summed cost and quality regret and why trade-offs were not considered. This is closely related to reviewer CGLm's point in Q3 and so we wanted to bring our response to reviewer wsgZ to their notice.
> > >
> > > We responded in [response (2/3) here](https://openreview.net/forum?id=eB7T1bqthA&noteId=iak6ZjhWYa).
> > >
> > > ---

---

> > > > ### Author Response · Authors · 2024-12-02
> > > > **Response requested**
> > > >
> > > > ### Response requested from reviewer CGLm
> > > >
> > > > ---
> > > >
> > > > As there was only about one day left to the deadline for reviewer comments the authors wanted to request a response from reviewer CGLm.
> > > >
> > > > We believe that we made a very convincing case for the strengths of our work and responded to the reviewers concerns thoroughly. As a result we were eager to hear back and wanted to ensure that the reviewer got a chance to consider a score increase from their initial assessment.
> > > >
> > > > ---

---

> > > > > ### Comment · Reviewer_CGLm · 2024-12-02
> > > > >
> > > > > The reviewer appreciates the authors' comprehensive responses. After carefully reading the rebuttal and the revised submission, the reviewer thinks:
> > > > > - For the novelty and contribution part, the reviewer still believes the current algorithm design and upper/lower bound analysis are not novel enough.
> > > > > - For the lower bound and optimality part, the reviewer is happy to know that PE is near-optimal in the simplified case (with reference arm) but not in the general case. Further optimal algorithm design for the general case would be an exciting investigation direction.
> > > > > - For the additional trade-off experiments, the reviewer suggests varying some parameters to see a clearer trade-off of the proposed algorithm in the next version of this paper. For example, by varying the parameter $\alpha$, how the performance of the algorithm would change.
> > > > >
> > > > > Given the above understanding, the reviewer will maintain their slightly negative evaluation.

---

> ### Author Response · Authors · 2024-12-04
> **Response to reviewer CGLm**
>
> ### Response to reviewer CGLm (1/2)
>
> ---
>
> The authors thank the reviewer for reviewing our rebuttal response and for their suggestions for directions of future work. We would like to respond to them with the following.
>
> First, to counter the reviewer's assessment of novelty we provide an intuitive explanation for why our algorithms and results are novel, interesting, and even unexpected. The arguments there assume we have an agreement on the utility of the model. In case that is not the case we would request that the reviewer see our examples in the [response linked here](https://openreview.net/forum?id=eB7T1bqthA&noteId=oAfGH9H9nJ).
> Second, we directly address the point on trade-offs and new experiments raised by the reviewer. This part has overlap with the responses to other reviewers [response W4 here](https://openreview.net/forum?id=eB7T1bqthA&noteId=fUQDD4yrLs) and [response (2/3) for W3 here](https://openreview.net/forum?id=eB7T1bqthA&noteId=iak6ZjhWYa).
>
> —
>
> ### Novelty
>
> In our response to the original review we listed out the original contributions of our work. Here we provide a narrative explanation of the core novelty of the work.
> Our lower bound results (Section 3, Thm 3.1 and 3.3) reveal that only a reduced number of samples are needed from *high cost arms* in both the known reference arm and subsidized best reward settings. This reality, which is an original contribution in itself makes it clear that an order-optimal algorithm for these settings must consider (and eliminate) candidate arms one at a time in an episodic manner. This also means that a traditional UCB-like approach (UCB-CS in Sinha et al. for example) in which all arms are always (through $T$ samples) in contention for sampling cannot be order optimal. As the reviewer recognizes, PE (using elimination) is order optimal for the known reference arm setting. We highlight that PE-CS is order optimal in its regret contribution from low cost arms (for all instances) and for the best reward arm for some instances that we characterize.
>
> Moreover, as stated in Theorem 3.3 the precise lower bound on the expected number of samples for the best reward arm $i^*$ under PE **depends on the expected reward of 4 distinct arms**. Namely the optimal action $a^*$, the best reward cheap arm $\arg \max_{i < a^*} \mu_i$, the best reward arm $i^*$, and the overall second best reward arm $\arg \max_{i \neq i^*} \mu_i$. This is a result that is unexpected and cannot be seen without carefully working out the details like we do in the proof of Theorem 3.3, Remark E.5 in Appendix E.
>
> - - -

---

> ### Author Response · Authors · 2024-12-04
>
> ### Response to reviewer CGLm (2/2)
>
> - - -
>
> ### Trade Offs
> The reviewer pointed out in their initial assessment that discussion and experimental results around trade-offs between cost and quality regret were lacking. We however highlight that from the point of view of the optimal policy on MAB-CS **there is no tradeoff** between cost and quality regret. This is because optimal policy sampled arm $a^*$ as often as possible and arm $a^*$ has zero incremental cost and quality regret.
>
> The zero-clipping in regret definitions ensures that samples from 'an arm with high quality and low cost regret' can not be traded off (in terms of its informativeness about $a^*$) with samples from 'an arm with low quality and high cost regret'. This is why in the experiments we plot the summed together values of cost and quality regret on the y-axes to gauge how close each method is to the optimal policy.
>
> We acknowledge that this might not have been clear from the writing in the paper and the initial rebuttal response. We realized that this point had to be made clear while responding to reviewer wsgZ in [response (2/3)](https://openreview.net/forum?id=eB7T1bqthA&noteId=iak6ZjhWYa).
>
> We do in fact perform an experiment where we vary $\alpha$ for both the known reference arm and subsidized best reward setting like the reviewer asks (Figures 1 and 2 in Section 4 and associated observations in Sections 4.1 and 4.2).
>
> It is unclear in their most recent comment: "For the additional trade-off experiments … performance of the algorithm would change" was referring to performance as we interpret it (summed cost and quality regret as explained above) (called (a)) or as the trade-off between cost (y-axis) and quality regret (x-axis) as they referred to in their official review (called (b)). We address both:
>
> Fig. 1 (c), (d), Fig. 2 (a), (b) plot the terminal summed regret of all algorithms for different values of $\alpha$ for the same bandit instance. We chose to use the plot to discuss how our methods (PE / PE-CS) do better overall across all values of $\alpha$ ; however the plots could also be used to gauge the effect of $\alpha$ on regret performance.
> To see if the various methods systematically make choices that favor one of either cost or quality regret we plot the entire data from this experiment (i.e. across 25-50 runs each of for a range of values of $\alpha$) in Figures 5 and 6 (bottom row only in Fig. 6, top row of Fig. 6 represents data from varying $\ell$) as a scatter plot. This however is tangential to the goal of gauging performance and so did not make it to the main paper and is in Appendix A.
>
> ---

---

### Official Review · Reviewer_MwCh · 2024-11-01

**Soundness:** 3
**Presentation:** 3
**Contribution:** 3
**Rating:** 6
**Confidence:** 2

**Summary:**

This work studies the Multi-Armed Bandits with Cost Subsidy (MAB-CS) problem, where a primary metric (cost) is constrained by a secondary metric (reward), and there is an inability to explicitly determine the trade-off between these metrics. The authors introduce the Pairwise-Elimination (PE) algorithm for a simplified variant of the cost-subsidy problem with a known reference arm, and generalize PE to PE-CS to solve the MAB-CS problem in the setting where the reference arm is the unidentified optimal arm. The authors provide instance-dependent bounds for PE and PE-CS for both Cost Regret and Quality Regret. They also conduct experiments to support the theoretical claims.

**Strengths:**

1. The problem is well-motivated. The authors provide interesting examples of applications of the MAB-CS framework.
2. The paper is well-written.
3. This work extends the MAB-CS framework to include two new settings, and develops two novel algorithms PE and PE-CS . The authors also provide instance-dependent bounds for the proposed algorithms.
4. The authors conduct experiments on real-world data to support the theoretical claims.

**Weaknesses:**

I feel some statements are somehow overclaimed. In Lines 115-126, the authors claim that the regret bounds of their proposed algorithms are $O(\log T)$, while for ETC-CS it is $O(T^{2/3})$. However, their bounds are instance-dependent and are not $O(\log T)$ in the worst case, while the  $O(T^{2/3})$ is the worst-case bound. Therefore, such a comparison seems to be over-claimed.

**Questions:**

1. See the weakness.
2. Is it possible to provide some lower bounds for this problem?

---

> ### Author Response · Authors · 2024-11-27
> **Response to reviewer MwCh**
>
> Response to reviewer MwCh
>
> Response Key:
> Wi: Direct response to weakness number i
>
> Qi: Direct response to question number i
>
>  #################
>
> W1
>
> - We thank the reviewer for highlighting the over claimed comparison between the worst case regret bound of ETC-CS and our instance dependent bounds. We have deleted the direct comparison from the revised Section 1.2 Key Contributions. The only reference to the different regret upper bounds now is in Section 4 and is a more nuanced comparison (Lines 513-521).
>
> #################
>
> Q1 = W1
>
> Q2
>
> - We have worked out lower bounds for the known reference arm and subsidized best reward cases and these are now stated in Theorems 3.1 and 3.3 respectively (line 216, 322). Extended discussion and L.B. proofs are available in Appendix E.
>
> - In particular we would like to highlight that PE is now shown to be an order optimal algorithm for the known reference arm setting (lines 305-310). Moreover, we find that while PE-CS is not order optimal, there is a class of bandit instances for which it is order optimal (Lines 407-410, and Remark E.7 in Appendix E).
>
> - Overall we believe that with our novel instance dependent lower bounds our rebuttal revised paper is significantly stronger than our original submission.

---

> > ### Comment · Reviewer_MwCh · 2024-11-27
> >
> > Thanks for your response, I will keep my original score.

---

### Official Review · Reviewer_fMRh · 2024-11-03

**Soundness:** 3
**Presentation:** 3
**Contribution:** 3
**Rating:** 6
**Confidence:** 2

**Summary:**

This paper considers multi-armed bandits. Instead of the traditional problem of maximizing total reward, they consider the problem of minimizing cost subject to a reward constraint. This formulation has been recently proposed in SSKA '21. In this paper, the authors extend the formulation to two settings:

1. Known threshold where there is a fixed reward threshold that should be met in expectation.

2. A reference threshold to an (initially) unknown arm that should be met in expectation.

For both problems, they consider A) the cost regret which is the zero-clipped total expected cost of the algorithm relative to the lowest-cost arm that meets the reward constraint and B) the quality regret which is the zero-clipped difference in expected reward of the algorithm relative to the reward threshold.

They propose a pairwise elimination algorithm that compares arms in a pairwise manner while recording the history of arms to improve efficiency. For the unknown reference threshold setting, they add an exploration phase where they learn they try to learn the reward of the unknown arm. In both settings, they show logarithmic bounds on the cost and quality regret in terms of the number of steps $T$ and the maximum instance-wise quality and regret differences.

They try these algorithms on a few datasets and find that it performs better than other algorithms (which, to be fair, were not designed for this setting).

**Strengths:**

* They consider new settings which seem reasonable to study.

* They propose new algorithms.

* They show bounds on the performance of their algorithms.

* They evaluate their algorithms in practice.

**Weaknesses:**

* Without reading the appendix, it's unclear to me what tools they use in the analysis of their algorithms.

* The statement of the bounds is difficult to parse (classic ML with too many terms which are hard to interpret).

* The algorithms are similarly difficult to understand. For example, I don't see how the history is recorded to "intelligently re-use samples for downstream comparisons".

* The algorithms are only tested on the movielens dataset and a toy dataset under specific hyperparameter settings. I would prefer at least three datasets and plots of performance for each hyperparameter ($\ell$, $\mu_\ell$, and $\mu_{CS}$) to validate that the performance is consistent in different settings.

**Questions:**

* What tools do you use in the analysis of your algorithms? What is the general strategy?

* How do you re-use samples for downstream comparisons in your algorithms?

* Can you make the case for the zero-clipped comparison again? I wasn't persuaded by the (repetitive) paragraph about stellar performance for different ad products in section 1.1. In addition, does your analysis require the zero-clipped comparison?

With additional experiments on other datasets and settings, and satisfactory answers to the above questions, I would be happy to increase my score to at a 6.

---

> ### Author Response · Authors · 2024-11-23
> **Clarification requested from reviewer fMRh**
>
> As we are working on the fresh experiments on new datasets and more detailed experiments from varying hyper-parameters the authors would like to request a clarification from reviewer fMRh on one of their questions.
>
> The reviewer says under W4 (Weakness bullet 4) that they would prefer among other things:
> "at least three datasets and plots of performance for each hyperparameter $\ell, \mu_{\ell}$, and $\mu_{\text{CS}}$ to validate that the performance is consistent in different settings"
> This point is well received and these results have already been worked out for the MovieLens dataset we already used and are still being tested on the additional new datasets. However it is not clear to us what they mean by varying $\mu_{\ell}$ since \mu_{\ell} is uniquely determined by the choice of arm $\ell$ and the dataset (from which we infer explicitly or implicitly the values of the expected return of each arm).
>
> A clarification would be greatly appreciated in this regard.

---

> > ### Comment · Reviewer_fMRh · 2024-11-23
> > **Response to Clarification**
> >
> > Good point! I had not realized that $mu_\ell$ is uniquely specified given the arm $\ell$ and the dataset.
> >
> > I would like to see the following experiments (and I believe this would objectively strengthen the paper):
> > * The constraint $\mu_{\textnormal{CS}}$ is varied (it probably makes the most sense to vary it as a fraction
> > * There are multiple arms that are considered in aggregate.
> >
> > Please let me know if I have an additional misunderstanding of your problem. Generally, I would just like to see experiments that explore algorithmic performance in more general settings.

---

> > > ### Author Response · Authors · 2024-11-23
> > >
> > > Thanks to reviewer fMRh for clarifying the type of experiments.
> > >
> > > So far we have promising new results (yet to be incorporated into the paper draft) that:
> > > 1. For the Full cost subsidy setting (For which we propose PE-CS, Section 4.2) varies the choice of subsidy factor $\alpha$ to modulate $\mu_{\text{CS}  = (1 - \alpha) \mu^*$. Here $\mu^*$ is uniquely determined from the dataset.
> > > 2. For the known reference arm $\ell$ setting, we vary the choice of reference arm among the many arms of the bandit instance (bandit instance either inferred or explicitly formulated using dataset).
> > >
> > > We believe (2) is what reviewer fMRh is referring to an experiment where: "multiple arms are considered in aggregate.". But please correct us if not.
> > >
> > > Another new style of experiment is one in which cost regret is plotted against quality regret for each algorithm to understand the relative trade-offs (mentioned by reviewer CGLm).

---

> ### Author Response · Authors · 2024-11-27
> **Response to reviewer fMRh**
>
> Response to reviewer fMRh
>
> #################
>
> Response Key:
>
> Wi: Direct response to weakness number i
>
> Qi: Direct response to question number i
>
>  #################
>
> Q1: We agree with the reviewer that an explanation around analysis techniques was lacking in our original submission. We have made an effort to amend this at several places in the rebuttal revised version of the paper. We highlight the line numbers in the revised paper where changes were made and share a version of the same content here for convenience.
>
> The general strategy for both PE and PE-CS regret upper bound proofs is to bound the expected number of samples of every arm type (low cost, high cost, or reference) by conditioning the expected number of samples on a nominal (good) event while simultaneously bounding the probability of the good event not happening by a small quantity.
>
> - For analysis of PE in the known reference arm setting - (Theorem 3.2, Lines 283-310)
>
> There are two dimensions to the execution of PE being nominal (good). First within an episode of PE the worse quality arm should be eliminated after a reasonable amount of sampling. Second is that execution across PE episodes should terminate in episode $a^*$
>
> We separately bound the expected number of samples of arms with cost lower than the optimal action $a^*$, higher than the optimal action $a^*$, and the reference arm $\ell$. To bound these samples we condition on the desirable and anticipated events at both the intra and inter episode levels and show that the probability of these desirable events not occurring is small.
>
> Finally in the regret upper bound Theorem 3.2 statement, we have also added underbraces to explain the origin of each regret contribution term.
>
> - For analysis of PE-CS in the subsidized best reward setting - (Theorem 3.4, Lines 377-406)
>
> We bound the expected number of samples of any arm under PE-CS by first conditioning on the outcome of the BAI-stage being either proper (meaning $\ell = i^*$ identified correctly) or improper (meaning $\ell \neq i^*$ and there was a mistake in BAI). The phased nature of PE-CS admits a modular analysis where conditioned on the outcome of BAI being proper, the bounds on samples shown in Theorem 3.2 for PE hold with slight modifications. In particular the three leading terms in the PE-CS bounds (referring to the bound expression in lines 380-395) correspond directly (in order) to the three terms in the PE bounds for cost and quality regret (referring to Theorem 3.2 bound expression in lines 286-295). The fourth term in both the PE-CS cost and quality regret bounds corresponds to samples of high cost arms accrued during the BAI stage and the fifth term is the constant contribution to regret from an improper BAI outcome.
>
>  #################
>
> Q2: Sample re-usage occurs in two contexts: Vanilla PE (our focus) and asymmetric-PE (our practical extension to PE). We have updated the draft to improve the explanation of sample re-usage. This explanation is now at lines 275-279 for sample re-usage in vanilla PE and in lines 311-319 for asymmetric-PE.
>
> - PE computes a sample prescription $\tau$ using the current round number $\omega_i$ of episode $i$. During episode $i$ we are sampling arm $i$ and (possibly) arm $\ell$. Once the sample prescription $\tau$ is met for both arm $i$ and arm $\ell$, we check for elimination. Since no samples are ever discarded, episodes that are further downstream will re-use samples of arm $\ell$ from prior episodes when $n_\ell > \tau$ and forgo any additional sampling.
>
> - In PE as presented in Algorithm 1, during an arbitrary episode evaluating the candidacy of arm $i$, the reference arm $\ell$ shall only ever have to be sampled if the samples of arm $i$ start to exceed the samples of reference arm $\ell$ that were already available. In practice, we implement another version of PE called asymmetric-PE. Asymmetric-PE allows for a mismatch between the number of samples of arms $i$ and $\ell$ that go into computing the exploration bonus terms $\beta$. Reusing all the samples of arm $\ell$ that have been amassed in this manner (from episodes leading up to the current episode) leads to earlier elimination decisions (Details in Appendix C).

---

> ### Author Response · Authors · 2024-11-27
> **Response to reviewer fMRh**
>
> Response to reviewer fMRh
>
> #################
>
> Q3
>
> We appreciate the reviewer pointing out a lack of clarity in the need for zero-clipping. To address this we have re-written the corresponding paragraph in Section 1.
>
> - Revised motivation for zero-clipping in Section 1, Introduction, lines 103-114:
>
> With the MAB-CS framework, we target applications that are agnostic to the level of quality as long as the quality exceeds threshold $\mu_{\text{CS}}$. This structure necessitates the zero-clipped operation inside the cost and quality regret definitions. Consider for example a problem where it is known that customers need to be provided a certain (possibly unknown) service quality level for them to continue their subscription. Any quality on top of the feasibility level $\mu_{\text{CS}}$ would not improve the performance of our solution. For the marketing communication example from earlier, the quality threshold represents a sales conversation rate beyond which the profitability of the campaign is ensured. In this case too we would like decisions that are agnostic to sales success over $\mu_{\text{CS}}$.
>
> In the absence of the zero-clipped structure, cost and quality regret that are sub-linear in horizon $T$ could be achieved for our examples in an unintended fashion. An algorithm that samples sub-optimal arms $i \neq a^*$ a linear fraction of times but balances positive regret from infeasible decisions, by negative regret from stellar decisions would satisfy the un-clipped quality constraint but would be unsuitable for our example applications.
>
> - Is the zero-clipped comparison required for the analysis?
>
> While the zero-clipped gaps/comparison are not required for the analysis, they are closely tied to the analysis. The zero-clipping (or lack thereof) does not change the upper or lower bounds on the expected number of samples of an arm (arm of any category: low cost, high cost, reference). This is because the number of samples needed to make an elimination decision with high probability of accuracy is a function of the gap between their rewards regardless of the sense of the gap (i.e. regardless of which arm is higher quality). However the zero-clipped comparison does effect the the final regret expression as can be seen through lines 74-78.
>
> #################
>
> W1 = Q1
>
> W2: We appreciate the feedback on the bound expressions being hard to parse. To address this weakness we have made the following changes:
>
> 1. For the PE upper bound (Thm 3.2, Line 283) we have added underbraces to every term in the bounds explaining the source of the term from the point of view of the analysis. This improves readability in that the $ \sum_i E [ n_i ] \Delta_i $ form of expression becomes more apparent. Particularly the fact that some terms stem from "nominal" execution and others from an improper run.
> 2. Right after the PE upper bound statement in Thm 3.2 (Line 298) we have added a new explanation of the terms in the upper bound. In particular we have mentioned how the terms correspond to low cost arms, high cost arms, and the reference arm $\ell$.
> 3. For the PE-CS upper bound right after the bound expressions in Thm 3.4 starting line 399 we have mentioned that the first three terms correspond precisely to the three terms in the PE bound.
> 4. Right after the PE-CS bound, we have explained the origin of the fourth and fifth terms there too (line 404) and connected them to the proof technique of conditioning on the proper outcome of the BAI stage.
> 5. We have reduced the number of unique symbols in the regret expression, and have defined every symbol used (outside of ones that have already been defined and used repeatedly) at the end of the Theorem statement.
>
> W3 = Q2
>
> W4: The requested experiments are underway but have not been finished yet. Please expect another PDF revision and accompanying comment tomorrow addressing W4 on experiments.
>
> #################
>
> Lower Bounds
>
> - To conclude we would like to highlight that with our novel instance dependent lower bounds (Thm 3.1 for known reference arm setting and Thm 3.3 for subsidized best reward setting) the rebuttal revised paper is significantly stronger than our original submission.

---

> ### Author Response · Authors · 2024-11-28
> **Response to reviewer fMRh**
>
> Response to reviewer fMRh
>
> #################
>
> W4: In our original submission there were two experiments on the MovieLens dataset separately for the known reference arm (setting 1) and subsidized best reward (setting 2) settings. These experiments were limited in that they worked with a single arbitrarily chosen reference arm $\ell$ for setting 1 and a single value of the subsidy factor $\alpha$ for setting 2.
>
> In the most recent rebuttal revised version of the paper we have included more extensive experiments. We discuss the details of these updates below. Now there are three new and more extensive experiments performed on two datasets MovieLens and Goodreads (new). We thank the reviewer for inspiring these new experiments with their review.
>
> - The first new experiment is for the known reference arm setting and varies the reference arm $\ell$ while keeping $\alpha = 0$ fixed. It looks at the terminal sum cost and quality regret for each of these choices of $\ell$ at the end of T = 5 million samples. The result is a scatter plot showing the distribution across 25 independent runs of PE and competitor UCB-CS. Available in Fig. 1(a), (b).
>
> - The second new experiment is also for the known reference arm setting. It keeps $\ell$ fixed at an arbitrarily chosen arm from the bandit instances associated with the datasets and varies $\alpha$ between 0 and 0.45 in intervals of 0.05 (so 10 unique $\mu_{\text{CS}}$ values). This experiment too looks at the 5 million sample terminal regret across 25 independent runs for PE and UCB-CS. Available in Fig. 1(c), (d)
>
> - The third new experiment is for the subsidized best reward setting where we vary the only available lever of the subsidy factor $\alpha$ between 0.05 and 0.45 at intervals of 0.05 (9 $\mu_{\text{CS}}$ values). Similar to the first two experiments we scatter plot terminal regret for 50 independent experiments and compare with ETC-CS, UCB-CS, TS-CS baselines from the literature (Fig. 2(a), (b)).
>
> #################
>
> **Added in Edit:**
>
> We get the following key takeaways about performance of algorithms from these experiments (Figures 1 and 2 of Section 4).
>
> -  PE beats UCB-CS (modified with $\ell$) most of the times outside of cases when $\alpha > 0.3$ (so last three values of $\alpha$). We see this both in MovieLens and Goodreads.
>
> - PE-CS always beats ETC-CS (exception being a single Goodreads experiment with $\alpha=0.05$ where ETC-CS is slightly better). Large $\alpha$ values do favor TS-CS and UCB-CS to the point where PE-CS is not comparable to them.
>
> #################
>
> - We would also like to take the opportunity to clarify one of the reviewer's observations in their reviewer paper summary. They said that the baselines "were not designed for this setting". This is not true since the ETC-CS, UCB-CS, and TS-CS algorithms which we use as baselines against PE-CS were introduced in Sinha et al. 2021 precisely for subsidized best reward setting (which Sinha et al simply call MAB-CS). It is true however that we modify UCB-CS to work as a baseline in our novel fixed reference arm setting so the reviewer's observation is correct for the known reference arm setting since without the modification UCB-CS cannot be used in the setting.
>
> - In addition to these three kinds of new experiments on the two datasets, we retain the toy experiment we had designed to highlight the issues with the baselines that lack guarantees namely UCB-CS and TS-CS. However, in the revised paper this has been moved to Figure 8, Appendix A.
>
> - Another visualization of the data from the aforementioned three new experiments is the trade off between cost and quality regret by plotting them on the y and x axis respectively. Since the insights from these visualizations are tangential to the aim of Section 4, they were put in Appendix A.
>
> - We realize that one of the reviewer's conditions for increasing their score was experiments on three datasets. However between the several kinds of revisions needed there was not enough time to add a 3rd dataset and its associated set of three experiments. We can however add the experiments we did not get a chance to complete during the rebuttal in the camera ready version if ICLR policy allows.
>
> - Moreover, we highlight that lower bounds for setting 1 and setting 2 are now included in Section 3, Thm 3.1 and 3.3. Since the regret upper bounds for PE match the lower bound developed in Theorem 3.1 up to constant factors, PE is shown to be order optimal. This reduces the reliance on experiments for validating PE's performance. In light of this, we would ask that the reviewer consider going through with increasing their score to 6.

---

> > ### Comment · Reviewer_fMRh · 2024-11-30
> >
> > Thank you to the authors for a thorough response and additional experiments. I will increase my score to a 6.

---

### Official Review · Reviewer_wsgZ · 2024-11-30

**Soundness:** 3
**Presentation:** 3
**Contribution:** 3
**Rating:** 6
**Confidence:** 3

**Summary:**

This work studied the bandit problem with cost subsidy. It proposed the PE-CS algorithm and derived an upper bound on its cost regret and quality regret. The paper provided a detailed description on the design of the algorithm. It also evaluated its performance with numerical simulations.



=========

After rebuttal: score increased to 6.

**Strengths:**

1. The paper is overall easy to follow.
1. The paper reviewed the various related works.
1. Algorithms are evaluated with both toy data set and real-life MovieLens data set.

**Weaknesses:**

1. As the CS model is built on Sinha et al. (2021), I would suggest the author(s) to compare the theoretical results with Sinha et al. (2021) after stating the theorems.
1. There are various related works with slightly different models, are the algorithms and their performance comparable? Can those algorithms work under this setting and what are their performance?
1. Considering the plots in Section 4, the target of the algorithm seems to be minimizing the sum of the cost regret and the reward regret. Is that true? I think the target of an algorithm in this problem should be clarified. Besides, is there a trade off between minimizing the cost regret and the reward regret? Why are we interested in the sum of the two regrets?
1. As there are already a number of similar models, what is the motivation to consider this model?

Minor comment:
1. Line 168: Should there be '' We build up our approach to optimize sth. for the regret objectives ..."?

**Questions:**

Please see the *Weaknesses* section.

---

> ### Author Response · Authors · 2024-12-01
> **Response to reviewer wsgZ**
>
> ## Response (1/3) to reviewer wsgZ
>
> Response Key:
> Wi: Direct response to weakness number i
>
> ---
>
> We thank the reviewer for their insightful review, and would like to share our rebuttal response promptly as the discussion closure deadline is almost here.
>
> At the outset we would like to highlight that the paper has been updated and the locations in the paper (line numbers, theorem's, and figure numbers) we refer to in this response are those in the revised version.
>
> ### W1
>
> - While setting three in our work (a.k.a. the *subsidized best reward* setting) is indeed identical to the cost subsidy setting introduced in Sinha et al. 2021, in our work we prove instance-dependent guarantees (lower and upper regret bounds), while Sinha et al. worked on worst-case guarantees. Thus a direct comparison between the bounds is not suitable.
>
> - While we do not make direct comparisons between the bounds (for the above stated reasons), in Section 4, lines 517-519, and in lines 524-526 we talk about why and how the empirical performance of ETC-CS (The only algorithm with a guarantee in Sinha et al. 2021) compares to that of PE-CS (our approach).
>
> ### W2
>
> We had given considerable thought to adapting the approaches from other related works as baselines, a discussion of which never made it to the paper therefore we think this is a great question!
>
> The related works to ours in the Cost-Subsidy framework can be classified broadly into (a) *Bandits with constraints*, (b) *Multi-objective multi-armed bandits* (MO-MAB), and (c) *Bandits with costs*. We separately address why approaches from none of these families worked out as baselines.
>
> (a) **Bandits with constraints:** This class of work accounts for some kind of a constraint on reward in decision making. An example would be *Conservative Bandits* (Wu et al. 2016). The problem with adapting any of these methods into MAB-CS is the absence of any considerations for the primary metric of cost in their design. While costs can be shoehorned into these methods, their cost-augmented variants when tested on MAB-CS instances would in general incur linear regret.
>
> (b) **MO-MAB:** While the MAB-CS line of work (ours and Sinha et al. 2021) do fall into this category of multiple objective multi-armed bandits, the other methods in this class can be thought of as solving the “Pareto bandits” problem, for example the paper “Pareto Regret Analyses in Multi-objective Multi-armed Bandit” (Xu et al. 2023). These methods work quite generally with multiple objectives by working in the world of “reward vectors”. While useful for many applications, these have no built-in notion of an explicit quality constraint like that imposed by MAB-CS. This unfairly sets them up to incur linear regret on MAB-CS. We discuss a related point in our response W4.
>
> (c)  **Bandits with costs:** We are particularly thinking of the bandits with knapsacks (BwK) line of work (Badanidiyuru et al., 2018). While bandits with knapsacks is similar to MAB-CS in that it assumes there is a known cost of sampling each arm, the critical difference is that BwK imposes strict cost budget constraints, and the objective of the algorithms is to maximize reward subject to these cost constraints. MAB-CS instead imposes constraints under expectation on the apriori unknown rewards. Moreover MAB-CS does not impose and strict budget constraints on cost/quality. So there was no way to augment BwK to work in the MAB-CS without proposing entirely novel algorithms.
>
> Any comparison of the methods categorized (a), (b), and (c) to our PE / PE-CS would have unreasonably favored our approach and hence we decided to not use them as baselines in our simulations.
>
> ---

---

> ### Author Response · Authors · 2024-12-01
> **Response to reviewer wsgZ (2/3)**
>
> ## Response (2/3) to reviewer wsgZ
>
> ---
>
> ### W3
>
> We appreciate the reviewer raising this point and acknowledge that this had not been made clear in our original writing. In fact another reviewer (reviewer CGLm) pointed this out as well.
>
> - The objective of the cost-subsidy framework is to solve decision making problems where a primary metric (cost) is constrained by a secondary metric (reward) (Abstract lines 17-20).
> Looking at the cost or quality regret metrics individually is not meaningful. Since the cost of sampling each arm is known, a bad policy could trivially make cost regret zero by sampling cheap arms regardless of quality. In reality there is no trade-off between cost and quality regret, and the underlying goal is to converge onto sampling *optimal arm* $a^*$ as sampling $a^*$ incurs neither cost nor quality regret.
>
> - The reason there is no trade-off is due to the zero-clipping in the regret definitions. The zero-clipping ensures that samples from 'an arm with high quality and low cost regret' can not be traded off (in terms of its informativeness about $a^*$) with samples from 'an arm with low quality and high cost regret'.
>
> - This is why in the experiments we plot the summed together values of cost and quality regret on the y-axes. We implicitly aim to minimize the sum as it is the natural objective. We would be happy to make this point clearer in the final version of our paper.
>
> - Lastly, we would like to highlight that as of the rebuttal revised paper we now have **instance dependent lower bounds** on the number of pulls of sub-optimal arms for both the known reference arm and subsidized best reward settings. From our lower bounds (Section 3, Theorem 3.1 and 3.3) we prove that among a class of consistent policies PE is the policy that samples all sub-optimal arms an order-wise optimal number of times. Meanwhile, PE-CS achieves this goal for some sub-optimal arms. Where consistent policies represent the policy class that does not incur linear summed regret on any bandit instance.
>
> ---

---

> > ### Author Response · Authors · 2024-12-01
> > **Response to reviewer wsgZ (3/3)**
> >
> > ## Response (3/3) to reviewer wsgZ
> >
> > ---
> >
> > ### W4
> >
> > In response to W4, we highlight two applications from our paper, a third example that compares with another paper currently under review at ICLR, and a fourth and final example from the healthcare domain.
> >
> > 1. First, we take the example of a marketing agency where the bandit arms are the various communication modalities available to the company. The cost of deploying any modality is known and the sales conversion rate (reward) is uncertain. A natural problem that MAB-CS can solve for the company is to minimize costs while maintaining a prescribed sales rate. Section 1, lines 49-56.
> >
> > 2. Second, the problem of subscription service customer retention. Naturally, there shall be a certain quality of service threshold level above which we can retain customers and below which we shall lose customers and the business fails. The objective then becomes to minimize operating costs while maintaining this quality level in the aggregate. Since the metric of customer retention is agnostic to service quality above the threshold $\mu_\text{CS}$, MAB-CS captures this setting perfectly. Section 1.1, lines 105-110.
> >
> > 3. Our third example is from another paper currently under review at ICLR 2025 titled ["Provably Efficient Multi-Objective Bandit Algorithms under Preference-Centric Customization"](https://openreview.net/forum?id=JaTmg8FX3k&referrer=%5BReviewers%20Console%5D(%2Fgroup%3Fid%3DICLR.cc%2F2025%2FConference%2FReviewers%23assigned-submissions)) (pareto-pcc2021) which is a pareto bandits paper. The algorithms presented in pareto-pcc2021 overlay a framework for explicit user preference onto the traditional pareto bandits framework. Their running example (Figure 1, Section 1, pg2) is that of recommending restaurants to users. In the decision-making, the dual objectives of cost (known per person rate) and taste (uncertain quality) need to be honored. The authors of pareto-pcc2021 propose a general bandit model with explicit user preferences. We argue that MAB-CS may allow for a more sample-efficient solution to this problem. This is because explicit user preferences have to be learned by pre-training on expensive per user-interaction data, whereas for MAB-CS all we require is a constraint on taste/quality.
> >
> > 4. In the world of healthcare cost of treatment is always a consideration and MAB-CS captures the solution to these decision making problems. While the best available course of treatment can be administered under abundance, minimizing cost of treatment is a realistic concern that has to be dealt with in times of crisis such as pandemics. The quality threshold can represent an acceptable level of treatment where the survival or adequate quality of life post treatment can be ensured, and our MAB-CS policies represent a promising solution.
> >
> > ---
> >
> > We would be more than happy to continue the discussion if the reviewer believes any concerns or apprehensions have not yet been addressed! In the meantime, in the light of the many strengths of our work, we would greatly appreciate it if the reviewer considered increasing their score.
> >
> > ---
> >
> > Citations
> >
> > [1] Sinha, Deeksha, et al. "Multi-armed bandits with cost subsidy." International Conference on Artificial Intelligence and Statistics. PMLR, 2021.
> >
> > [2] Wu, Yifan, et al. "Conservative bandits." International Conference on Machine Learning. PMLR, 2016.
> >
> > [3] Xu, Mengfan, and Diego Klabjan. "Pareto regret analyses in multi-objective multi-armed bandit." International Conference on Machine Learning. PMLR, 2023.
> >
> > [4] Badanidiyuru, Ashwinkumar, Robert Kleinberg, and Aleksandrs Slivkins. "Bandits with knapsacks." Journal of the ACM (JACM) 65.3 (2018): 1-55.
> >
> > [5] Anonympus authors. "Provably Efficient Multi-Objective Bandit Algorithms under Preference-Centric Customization." Under review at ICLR 2025.

---

> > > ### Author Response · Authors · 2024-12-02
> > > **Response requested**
> > >
> > > ### Response requested from reviewer wsgZ
> > >
> > > ---
> > >
> > > We were eager to hear the reviewers thoughts on our rebuttal response to their recent review.
> > >
> > > While we realize that it had been less than a day since we posted our response we wanted to share a reminder since there was only about a day left to the deadline for reviewer response comments.
> > >
> > > We would be more than happy to follow up to any additional comments, observations, or concerns the reviewer may have. If the reviewer finds that our response adequately addresses their concerns then we would ask them to consider a score increase from their initial assessment.
> > >
> > > ---

---

> > > > ### Comment · Reviewer_wsgZ · 2024-12-02
> > > >
> > > > Thanks for your response. The score is increased to 6.

---

### Author Response · Authors · 2024-11-20
**Update 1: Lower bounds for MAB-CS problem**

As our first update in response to the limitations and criticisms stated by reviewers fMRh, MwCh, and CGLm (particularly the latter two among our three reviewers) we would like to share a new section from the revised Appendix of the paper working out instance-dependent lower bounds on the number of samples of sub-optimal arms (arms other than the best action $a^*$).

These bounds are available in a PDF as the sole supplementary materials file. This update is not yet reflected in the paper PDF which remains the pre-rebuttal version.

We would like to add that the proved lower bounds on samples match the upper bounds for our PE (up to a constant factor) in the known reference arm $\ell$ setting. For the full cost subsidy case, our PE-CS still matches the lower bounds (up to a constant factor) for arms cheaper than the optimal action (a.k.a best action $a^*$) and in some cases (which have been worked out explicitly) for the arm $i^*$.
However for arms $i > a^*$ and different from $i^*$ (i.e. that is the so-called high cost arms) there seems to be a room for improvement in PE-CS which we have tabled as future work.

---

> ### Author Response · Authors · 2024-11-26
>
> Update: Supplementary PDF deleted, lower bounds incorporated in main paper and paper PDF updated.
>
> Responses directed to individual reviewers to follow shortly.

---

### Author Response · Authors · 2024-12-04
**Author’s post reviews summary**

### Author’s post reviews summary

---

This paper takes on a MAB framework with important applications and introduces two novel extensions to it. We propose algorithms for each of the three formulations with regret upper and lower bounds. Novel technical contributions include the first instance-dependent regret lower bounds for the original CS formulation, order-optimal algorithm for known reference arm case, and an algorithm for the original CS setting with a logarithmic instance dependent regret upper bound. Our algorithm outperforms baselines from the literature either outright (in case of ETC-CS) or when performance and reliability are both taken into account (in case of UCB-CS, TS-CS).

We are glad that we were able to satisfactorily address the concerns of all reviewers, two of which increased their scores from their initial assessments. Importantly our work showed that our algorithm PE was order-optimal on cost and quality regret for the known reference arm setting. Moreover PE-CS, the generalization of PE that leverages a best-arm-identification stage, was shown to be order-optimal on the expected number of samples of a certain subset of arms. Future work inspired by our paper would develop elimination based regret minimization algorithms that close the order-gap for the remaining arms.

---

---

### Meta-Review · Area_Chair_YyTZ · 2024-12-21

**Metareview:**

This paper explores the Multi-Armed Bandits with Cost Subsidy (MAB-CS) framework, a problem of practical importance where decisions minimize costs under reward constraints. The authors propose two algorithms: Pairwise Elimination (PE) for cases with a known reference arm and its extension, PE-CS, for settings where the optimal reference arm is unknown. They provide instance-dependent guarantees on cost and quality regret, achieving logarithmic bounds. Experiments on real-world datasets like MovieLens demonstrate the algorithms' effectiveness.

The authors addressed critical reviewer concerns during the rebuttal. They provided instance-dependent lower bounds for both the known reference arm and general MAB-CS settings, strengthening the theoretical foundation. They revised the presentation to clarify notations, objectives, and experimental setup. New experiments were conducted, exploring variations in parameters across multiple datasets, including a newly added Goodreads dataset. These enhancements addressed criticisms about limited experimentation and strengthened confidence in the algorithms' robustness and generalizability.

The detailed responses effectively addressed the reviewers' comments, leading to raised scores. Considering the above novelties, I recommend acceptance of the paper once the authors incorporate all the reviewer's feedback in the final version.

**Additional Comments On Reviewer Discussion:**

See above

---

### Decision · Program_Chairs · 2025-01-22

Accept (Poster)